# The accuracy of weather radar in heavy rain: a comparative study for Denmark, the Netherlands, Finland and Sweden

Marc Schleiss[1], Jonas Olsson[2], Peter Berg[2], Tero Niemi[3,5], Teemu Kokkonen[3], Søren Thorndahl[4], Rasmus Nielsen[4], Jesper Ellerbæk Nielsen[4], Denica Bozhinova[2], and Seppo Pulkkinen[5,6]

[1]Dept. of Geoscience and Remote Sensing, Delft University of Technology, Netherlands
[2]Hydrology Research Unit, Swedish Meteorological and Hydrological Institute SMHI, Norrkoping, Sweden
[3]Dept. of Built Environment, Aalto University, Finland
[4]Dept. of Civil Engineering, Aalborg University, Denmark
[5]Finnish Meteorological Institute FMI, Helsinki, Finland
[6]Dept. of Electrical & Computer Engineering, Colorado State University, Fort Collins, United States

**Correspondence:** Marc Schleiss (m.a.schleiss@tudelft.nl)

**Abstract.** Weather radar has become an invaluable tool for monitoring rainfall and studying its link to hydrological response. However, when it comes to accurately measuring small-scale rainfall extremes responsible for urban flooding, many challenges remain. The most important of them is that radar tends to underestimate rainfall compared to gauges. The hope is that by measuring at higher resolutions and making use of dual-polarization radar, these mismatches can be reduced. Each country has developed its own strategy for addressing this issue. However, since there is no common benchmark, improvements are hard to quantify objectively. This study sheds new light on current performances by conducting a multinational assessment of radar's ability to capture heavy rain events at scales of 5 min up to 2 hours. The work is performed within the context of the joint experiment framework of project MUFFIN (Multiscale Urban Flood Forecasting), which aims at better understanding the link between rainfall and urban pluvial flooding across scales. In total, 6 different radar products in Denmark, the Netherlands, Finland and Sweden were considered. The top 50 events in a 10-year database of radar data were used to quantify the overall agreement between radar and gauges as well as the bias affecting the peaks. Results show that the overall agreement in heavy rain is fair (correlation coefficient 0.7-0.9), with apparent multiplicative biases in the order of 1.2-1.8 (17-44% underestimation). However, after taking into account the different sampling volumes of radar and gauges, actual biases could be as low as 10%. Differences in sampling volumes between radar and gauges play an important role in explaining the bias but are hard to quantify precisely due to the many post-processing steps applied to radar. Despite being adjusted for bias by gauges, 5 out of 6 radar products still exhibited a clear conditional bias with intensity of about 1-2% per $\text{mmh}^{-1}$. As a result, peak rainfall intensities were severely underestimated (factor 1.8-3.0 or 44-67%). The most likely reason for this is the use of a fixed Z-R relationship when estimating rainfall rates (R) from reflectivity (Z), which fails to account for natural variations in raindrop size distribution with intensity. Based on our findings, the easiest way to mitigate the bias in times of heavy rain is to perform frequent (e.g., hourly) bias adjustments with the help of rain gauges, as demonstrated by the Dutch C-band product. An even more promising strategy that does not require any gauge adjustments is to estimate rainfall rates using a combination of reflec-

tivity (Z) and differential phase shift (Kdp), as done in the Finnish OSAPOL product. Both approaches lead to approximately similar performances, with an average bias (at 10 min resolution) of about 30% and a peak intensity bias of about 45%.

## 1 Introduction

The ability to measure short-duration, high-intensity rainfall rates is of paramount importance in predicting hydrological response. Indeed, several studies have shown that the resolution of the rainfall data directly impacts the shape, timing and peak flow of hydrographs (Aronica et al., 2005; Löwe et al., 2014; Ochoa-Rodriguez et al., 2015; Rico-Ramirez et al., 2015; Cristiano et al., 2017). Previous research has shown that in order to obtain reliable results in small urban catchments, the rainfall data should have a resolution of at least 10 min and 1 km (Schilling, 1991; Ogden and Julien, 1994; Berne et al., 2004). If the resolution is insufficient compared with what is needed for the runoff simulations, the accuracy of flood predictions is likely to be compromised (Andréassian et al., 2001; Aronica et al., 2005; Bruni et al., 2015; Rafieeinasab et al., 2015).

Another important issue besides resolution is the accuracy of the rainfall data themselves. Currently, only weather radar offers the spatial coverage, resolution and accuracy needed to study the complex link between the spatio-temporal characteristics of rain events and hydrological response (Wood et al., 2000; Berne et al., 2004; Smith et al., 2007; He et al., 2013; Thorndahl et al., 2017). The most common application of radar in hydrology is the study and characterization of heavy rain events associated with flooding (Baeck and Smith, 1998; Delrieu et al., 2005; Collier, 2007; Ntelekos et al., 2007; Anagnostou et al., 2010; Villarini et al., 2010; Wright et al., 2012; Zhou et al., 2017). However, there have been many other successful applications of radar in urban hydrology, such as generating detailed runoff predictions or creating flood maps (Wright et al., 2014; Thorndahl et al., 2016; Yang et al., 2016). Steady progress in radar technology over the past decades and in particular the switch from single to dual-polarization has lead to significant progress in terms of clutter suppression, hydrometeor classification and attenuation correction, greatly improving the accuracy of radar rainfall estimates (Zrnic and Ryzhkov, 1996; Ryzhkov and Zrnic, 1998; Zrnic and Ryzhkov, 1999; Bringi and Chandrasekar, 2001; Gourley et al., 2007; Matrosov et al., 2007). Polarimetry also fundamentally changed the way we estimate rainfall from radar measurements, with traditional Z-R power law relationships being increasingly replaced by alternative methods based on differential phase shift (Ryzhkov and Zrnic, 1996; Zrnic and Ryzhkov, 1996; Brandes et al., 2001; Matrosov et al., 2006; Otto and Russchenberg, 2011). This has promoted the development of smaller, cheaper and higher-resolution X-band polarimetric radars for use in urban flood forecasting (Wang and Chandrasekar, 2010; Ruzanski et al., 2011). The hope is that by moving to higher resolutions and taking advantage of dual-polarization, the accuracy of radar-based rainfall estimates and flood predictions will increase. However, this is a delicate process as higher resolution and more elaborate retrieval algorithms also increase sampling uncertainty. A higher resolution therefore does not automatically translate into more accurate rainfall estimates (Krajewski and Smith, 2002; Seo et al., 2015; Cunha et al., 2015). Also, the space/time correlation structure of radar errors and their dependence on precipitation type and distance to the radar means that there are practical limits to what can be achieved in terms of predictive skill in hydrological models (Rafieeinasab et al., 2015; Courty et al., 2018).

Despite decades of research, quantifying individual errors and biases in radar retrievals remains hard (Einfalt et al., 2004; Lee, 2006; Krajewski et al., 2010; Villarini and Krajewski, 2010; Berne and Krajewski, 2013). One aspect that is still poorly documented concerns the overall accuracy of radar in times of heavy rain. Because radar hardware, software and data processing techniques are subject to frequent replacements and updates, most homogeneous radar records currently available for analysis only span 10-15 years. This is likely to improve in the future thanks to open data policies and the automatic exchange of radar data between countries, such as OPERA (Huuskonen et al., 2014; Saltikoff et al., 2019). However, until now, datasets are limited and studies have mostly looked at performances of individual radar systems and/or national networks. The few results that are available suggest that radar tends to underestimate rainfall peaks compared with rain gauges (Smith et al., 1996; Overeem et al., 2009a; Smith et al., 2012; Peleg et al., 2018). For example, based on a 12-year archive of $1 \times 1$ km and 5-min radar rainfall estimates for Belgium, Goudenhoofdt et al. (2017) found that hourly radar extremes around Brussels tend to be 30-70% lower than those observed in gauge data. The underestimation is partly attributed to differences in sampling volumes between radar and gauges. But other factors such as calibration issues, range effects, signal attenuation or saturation of the receiver channel can also play a role. At very high resolutions (e.g., 5 min and 1 km), wind effects and vertical variability of rainfall can also introduce substantial biases between radar and gauge measurements (Dupasquier et al., 2000; Vasiloff et al., 2009; Dai and Han, 2014). Another series of studies in the Netherlands showed that in principle, it is possible to derive robust intensity-duration-frequency curves (Overeem et al., 2009b, a) and areal extremes (Overeem et al., 2010) from long radar data archives. However, the authors clearly mention that the radar data need to be carefully quality controlled and bias corrected first.

Since radar measurements are inherently prone to errors and knowledge about microphysical processes in clouds and rain is limited, post-processing plays an important role. In addition to using better hardware, many weather services now offer gridded, quantitative rainfall products that combine measurements from different radar systems and have been corrected for various types of biases using rain gauges and other sources of information such as elevation, cloud cover and satellite imagery (Krajewski, 1987; Smith and Krajewski, 1991; Goudenhoofdt and Delobbe, 2009; Delrieu et al., 2014; Stevenson and Schumacher, 2014). During post-processing, many systematic biases due to attenuation, calibration, vertical variability and range effects are mitigated (e.g., Collier and Knowles, 1986; Young et al., 2000; Gourley et al., 2006; Overeem et al., 2009b; Delrieu et al., 2014; Berg et al., 2016). However, rain gauge data also contain errors and biases, the most important of which is an underestimation of the rainfall intensity due to local wind effects. For regular events, errors usually remain in the order of 5-10%. However, during heavy rain evens, wind-induced biases can exceed 30% (Nystuen, 1999; Sieck et al., 2007; Pollock et al., 2018). As a result, post-processed radar products might still contain important residual errors (Krajewski et al., 2010). For example, Smith et al. (2012), Wright et al. (2014), Thorndahl et al. (2014b) and Cunha et al. (2015) highlighted several major quality issues affecting post-processed quantitative precipitation estimates from NEXRAD, including range-dependent and intensity-dependent biases. Quantifying these residual errors and studying their propagation in hydrological models is crucial for improving the timing and accuracy of flood predictions (Cunha et al., 2012; Bruni et al., 2015; Courty et al., 2018; Niemi et al., 2017). For example, in their study, Stransky et al. (2007) estimated that the propagation of biased radar measurements

in urban drainage models could result in up to 30-45% errors in terms of peak flow magnitude. To limit error propagation, Schilling (1991) recommended that the bias affecting areal-averaged rainfall intensities should not exceed 10%.

Over the years, each country has developed its own strategy for mitigating errors and biases in operational radar rainfall estimates. However, since there is no common benchmark and few international studies are available, the merits and weaknesses of each approach remain difficult to quantify objectively. This study sheds new light on current performances by conducting a multinational assessment of radar's ability to capture heavy rain events at scales of 5 min up to 2 hours. In total, 6 different radar products across 4 European countries (i.e., Denmark, the Netherlands, Finland and Sweden) are considered. Special emphasis

is put on analyzing the performance during the 50 most intense events over the last 10-15 years. By comparing different types of radar products (C-band vs X-band, single vs dual-polarization) and identifying the main sources of errors and biases across scales, important recommendations about how to improve the accuracy of quantitative precipitation estimates for flash flood prediction and urban pluvial flooding can be drawn. The rest of this paper is organized as follows: Section 2.1 explains the methodology used to select events and extract the gauge and radar data. Section 2.2 gives a detailed description of the radar

products used for the analysis. Section 2.3 introduces the statistical models used to quantify the bias between gauges and radar. Section 3 presents the results and Section 4 summarizes the main conclusions.

## 2   Data & Methods

### 2.1   Event selection and data extraction methods

Event selection was done based on rainfall time series from the national networks of automatic rain gauges in Denmark, the

Netherlands, Finland and Sweden. Due to data availability and quality, only a small subset of all the existing gauges was used for analysis (i.e., 66 gauges for Denmark, 35 for the Netherlands, 64 for Finland and 10 for Sweden). Table 1 provides an overview of the number of gauges used, their temporal resolutions and length of the observational records for each country. Note that Denmark has two separate rain gauge networks. The first is operated by the Danish Meteorological Institute DMI and consists of OTT Pluvio2 weighing gauges (Vejen, 2006; Thomsen, 2016). The second belongs to the Water Pollution

Committee of the Society of Danish Engineers and consists of RIMCO tipping bucket gauges (Madsen et al., 1998, 2017). For this study, only the RIMCO tipping buckets were used. In the Netherlands, precipitation is measured using the displacement of a float in a reservoir (KNMI, 2000). The 10-min data from 2008-2018 used in this study have been validated internally by the Royal Netherlands Meteorological Institute KNMI using a combination of automatic and manual quality control tests. In Finland, weighting gauges of the type OTT Pluvio2 are used. Observations are made using a wind protector according

to World Meteorological Organization regulations (WMO, 2008). Automatic quality control tests are used to flag suspicious values which are then double checked manually by human experts. In Sweden, gauges are vibrating wire load sensors of the type GEONOR with an oil film to keep evaporation at very low amounts.

    Based on the available gauge data, the top 50 rain events (in terms of peak intensity) were determined for each country and observation period. For every gauge, a continuous 6 hour dry period was used to separate events from each other. This

was done separately for each gauge which means that some events were included multiple times into the dataset given that

they were observed by different gauges at different locations. To ensure quality, each identified event was subjected to a visual quality control test by human experts, making sure the rainfall rates recorded by the gauges and the radar (see Section 2.2) were plausible and consistent with each other in terms of their temporal structure. Cases for which the gauge or radar data were incomplete, obviously wrong or inconsistent with each other were removed and replaced by new events until the total number of events that passed the quality control tests reached 50 for each country. Overall, about 10% of the originally identified events had to be removed and replaced by new ones during these quality control steps, most of them because of incomplete or erroneous radar data.

The radar data for each country were extracted according to the following procedure. First, the 4 radar pixels closest to a given rain gauge were extracted. The 4 radar rainfall time series were then aggregated in time (i.e., averaged) to match the temporal sampling resolution of the considered rain gauge. Then, for each time step, the value among the 4 radar pixels that best matched the gauge was kept for comparison. The motivation behind this type of approach is that it can account for small differences in location and timing between radar and gauge observations due to motion, wind and vertical variability (Dai and Han, 2014). Note that this is a rather conservative and favorable way of comparing gauges with radar that leads to smaller overall discrepancies and more robust results than pixel-by-pixel comparisons. Other less favorable ways of extracting the radar data were also tested (e.g., using inverse distance weighted interpolation or the maximum value among the nearest neighbors). However, these only resulted in higher discrepancies and did not change the main conclusions and were therefore abandoned in subsequent analyses.

Figure 1 shows a map with the location of all rain gauges used for the final, quality-controlled rain event catalog for each country. As can be seen in Figure 2, the final catalog includes a large variety of rain events, ranging from single isolated convective cells to large organized thunderstorms and mesoscale complexes. Additional tables summarizing the starting time, duration, amount and peak rainfall intensity for each event and country are provided in the Appendix (see Tables A1-A5). Because events were selected based on peak intensity, it is not surprising to see that all of them occurred in the warm season between May and September during which convective activity is at its maximum (see Figure 3). Similar analyses confirm that the events mostly occurred during the afternoon and late evening hours, in agreement with the diurnal cycle of convective precipitation and rainfall intensity at mid-latitudes (Rickenbach et al., 2015; Blenkinsop et al., 2017; Fairman et al., 2017).

## 2.2 The radar products

This section gives a brief overview of the different radar products used for the analyses. A short summary of the most important characteristics of each product is provided in Table 2.

### 2.2.1 Radar data for Denmark

The weather radar network of the Danish Meteorological Institute (DMI) operates four 5.625 GHz C-band pulse radars with 1 degree beam width and 250 kW peak power located in Rømø, Sindal, Stevns, Virring and Bornholm (Gill et al., 2006; He et al., 2013). New dual-polarization radars have been installed at all sites between 2008 and 2017. However, for this study, only the single-polarization data from the Stevns radar were used. The latter is located near the coast, at 55.326°N 12.449°E and 53 m

elevation, approximately 40 km south of Copenhagen in an area of relatively flat topography with altitudes ranging from -7 m to 125 m above mean sea level. It was purchased in 2002 from Electronic Enterprise Corporation (EEC) and is operated using a combination of EEC and DMI software. The scanning strategy involves collecting reflectivity measurements at 9 different elevation angles of 0.5, 0.7, 1.0, 1.5, 2.4, 4.5, 8.5, 13.0 and 15.0 degrees with a range resolution of 500 m and a maximum range of 240 km. The reflectivity measurements $Z$ [dBZ] at these 9 elevations are projected to a pseudo-constant altitude plan position indicator (PCAPPI) at 1000 m height to generate a high-resolution gridded product with 10 min temporal resolution and 500$\times$ 500 m$^2$ grid spacing (Gill et al., 2006). The temporal resolution of the PCAPPI is then statistically enhanced to 5 min using an advection interpolation scheme (Thorndahl et al., 2014a; Nielsen et al., 2014). Ground clutter in the PCAPPI is removed by filtering out echoes with Doppler velocity smaller than 1 ms$^{-1}$. Rainfall-induced attenuation $K$ is estimated as $K = 6.9 \cdot 10^{-5} Z^{0.67}$ [dBZ km$^{-1}$] and attenuation-corrected reflectivity estimates are converted to rainfall rates $R$ based on a fixed Marshall-Palmer Z-R relationship given by $Z = 200 R^{1.6}$. To take into account calibration errors and variations in raindrop size distributions, a daily mean field bias correction is applied to the high-resolution radar rainfall estimates based on the measurements from a network of 66 RIMCO tipping bucket rain gauges in the region operated by the Water Pollution Committee of the Society of Danish Engineers (Madsen et al., 1998, 2017). Note that the final 500 m, 5 min bias-corrected product used in this study is not operational but developed for research purposes by Aalborg University.

### 2.2.2 Radar data for the Netherlands

The used product is a 10-year archive of 5 min precipitation depths at 1$\times$1 km$^2$ spatial resolution based on a composite of radar reflectivities from 2 C-band radars in De Bilt and Den Helder operated by the Royal Netherlands Meteorological Institute (KNMI). Note that the Netherlands recently upgraded their radars to dual-polarization. However, the dual-polarization rainfall estimates are not fully operational yet and all radar rainfall estimates used in this study were produced with the single-polarization algorithms. Also, the radar in De Bilt stopped contributing to the composite in the course of January 2017, at which point it was replaced by a new polarimetric radar in the nearby village of Herwijnen. For a detailed description of the processing chain, the reader is referred to Overeem et al. (2009b). The radars used in this study were two single-polarization Selex (Gematronik) METEOR 360 AC Pulse radars with a wavelength of 5.2 cm, peak power of 365 kW, pulse repetition frequency of 250 Hz and 3-dB beam width of 1 degree. The scanning strategy consists of four azimuthal scans of 360 degrees at 4 elevation angles of 0.3, 1.1, 2.0, and 3.0 degrees. The data from these scans are combined into 5-min PCAPPI at 800 m height according to the following procedure: for distances up to 60 km from the radar, only the highest elevation angle is used to reduce the risk of ground clutter and beam blockage. For distances of 15-80 km from the radar, the PCAPPI is constructed by bilinear interpolation of the reflectivity values (in dBZ) of the nearest elevations below and above the 800-m height level. For distances of 80-200 km from the radar, only the reflectivity values of the lowest elevation angle are used, whereas it should be pointed out that the 800 m level only stays within the 3-dB beam width of the lowest elevation up to a range of about 150 km. Values beyond 200 km from the radar are ignored. Once the PCAPPI have been constructed, ground clutter and anomalous-propagation are removed using the procedure of Wessels and Beekhuis (1995) also described in Holleman and Beekhuis (2005). Spurious echoes within a radius of 15 km from the radar are mitigated based on the procedure described in

Holleman (2007). A fixed Marshall-Palmer Z-R relation of $Z = 200R^{1.6}$ is used to convert the reflectivities in the PCAPPI to rainfall rates. During the conversion, reflectivity values are capped at 55 dBZ to suppress the influence of echoes induced by hail or strong residual clutter. Because of this, the maximum rainfall rate that can be estimated with this approach is 154 mm/h. Individual rainfall estimates from the two radars are then combined into one final composite using a weighting factor as a function of range from the radar, as described in Eq. 6 of Overeem et al. (2009b). During the compositing, accumulations close to the radar are assigned lower weights to limit the impact of bright bands and spurious echoes. The composited rainfall rates are then adjusted for bias on an hourly basis using a network of 32 automatic rain gauges at 10 min resolution and 322 manual gauges at daily resolutions following the procedures of Holleman (2007) and Overeem et al. (2009b). Note that the additional bias correction at daily timescale (downscaled to 10 min scales) is primarily used to improve the large-scale spatial consistency of the radar and gauge estimates and is therefore not extremely important in the context of this study.

### 2.2.3 Radar data for Finland

The Finnish radar product is an experimental product from the Finnish Meteorological Institute (FMI) OSAPOL-project, which differs from the operational product used by the FMI mainly by making a better use of dual-polarization. The product is based on the data from the years 2013-2016, during which the old single-polarization radars were being replaced by C-band dual-polarization Doppler radars. The product is therefore based on data from 4-8 dual-polarization radars depending on how many were available each year. The beam width is 1 degree, range resolution is 500 m and the scanning is done in Pulse Pair Processing (PPP) mode. Doppler filtering is done first in the signal processing stage, and reflectivity measurements are calibrated based on solar signals (Holleman et al., 2010). Next, non-meteorological targets are removed using statistical clutter maps and fuzzy-logic-based HydroClass classification by Vaisala (Chandrasekar et al., 2013). The reflectivity $Z$ is attenuation-corrected (Gu et al., 2011), and the differential phase shift Kdp is estimated using the method described in Wang and Chandrasekar (2009). For hydrometeors classified as liquid precipitation, two alternative rain rate conversions are used. For heavy rain, i.e., Kdp>0.3 and Z>30 dBZ, the R(Kdp) relation given by $R = 21\text{Kdp}^{0.72}$ is used (Leinonen et al., 2012). For low to moderate intensities, i.e., Kdp≤0.3 or Z≤30 dBZ and for radar bins where HydroClass indicates non-liquid precipitation, a fixed Z(R) relation given by $Z = 223R^{1.53}$ is used (Leinonen et al., 2012). Using the estimated rainfall rates at the 4 lowest elevation angles, a PCAPPI at 500 m height is produced using inverse distance-weighted interpolation with a Gaussian weight function. Finally, a composite VPR correction map (Koistinen and Pohjola, 2014) is applied to the PCAPPI to generate a $1 \times 1$ km$^2$ and 5 min resolution product. The OSAPOL is the only radar product in this study that is not gauge-adjusted.

### 2.2.4 Radar data for Sweden

The considered product is the so-called BRDC (BALTEX Radar Data Center) produced by SMHI. It is a $2 \times 2$ km, 15 min composite product of PCAPPIs sourced from 12 operational single-polarization C-band Doppler radars in Sweden between the years 2007 and 2016 (see Figure 1 in Norin et al. (2015)). After that, the product was discontinued and replaced by the newer BALTRAD product (Michelson et al., 2018). Note that Swedish radars are being used for real-time operational production, and therefore prone to frequent changes and re-tuning. For example, the beam width of the radars has changed over time due

to hardware upgrades. Also, the scanning strategies, filters and processing chains have been updated several times. Describing all these changes is not feasible within the context of this study. Therefore, the differences between gauge and radar estimates in Sweden include both a technical component (related to the hardware and number of radars) and a component related to the operation strategies over the years (i.e., human and algorithm). The technical aspects of the quantitative precipitation estimation in the BRDC product are explained in Section 2.2 of Norin et al. (2015). Azimuthal scans of reflectivity measurements at up to 10 different elevation angles between 0.5 and 40 degrees are projected into a PCAPPI at 500 m height. Ground clutter is removed by filtering all echoes with radial velocities less than 1 ms$^{-1}$. Remaining non-precipitation echoes are removed by applying a consistency filter based on satellite observations (Michelson, 2006). The effect of topography is accounted for by applying a beam blockage correction scheme described in Bech et al. (2003). Rainfall rates on the ground are estimated from the PCAPPI through a constant Marschall-Palmer Z-R relationship Z=200R$^{1.6}$. To reduce errors and biases, a method called HIPRAD (HIgh-resolution Precipitation from gauge-adjusted weather RADar) is applied (Berg et al., 2016). The latter was developed to make radar data more suitable for hydrological modeling by applying 30-day mean correction factors to correct for mean field biases and range dependent biases. Note that although several radars are available in Sweden, the system is currently set up such that each radar has a predetermined non-overlapping measurement area. The final radar-estimated rainfall rates at each location are therefore obtained by only taking into account the data from a single radar (i.e., usually the nearest one) and no attempt is made to take advantage of possibly overlapping measuring areas (except for bias-correction using gauges). Better radar compositing methods are currently being developed at SMHI but are not yet implemented operationally.

### 2.2.5 Additional radar products

In addition to the 4 main radar products described above, two additional datasets were considered. These are not the main focus of the paper and are only used to provide additional insights and help with the interpretation of the results. The first additional radar dataset is from a FURUNO WR-2100 dual-polarization X-band Doppler research radar system located in Aalborg, Denmark. The radar performs fast azimuthal scans at 6 different elevation angles in a radius of about 40 km around Aalborg with a high spatial resolution of $100\times100$ m$^2$ and temporal sampling resolution of 1 min. However, for this study, only the data from a single elevation angle (i.e., 4°) were used. Clutter is removed by applying a filter on the Doppler velocities and a spatial texture filter on reflectivity. Rainfall rates are estimated using a fixed Z-R relationship given by Z = 200R$^{1.6}$ (after attenuation correction). Similarly to the Danish C-band product, all rainfall rates are corrected for daily mean field bias using RIMCO tipping bucket rain gauges. Only two years of X-band radar measurements between 2016-2017 are available for analysis. Consequently, only the 10 most intense events were considered. Despite these limitations, the X-band data can be used to provide valuable insight into the advantages and challenges associated with using high-resolution X-band radar measurements in times of heavy rain.

The second additional radar product used in this study is an international composite at 15 min temporal and $2\times2$ km$^2$ spatial resolution derived from the BALTRAD collaboration (Michelson et al., 2018). The BALTRAD is almost identical to the BRDC product used in Sweden. The main difference is that it covers a much larger area and does not include the HIPRAD bias adjustments. Instead, bias correction in the BALTRAD is done by taking each 15-min time step and scaling it with the

ratio of 30-day aggregation of gauge and radar accumulations. The extended coverage in the BALTRAD product is made possible thanks to the automatic exchange of radar data between neighboring countries around the Baltic sea (i.e., Norway, Sweden, Finland, Estonia, Latvia and Denmark). The fact that the BALTRAD product spans multiple countries makes it particularly interesting for evaluating and comparing performances with respect to tailored national products. This means that direct comparisons with the BALTRAD are available for (most of) the top 50 events identified in Denmark, Finland and Sweden. Unfortunately, the Netherlands are currently not part of BALTRAD which means that no further comparisons are possible for the Dutch C-band product.

## 2.3 Comparison of radar and gauge measurements

Since radar and gauges measure rainfall at different scales using different measuring principles, one can not expect a perfect agreement between the two. Gauges are more representative of point rainfall measurements on the ground while radar provides averages over large resolution volumes several hundreds of meters above the ground. In addition, each sensor has its own measurement uncertainty and limitations in times of heavy rain. Gauges are known to underestimate intensity by up to 25-30% in heavy rain and windy conditions (e.g., Nystuen, 1999; Chang and Flannery, 2001; Ciach, 2003; Sieck et al., 2007; Goudenhoofdt et al., 2017; Pollock et al., 2018). On the other hand, radar is known to suffer from signal attenuation, non-uniform beam filling, clutter, hail contamination and overshooting (Krajewski et al., 2010; Villarini and Krajewski, 2010; Berne and Krajewski, 2013). Missing data in one or both of the sensors also further complicate the comparison (Vasiloff et al., 2009). Therefore, the main goal here will not be to make a statement about which sensor comes closest to the truth but to quantify the average discrepancies between the gauge and radar measurements as a function of the event, time scale, intensity and radar product. Such information can be useful to monitor the performance and consistency of operational radar and gauge products or study the propagation of rainfall uncertainties in hydrological models (Rossa et al., 2011).

### 2.3.1 Bias estimation

Discrepancies between radar and gauge observations are assessed with the help of a multiplicative error model:

$$R_r(t) = \beta \cdot R_g(t) \cdot \varepsilon(t) \tag{1}$$

where $R_r(t)$ (in mmh$^{-1}$) denote the radar measurements a time $t$, $R_g(t)$ (in mmh$^{-1}$) the gauge measurements, $\beta$ [-] the multiplicative bias and $\varepsilon(t)$ [-] are independent, identically distributed random errors drawn from a log-normal distribution with median 1 and scale parameter $\sigma_\varepsilon > 0$ (Smith and Krajewski, 1991). The multiplicative bias in Equation (1) can also be expressed in terms of the log-ratios of radar versus gauge values:

$$\ln\left(\frac{R_r(t)}{R_g(t)}\right) = \ln(\beta) + \ln(\varepsilon(t)) \tag{2}$$

where $\ln(\varepsilon(t))$ is a Gaussian random variable with mean 0 and variance $\sigma_\varepsilon^2$. Equation (2) can be used to detect the presence of conditional bias with intensity by checking whether the expected value of the log-ratio $\ln\left(\frac{R_r(t)}{R_g(t)}\right)$ depends on $R_g(t)$ or not. Note that the multiplicative bias model in Equations (1) and (2) has been shown to provide a better, physically more

plausible representation of the error structure between in-situ and remotely-sensed rainfall observations than the classical additive bias model used in linear regression (e.g., Tian et al., 2013). It assumes that the discrepancies between radar and gauge measurements are the result of two error contributions: a deterministic component $\beta$ that accounts for systematic errors in radar and gauge measurements (e.g., due to calibration, wind effects, wrong Z-R relationship, ...) and a random term $\varepsilon(t)$ that represents sampling errors and noise in radar and gauge observations. Since gauges are not seen as ground truth in this study, $\varepsilon(t)$ is assumed to contain all possible sources of errors in both the gauge and radar observations, including the ones due to differences in sampling volumes (Ciach and Krajewski, 1999b). The last point is particularly important as radar sampling volumes can be up to 7 orders of magnitude larger than that of rain gauges (Ciach and Krajewski, 1999a). This means that even if both sensors would be perfectly calibrated, their measurements would still disagree with each other due to the fact that rain gauge measurements made at a particular location within a radar pixel are usually not representative of averages over larger areas. In their paper, Ciach and Krajewski (1999a) proposed a rigorous statistical framework for assessing this representativeness error based on the spatial autocovariance function and the notion of extension variance. However, their approach was developed for an additive error model and can not be directly applied here. Instead, we propose a comparatively simpler approach in which the differences in sampling volumes are already included in the random errors $\varepsilon(t)$. Our approach is based on the assumption that the errors $\varepsilon(t)$ have a log-normal distribution with median 1 and scale parameter $\sigma_\varepsilon > 0$, which means that we must have $\mathbb{E}[\varepsilon(t)] = \exp(\frac{\sigma_\varepsilon^2}{2}) \neq 1$. Furthermore, if we assume that $R_g(t)$ and $R_r(t)$ are second-order stationary random processes with fixed mean $\mu_g$ and $\mu_r$ and variances $\sigma_g^2$ and $\sigma_r^2$ and that the random errors $\varepsilon(t)$ are identically distributed and independent from $R_g(t)$, then we get the following system of equations:

$$
\begin{cases}
\mathbb{E}[R_g(t)] &= \beta \cdot \mathbb{E}[R_r(t)] \cdot \mathbb{E}[\varepsilon(t)] = \beta \cdot \mu_r \cdot \exp(\frac{\sigma_\varepsilon^2}{2}) \\
\mathrm{Var}[R_g(t)] &= \beta^2 \cdot \mathrm{Var}[R_r(t)] \cdot \mathrm{Var}[\varepsilon(t)] = \beta^2 \cdot \sigma_r^2 \cdot \exp(\sigma_\varepsilon^2) \cdot \left(\exp(\sigma_\varepsilon^2) - 1\right)
\end{cases}
\tag{3}
$$

From the first equation we get $\beta^2 = \frac{\mu_g^2}{\mu_r^2} \cdot \exp(-\sigma_\varepsilon^2)$ which can be plugged into the second equation to get an estimate of the scale parameter $\hat{\sigma}_\varepsilon$:

$$
\hat{\sigma}_\varepsilon^2 = \ln\left(1 + \frac{\sigma_g^2 \mu_r^2}{\sigma_r^2 \mu_g^2}\right) = \ln\left(1 + \frac{\mathrm{CV}_g^2}{\mathrm{CV}_r^2}\right).
\tag{4}
$$

where $\mathrm{CV}_{g|r} = \frac{\sigma_{g|r}}{\mu_{g|r}}$ denotes the coefficient of variation of the gauge and radar values respectively. Substituting, we get the following estimate for $\beta$:

$$
\hat{\beta} = \frac{\mu_g}{\mu_r} \cdot \exp(-\frac{\hat{\sigma}_\varepsilon^2}{2}).
\tag{5}
$$

The first term $\frac{\mu_g}{\mu_r}$ in Equation (5) is known as the G/R ratio (Yoo et al., 2014) and it quantifies the apparent bias between radar and gauge measurements. The second term $\exp(-\frac{\hat{\sigma}_\varepsilon^2}{2})$ is a bias adjustment factor that accounts for the fact that gauge and radar measurements do not have the same mean and variance (e.g., due to differences in sampling volumes and/or different measurement uncertainties). The actual underlying model bias $\beta$ is obtained by multiplying the two terms together. However,

it is important to keep in mind that only the G/R ratio is directly observable from the data while $\beta$ is a theoretical bias that heavily depends on the assumptions that the errors are log-normally distributed with median 1 and independent from the radar observations. To avoid any confusion, the following terminology is adopted:

- The "apparent" bias (i.e., seemingly real or true, but not necessarily so) is the one that we see in the data. It is measured using the G/R ratio.

- The "actual" bias (i.e., existing in fact; real) is the unknown underlying bias, i.e., the bias that we would measure if radar and gauges would have the same sampling volumes. The actual bias is always unknown. The best we can do is approximate it with the help of a statistical model.

Note that $\sigma_\varepsilon$ and $\beta$ could also be estimated through Equation (2) by calculating the mean and standard deviation of $\ln\left(\frac{R_g(t)}{R_r(t)}\right)$. However, this approach is not recommended as the ratios for small rainfall rates can be very noisy and numerical errors will arise whenever one of the measurements is zero.

For readers not familiar with the interpretation of multiplicative biases, note that it is also possible to express the G/R ratio and model bias $\beta$ as an average relative error. In this case, we have:

$$Err_{\text{avg}} = \mathbb{E}\left[\frac{R_g(t) - R_r(t)}{R_g(t)}\right] = 1 - \frac{1}{\beta} \cdot \mathbb{E}\left[\frac{1}{\varepsilon(t)}\right] = 1 - \frac{\exp(\sigma_\varepsilon^2) \cdot \left(\exp(\sigma_\varepsilon^2) - 1\right)}{\beta} \tag{6}$$

where we used the fact that $\frac{1}{\varepsilon(t)}$ is also a log-normal with median 1 and scale parameter $\sigma_\varepsilon$. However, for simplicity and robustness, we prefer to report the median relative error which is independent of the variance of $\varepsilon(t)$:

$$Err_{\text{med}} = \text{Med}\left[\frac{R_g(t) - R_r(t)}{R_g(t)}\right] = 1 - \frac{1}{\beta} \cdot \text{Med}\left[\frac{1}{\varepsilon}\right] = 1 - \frac{1}{\beta} \tag{7}$$

### 2.3.2 Peak intensity bias

Equation (5) provides a convenient way to estimate the average bias between radar and gauge measurements over the course of an event. However, in reality, the bias is likely to fluctuate over time as a function of the spatio-temporal characteristics and intensity of the considered events and their location with respect to the radar(s). Consequently, the G/R ratio and model bias $\beta$ might not necessarily be representative of what happens during the most intense parts of an event. To account for this, we also consider the peak rainfall intensity bias (PIB) between radar and gauges. The PIB is defined as:

$$R_g^{\text{max}} = \text{PIB} \cdot R_r^{\text{max}} \tag{8}$$

where $R_g^{\text{max}}$ and $R_r^{\text{max}}$ denote the maximum rain rate values recorded by the gauges and radar over the course of an event. The PIB values are computed on an event-by-event basis, by aggregating the radar and gauge data to a fixed temporal resolution (using overlapping time windows) and extracting the maximum rain rate over the event at this scale. Note that this is done independently for the gauge and radar time series, which means that the maximum values may not necessarily correspond to the same time interval. The main reason for this is that it leads to more reliable and robust estimate of PIB at high spatial and temporal resolutions and reduces the sensitivity to small timing differences between radar and gauge observations due to wind and vertical variability.

### 2.3.3 Other metrics

To complement the bias analysis and provide a more comprehensive overview of the agreement between gauge and radar measurements, we also calculate standard error metrics such as the Spearman rank correlation coefficient (CC), root mean square difference (RMSD) and relative root mean square difference $\mathrm{RRMSD} = \frac{\mathrm{RMSD}}{\mu_g}$ between gauge and radar values. All these statistics are calculated on an event-by-event basis at a fixed aggregation time scale.

## 3 Results

### 3.1 Agreement during the 4 most intense events

Figure 4 shows the time series of rainfall intensities for the top events in each country (i.e., Denmark, the Netherlands, Finland and Sweden respectively). Each of these events is highly intense, with peak intensities reaching 204 mmh$^{-1}$ in Denmark, 180 mmh$^{-1}$ in the Netherlands, 89.1 mmh$^{-1}$ in Finland and 91.2 mmh$^{-1}$ in Sweden. The July 2, 2011 event in Denmark was particularly violent, affecting more than a million people in the greater Copenhagen region and causing an estimated damage of at least 800 million euros (Wójcik et al., 2013). During the third rainfall peak in Denmark, rain rates remained well above 125 mmh$^{-1}$ for three consecutive 5-min time steps, resulting in more than 41 mm of rain (e.g., about one month's worth of rain for the Copenhagen region). During the same 15 minutes, the radar only recorded 12.1 mm, which is 3.39 times less than what was measured by the gauge. Note that this does not necessarily imply that the radar estimates are wrong, as rain gauge data can also suffer from large biases in times of heavy rain and are not directly comparable to radar due to the large difference in sampling volumes. Nevertheless, all 4 depicted events show a strong, systematic pattern of underestimation by radar compared with the gauges. The G/R ratios, as defined in Equation 5, are 1.66, 1.37, 1.55 and 1.68 respectively, which corresponds to a relative difference in rainfall rates between radar and gauges of 27-40%. This order of magnitude is consistent with previous values reported in the literature. For example Goudenhoofdt et al. (2017) mentioned a 30% underestimation of radar compared with gauges in Belgium and Seo et al. (2015) found up to 50% underestimation on individual events in the United States.

Despite being biased, radar and gauge measurements are rather consistent with each other in terms of their temporal structure (e.g., rank correlation values of 0.92, 0.75, 0.80 and 0.85 for Denmark, the Netherlands, Finland and Sweden respectively). Also, a substantial part of the apparent bias is likely attributable to differences in sampling volumes. According to Equation (5), the bias adjustment factor $e^{-\sigma_\varepsilon^2/2}$ is 0.63, 0.59, 0.66, 0.70 in Denmark, the Netherlands, Finland and Sweden respectively. The actual underlying model bias $\beta$ for the 4 depicted events is therefore estimated to be 1.04, 0.81, 1.02 and 1.18. In other words, once the differences in scale between radar and gauge data have been accounted for, radar only appears to underestimate rainfall rates by a factor 1.04 (3.8%) in Denmark, 1.02 (2.0%) in Finland and 1.18 (15.3%) in Sweden. In the Netherlands, the radar values even seem to be overestimated by a factor 1/0.81 = 1.23 (18.7%). The fact that radar might overestimate rainfall rates compared with gauges may seem contradictory at first (given that actual values are lower) but can be explained by the fact that $\beta$ also accounts for the relative variability of the radar and gauge observations. Nevertheless, $\beta$ values should be interpreted very carefully as they rely on the assumption that the errors between radar and gauges are independent and log-normally

distributed with median 1. Figure 4 suggests that this might not always be the case. In particular, the bias between radar and gauges appears to increase during the peaks (see Section 3.3 for more details). In this case, the peak intensity biases for the top events in each country were 2.17 (Denmark), 2.09 (Finland), 1.98 (Netherlands) and 1.73 (Sweden), which is consistently larger than the average bias (as measured by the G/R ratio).

## 3.2 Overall agreement between radar and gauges

In the following, we consider the overall agreement between radar and gauges for each country. Figure 5 shows the rainfall intensities of radar versus gauges for each country (at the highest temporal resolution). Each dot in this figure represents a radar-gauge pair and all 50 events have been combined together into the same graph. Results show a good consistency between the two sensors (i.e., rank correlation coefficients between 0.77-0.91). However, the intensities measured by radar are clearly lower than that of the gauges. The G/R ratios are 1.59 for Denmark, 1.40 for the Netherlands, 1.56 for Finland and 1.66 for Sweden, corresponding to median relative differences of 37.3%, 28.4%, 35.9%, and 39.7% respectively. In addition to the bias, we also see a significant amount of scatter with relative root mean squares differences between 116.4% and 139.1% (depending on the country). This is characteristic for sub-hourly aggregation time scales and can be explained by the large spatial and temporal variability of rainfall and the fact that radar and gauges do not measure precipitation at the same height and over the same volumes.

Since it can be hard to compare gauge and radar measurements over short aggregation time scales, additional analyses were carried out to better understand how resolution affects the discrepancies between the two rainfall sensors. Figure 6 shows the scatter plot of radar versus gauge estimates when the data are aggregated to the event scale. Each dot in this graph represents the total rainfall accumulation (in mm) over an event. The aggregation to the event scale strongly reduces the scatter (i.e., RRMSD between 38.8% and 47.7%) and further increases the correlation coefficient (i.e., 0.80-0.92), making it easier to see the bias. The G/R ratio remains the same, as values only depend on total accumulation and not on the temporal resolution at which the events are sampled. The fact that radar and gauges agree more at the event scale than at the sub-hourly scale is encouraging. However, improvements are mainly attributed to the fact that many of the large discrepancies affecting the rainfall peaks get smoothed out during aggregation. This leads to an overly optimistic assessment of the agreement between radar and gauges that is not necessarily representative of what happens during the most intense parts of the events.

Based on the values of the G/R ratio in Figure 5, the Dutch C-band radar composite has the lowest apparent bias of all products (28.4%), followed by Finland (35.9%), Denmark (37.3%) and Sweden (39.7%). However, such direct comparisons are not really fair, as they do not take into account the different spatial and temporal resolutions of the radar products, the number of radars used during the estimation and their distances to the considered rain gauges. They also ignore the fact that the top 50 events in each country do not have the same intensities, durations and spatio-temporal structures. For example, the events in Denmark are significantly more intense compared with the Netherlands, Finland and Sweden, which might explain some of the differences. Also, the longest event in the Danish database only lasted 4 hours, which is shorter than for the other countries. To better understand the origin of the bias and interpret the differences between the countries, additional, more detailed analysis are necessary.

The first analysis we did was to estimate the model bias $\beta$ in Equation (5) under the assumption that the errors are log-normally distributed with median 1. Table 3 shows the estimated values of $\mu_g$, $\mu_r$, $\sigma_g$, $\sigma_r$ and $\sigma_\varepsilon$ at the highest available temporal resolution for each radar product (all 50 events combined). The obtained $\beta$ values are 1.04 for Denmark, 0.94 for the Netherlands, 1.11 for Finland and 1.11 for Sweden. This leads to a radically different assessment of the bias between radar and gauge values than with the G/R ratio. According to the $\beta$ values, the Danish product has the lowest model bias (3.8%), followed by the Netherlands (-6.4%), Finland (9.9%) and Sweden (9.9%). The Dutch radar product again appears to slightly overestimate the rainfall intensity, which is counter-intuitive given that the radar values are 30-40% lower than the gauges on average. However, this can be explained by the fact that $\beta$ is a theoretical bias that accounts for the relative variability of the rain gauge and radar observations around their respective means (see Equations 4-5). Products for which $CV_g$ is larger than $CV_r$ therefore see their bias reduced. This makes sense as gauge measurements are expected to have a larger coefficient of variation than radar due to their smaller sampling volume (i.e., point estimate versus areal average). Another reason is that gauges are known to suffer from relatively large sampling uncertainties at sub-hourly time scales. The fact that Denmark uses RIMCO tipping bucket gauges (as opposed to the float gauges in the Netherlands and weighing gauges in Finland and Sweden) therefore also makes a difference when calculating $\beta$. The bias adjustment factor $\exp(\frac{-\sigma_\varepsilon^2}{2})$ combines all these different factors together, which leads to a fairer comparison of the different radar products. The fact that the theoretical bias after accounting for differences in mean and variance might be as low as 10% (despite what the G/R ratio suggests) and that products with higher spatial/temporal resolutions seem to be affected by lower biases (in absolute value) is quite encouraging. However, one has to keep in mind that the representativity of $\beta$ strongly depends on the adequacy of the model proposed in Equation (1). Further analyses presented in the next section show that some of these assumptions might not be very realistic.

### 3.3 Conditional bias with intensity

The analyses performed in Sections 3.1 and 3.2 are useful to understand the overall agreement between radar and gauges over a large number of events but the estimated values strongly depend on the assumption that the bias $\beta$ in Equation (1) is constant. Our initial analysis in Section 3.1 already showed that in reality, the bias is likely to fluctuate over time, increasing in times of heavy rain. As mentioned in the introduction, time and intensity-dependent biases in radar or gauge estimates are highly problematic because they affect the timing and magnitude of peak flow predictions in hydrological models. Here, we perform a more quantitative assessment of this effect by studying the conditional bias between radar and gauges with respect to the rainfall intensity. Conditional biases are detected and quantified on the basis of the multiplicative bias model in Equations (1) and (2). If our assumptions are correct and there is no conditional bias, Equation (2) tells us that the average log-ratio between rain gauge and radar estimates should be a Gaussian random variable with constant mean and variance. Moreover, this result must hold independently of the rainfall intensity $R_g(t)$. To detect the presence of a conditional bias in the G/R ratio, we therefore plot the values of $\ln(\frac{R_g(t)}{R_r(t)})$ vs $R_g(t)$ (at the highest available temporal resolution) and calculate the slope of the corresponding regression line, as shown in Figure 7. If the slope is positive, the bias increases with intensity. The relative rate of increase (in percentage) of the G/R ratio per mm/h is then given by $100(e^m - 1)$, where $m$ is the slope of $\ln(\frac{R_g(t)}{R_r(t)})$ vs $R_g(t)$.

The fitted regression lines in Figure 7 show that three out of the four main radar products exhibit a clear positive conditional bias with intensity. The only product for which the bias does not increase with intensity is the Finnish OSAPOL. Incidentally, the Finnish OSAPOL is also the only product in which heavy rainfall rates are estimated through differential phase instead of reflectivity, pointing to the advantage of polarimetry over fixed Z-R relationships. The relative rates of increase for the G/R ratio are 1.09% per mmh$^{-1}$ in Denmark, 0.86% in the Netherlands, 0.09% in Finland and 2.12% in Sweden. This may not

seem large but can make a big difference when rainfall intensities vary from 1 mmh$^{-1}$ to more than 100 mmh$^{-1}$. For example, in Denmark, the G/R ratio (conditional on intensity) increases from 0.92 at 1 mmh$^{-1}$ to 2.69 at 100 mmh$^{-1}$. In Sweden, the conditional G/R ratio varies from 1.49 at 1 mmh$^{-1}$ to 11.96 at 100 mmh$^{-1}$. By contrast, the conditional G/R ratios at 100 mmh$^{-1}$ for the Netherlands and Finland only reach values of 2.48 and 2.40 respectively. The fact that both the Danish and Swedish products have large conditional biases also explains why their overall bias (as measured by the G/R ratio without

conditioning on intensity) is slightly larger than for the Netherlands and Finland. However, since large rainfall intensities are rare, the net effect of the conditional bias on the overall G/R ratio remains rather small.

      The most likely explanation for the conditional bias with intensity is the fact that 3 out of the 4 main radar products use a fixed Marshall-Palmer Z-R relationship to estimate rainfall rates from reflectivity. The bias therefore increases/decreases whenever the raindrop size distribution starts to deviate significantly from the Marshall-Palmer, as is usually the case during

strong convective precipitation and high rainfall intensities. The mean field bias-adjustments based on rain gauge data can help reduce the overall bias by tuning the prefactor in the Z-R relationship. However mean field bias adjustments are insufficient to account for the rapid changes in raindrop size distributions in heavy rain. Previous studies suggest that the best way to mitigate biases and ensure accurate hydrological predictions is to frequently adjust the radar data over time (Löwe et al., 2014). This might also explain why the Swedish and Danish radar products which are corrected using daily gauge data have a

stronger conditional bias with intensity than the Dutch product which uses hourly corrections. Another even better strategy, as demonstrated by the low conditional bias of the Finnish OSAPOL product, is to replace the Z-R relation by a R(Kdp) retrieval which is known to be less sensitive to variations in drop size distributions and calibration effects (Wang and Chandrasekar, 2010).

### 3.4   Other sources of bias

The conditional bias with intensity explains a lot of the differences between the radar products. However, this is only one part of the story and other confounding factors such as the distance between the radar(s) and the gauges also need to be considered. Figure 8 shows the log-ratio of gauge versus radar estimates $\ln(\frac{R_g(t)}{R_r(t)})$ as a function of the distance to the nearest radar. Compared with intensity, the trend with distance appears to be much weaker. Out of the 4 considered products, only the Danish C-band exhibits a trend that is significantly different from zero (at the 5% level). This makes sense given that the Danish product

only considers data from a single radar and only applies a mean field bias correction, making it more likely to be affected by range effects such as overshooting, non-uniform beam filling and attenuation. Based on our analyses, the multiplicative bias $\beta$ increases by 0.73% per km. However, since the range of distances between radar and gauges in Denmark is relatively small (from 29.2 to 74.2 km), bias values only vary from 1.06 to 1.47 at minimum and maximum distances respectively. Distance

therefore only plays a minor role in explaining the variations in bias compared with intensity. Interestingly, the composite products in the Netherlands and Finland do not seem to suffer from significant conditional biases with distance, highlighting the advantage of combining data from different radars and viewpoints to mitigate range effects. The Swedish product currently does not combine measurements from multiple radars in an optimal way, only using the measurements from the best (i.e., nearest) radar. However, the Swedish BRDC also contains an additional range-dependent bias correction (see Section 2.2.4) that appears to be rather efficient at removing large-scale trends with distance. However, the strong conditional bias with intensity in the Swedish BRDC also makes it harder to see potential range-dependent biases in the first place.

Another important aspect that needs to be considered when comparing the radar products is the difference in spatial and temporal resolutions. One way to study this would be to aggregate all radar products to a $2 \times 2$ km$^2$ and 30 min time scales before comparing them. However, this is not recommended as simple arithmetic averaging of processed radar fields does not really mimic what a lower resolution radar would see (e.g., due to the non-linear relation between rain rate and reflectivity and the multiple post-processing steps applied to the rainfall estimates). A better approach is to derive so-called areal-reduction factors (ARFs). Several ways to estimate ARFs have been proposed in the literature. ARFs can be estimated through the analysis of the spatial correlation structure (Rodríguez-Iturbe and Mejía, 1974; Ciach and Krajewski, 1999a) or more empirically as the ratio between maximum areal-averaged rainfall intensities between radar and gauges (Thorndahl et al., 2019). Here, the latter approach is used, specifically, Equation (8) in Thorndahl et al. (2019) with $b_1 = 0.31$, $b_2 = 0.38$ and $b_3 = 0.26$. Using the calculated ARFs, we estimated that the average bias between a point measurement and the Danish radar estimates (0.25 km$^2$, 5 min) should be in the order of 13%. For Finland and the Netherlands (1 km$^2$, 10 min), the average underestimation should be about 19% and 30% for Sweden (4 km$^2$, 15 min). Table 4 summarizes the G/R ratios before and after subtracting the areal-reduction factors above. The new multiplicative biases between radar and gauges after taking into account the ARFs are 1.39 in Denmark, 1.14 in the Netherlands, 1.27 in Finland and 1.17 in Sweden. This corresponds to median relative differences of 28%, 12.2%, 21.2% and 14.5% with respect to the gauges. The best products in terms of residual bias after applying the ARF would therefore be the Dutch, followed by the Swedish, Finnish and Danish. However, this is a rather simplistic way of accounting for the difference in scale that does not take into account the spatio-temporal structures and different characteristics of top 50 rain events in each country. Also, it is highly questionable whether it makes sense to apply areal-reduction factors to the radar data in the first place since most of the products (except the Finnish OSAPOL) have been bias-corrected using gauges. Part of the differences in measurement support bias should therefore already have been accounted for during the bias adjustments. Also, the fact that the ARFs used in this paper were derived from Danish radar data only and using a different collection of events might not be optimal. A more elaborate approach with variable ARFs for each country/event might provide a more realistic assessment of the support bias. Future studies with denser rain gauge networks could take a more detailed look at this. In particular, it would be interesting to know whether the conditional bias in Section 3.3 is mostly due to support bias (with higher rainfall intensities corresponding to higher ARFs) or to natural variations in raindrop size distributions (through the Z-R relation).

### 3.5 Agreement during the peaks

In this section, we take a closer look at how well the rainfall peaks are captured by the radar. Figure 9 shows the 10%, 25%,
50%, 75% and 90% quantiles of peak intensity bias between radar and gauges as a function of aggregation time scale. The
dashed horizontal lines denote the average apparent bias (i.e., the G/R ratio). We see that the Netherlands and Finland have
relatively low median peak intensity biases of 1.82 and 1.88 at 10 min resolution (approximately 1.2-1.3 times higher than the
average bias). Denmark and Sweden on the other hand have substantially higher median PIB values of 2.96 and 2.24, (1.86
respectively 1.35 times higher than the average). Moreover, the rate at which the PIB decreases with aggregation time scale is
different in each country. In Denmark and Sweden, the PIB remains well above the average bias for all aggregation time scales
up to 2 hours while in the Netherlands and Finland, the PIB converges much faster to the mean bias (i.e., after approx. 60 min
for the Netherlands and 20 min for Finland). This is no coincidence and can be explained by the fact that the Netherlands use
hourly rain gauge data to bias correct their radar estimates while the Danish and Swedish products use daily bias adjustment
factors. Thorndahl et al. (2014a) showed that switching from daily to hourly mean field bias adjustments can slightly improve
peak rainfall estimates but also pointed out that hourly bias corrections tend to be problematic in times of low rain rates due to
the small number of tips in the gauges. Therefore, in order to make a generally applicable adjustment that works for all rain
conditions, the authors argue that it is better to use daily adjustments. Here, we see that this strategy can result in a severe
increase of the peak intensity bias at sub-hourly scales, with some of the radar-gauge pairs differing by more than a factor 5.
The Dutch radar product also exhibits a rapid increase in PIB at sub-hourly scales. However, since the conditional bias with
intensity is rather small, the overall G/R ratio at 10 min resolution rarely exceeds more than a factor 3. The Finnish product is
interesting, as it is the only that has not been bias corrected with gauges. Its strength is that it makes use of polarimetry (i.e.,
Kdp) to estimate rainfall rates during the peaks. This results in almost identical performances in terms of PIBs than a traditional
approach based on Z-R relationship with hourly bias corrections, as used in the Netherlands. The only notable difference is the
rate at which the peak intensity bias converges to the average bias, with the Finnish product exhibiting a lower dependence on
the aggregation time scale than the Dutch product.

Another explanation for the high peak intensity biases in Denmark and Sweden could be that these two countries currently
do not take advantage of multiple overlapping radar measurements. By contrast, the Dutch and Finnish radar products are
"true composites" based on a weighted average of overlapping radar measurements (with weights depending on the distance
to the radar and the elevation angle). Clearly, the ability to combine measurements from multiple radars and viewpoints is an
advantage in times of heavy rain, as it reduces the spatial autocorrelation of radar-based errors due to environmental factors (i.e.,
such as range effects, vertical variability and attenuation). However, quantifying this more precisely would require additional
dedicated experiments (e.g., with/without compositing) that are beyond the scope of this study. Moreover, we have already
established that range-dependent biases only play a minor role in this study. The net effects of radar compositing on the
average G/R ratio and peak intensity bias within this study are therefore likely to be small and limited to a few events.

Another equally interesting result is the fact that the PIB for specific events does not necessarily decrease when the radar and
rain gauge data are aggregated to a coarser time scale. Figure 10 illustrates this point by showing the PIBs for the top event in

each country as a function of the aggregation time scale. The time series corresponding to these 4 events were already shown in Figure 4. While the PIB in the Netherlands and Finland exponentially decays with aggregation time scale, Denmark and Sweden exhibit a more complicated structure characterized by multiple ups and downs. Looking at event 1 for Denmark, we see that the peak intensity bias starts at 2.17 (53.9%) at 5 min, decreases to 2.1 (52.4%) at 10 min, increases again to 2.17 (53.9%)

at the 15 min time scale, decreases until 1.78 (43.9%) at 35 min only to increase again to 2.02 (50.4%) at 45-50 min. The multiple ups and downs can be explained by the intermittent nature of this event, with 4 successive rainfall peaks separated by approximately 15-45 min (see Figure 4). Each of these peaks is characterized by different random observational errors, causing extremes at certain scales to be captured better than others. The same applies to event 1 in Sweden, where the peak intensity bias starts at 1.73 (42.3%) at 15 min, decreases to 1.67 (40.1%) at 30 min and increases again to 1.75 (42.8%) at 45 min. In

this case, the event is less intermittent and there is only one single rainfall peak. However, Figure 4 clearly shows 3 consecutive time steps during which the radar underestimates the rainfall rate. These examples show that even though globally speaking, the average peak intensity bias between radar and gauges converges to the average G/R ratio when the data are aggregated to coarser time scales (as shown in Figure 9), this might not always be the case locally and does not necessarily apply to all events. The reason for this is that the PIB depends on a multitude of confounding factors (e.g., calibration errors, natural variations

in drop size distributions, range effects, wind, vertical variability, attenuation, etc...). When individual sources of error depend on each other or exhibit significant auto-correlation, their combined effect might cause the PIB to (locally) increase with aggregation time scale. In particular, strongly auto-correlated sources of bias such as changing drop size distributions, signal attenuation or wind effects can cause the PIB to increase with aggregation time scale.

The notion that peak intensity biases between radar and gauges can amplify when data are aggregated to coarser time scales

is not new in itself but has important consequences for the representation of peak rainfall intensities in hydrological models as it affects the choice of the optimal spatial and temporal resolution at which models should be run when making flood predictions. Another important finding of our study is that single-radar products with daily rain gauge adjustments are more likely to contain increasing PIBs with aggregation time scale than composite products with hourly bias corrections. This makes sense as mean field bias adjustments can (partly) compensate for the bias in rainfall rate due to deviations from the Marshall-Palmer

drop size distribution in the Z-R relationship. Similarly, radar compositing can mitigate the bias due to environmental factors such as range effects, vertical variability and attenuation. To show this, we computed, for each event, the time scale at which peak intensity bias reaches its maximum value. Figure 11 shows that in Denmark, 21/50 events exhibited a maximum PIB at a scale larger than that of the highest available temporal resolution. Similarly, for the Swedish radar product, 26/50 cases of locally increasing peak intensity biases with aggregation time scale could be identified. By contrast, the Finnish and Dutch

radar products, which make use of compositing and more frequent bias adjustments, only contained 14 and 8 such events, respectively. Further analysis reveals that most of the events with locally amplifying PIBs consist of two or more rainfall peaks separated by 10-30 min, with rapidly fluctuating rainfall intensities between them (i.e., high intermittency). Some events with single rainfall peaks during which radar strongly underestimated rainfall rates for two or more time steps in a row were also identified. However, due to the limited temporal autocorrelation in heavy rain, most peak intensity bias values reached their

maximum at time scales of 30 minutes or less.

## 3.6 Results for the additional radar products

Figures 12(a)-(d) summarize the results obtained for the X-band radar system in Denmark. Figure 12a) shows that there is a fairly good consistency between the radar and gauge estimates (rank correlation coefficient of 0.87). The average G/R ratio at 5 min is only 1.20 (16.7%), which is substantially lower than for the C-band products. The root mean square difference is 12.5 mmh$^{-1}$ (98.0%), which is high but lower than for the C-band products (116-139%). Part of the improvement could be due to the higher spatial resolution of the X-band radar. However, the statistics must be interpreted very carefully as only 10 events over 2 years were considered for the analyses (see Table A5 for more details). The good news is that peak rainfall intensities during these 10 events (70-95 mmh$^{-1}$) were rather high and in the same order of magnitude as for the top 50 events in the Netherlands, Finland and Sweden. The total rainfall amounts per event (10-30 mm) were lower though, and the events sampled by the X-band system were rather short and localized. The model bias $\beta$ in Equation (1) is 0.77, which suggests that after accounting for the relative variability of radar and rain gauge data, the X-band radar might actually overestimate the rainfall rates compared with the gauges. However, this is most likely a statistical artifact due to the assumption that the multiplicative error terms in Equation (1) are independent of intensity, which is unlikely to be true here. Indeed, it is important to keep in mind that multiplicative biases in the Danish X-band radar product were assessed on the basis of 5-min tipping bucket rain gauge. The latter are known to be affected by large sampling uncertainties and discretization effects, which could explain why the rain gauge data are significantly more variable (CV$_g$=1.61) compared with the radar measurements (CV$_r$=1.34). The large relative variability of the gauge data results in an overestimated noise term $\varepsilon(t)$ and consequently, an underestimated model bias $\beta$. In addition to the sampling issue, Figure 12b) also shows that there is a clear conditional bias with intensity (0.88% per mmh$^{-1}$) in the X-band data. The conditional bias with intensity affects the accuracy of the X-band radar in times of heavy rain, leading to high peak intensity biases. Figure 12d) shows that the median peak intensity bias at 5 min is 1.64 (39%) with 10% of the PIBs exceeding 3.1 (67.7%). One reason for this could be attenuation, which is known to play a major role at X-band. However, all reflectivity measurements have been corrected for attenuation prior to rainfall estimation. Also, Figure 12c) shows that there is no obvious change in the G/R ratio with the distance to the radar, as would be expected for attenuated signals. This leads us to conclude that similarly to the Danish and Swedish C-band products, the conditional bias with intensity is likely caused by the use of a fixed Z-R relation (together with daily bias adjustments). It also means that higher resolution alone is probably not enough to avoid strong conditional biases with intensity. The latter must be mitigated by other means, for example by replacing the fixed Z-R relationship with a R(Kdp) estimate in times of heavy rain or by performing more frequent bias adjustments with the help of gauges. Unfortunately, the current software of the Danish X-band radar does not offer the possibility to estimate R from Kdp yet. The improvements due to switching from Z to Kdp could therefore not be assessed within the context of this study. Similarly, KNMI and DMI are currently working on better exploiting the new polarimetric capabilities of their C-band radars to better account for natural variations in the raindrop size distributions. However, these upgrades still require more research and could not be assessed formally here.

Figure 13 compares the agreement between the 4 C-band radar products in Denmark, Finland and Sweden and the BAL-TRAD composite for the top 50 events in each country. The Netherlands are not included in this graph because they are not

covered by the BALTRAD. To avoid sampling issues, all values are compared at the common aggregation time scale of 15 min, which might introduce some additional sampling uncertainty. The spatial resolutions, however, remain unchanged. Overall, the BALTRAD seems to perform rather similarly to the national products. It has slightly lower rank correlation coefficients and higher root mean square differences. The bias (as measured by the G/R ratio) is also very similar, except in Sweden where the BALTRAD appears to underestimate more with respect to the gauges (1.77 versus 1.66). This makes sense given that the

BALTRAD does not include the HIPRAD adjustments which results in higher overall bias and conditional bias with intensity. Interestingly, the BALTRAD performs worse than the Danish C-band product in terms of overall bias but better in terms of median peak intensity bias. There are many possible explanations for these differences. One reason could be the difference in spatial resolution (2 km for BALTRAD versus 500 m for the Danish C-band). Another reason could be the differences in the bias adjustment schemes, more specifically the fact that BALTRAD uses monthly gauge data to correct for bias while the Dan-

ish C-band product is adjusted on a daily basis. However, this does not explain why the median peak intensity bias is lower in the BALTRAD. While this remains rather speculative, we think that the main reason BALTRAD agrees better with the gauges in times of heavy rain is because it includes data from multiple radars in the greater Copenhagen region. This offers more flexibility compared with a single-radar setup and makes sure that the closest possible radar gets selected with respect to the position and characteristics of the storm. However, this does not seem to result in systematic improvements across all events.

Indeed, it is worth pointing out that while the median PIB value is lower in BALTRAD, the average PIB value is slightly larger in BALTRAD (3.0) than for the Danish C-band product (2.63). The same applies to all the other countries as well (2.49 versus 2.05 for Finland and 3.27 versus 2.60 for Sweden). In other words, there are some events in the database for which BALTRAD has significantly larger PIB values than others. These are the events responsible for the strong conditional bias with intensity. For these events, the bias is most likely due large deviations from the theoretical Marshall-Palmer Z-R relationship, which can

not be mitigated with the help of compositing alone.

## 4   Conclusions

The accuracy of 6 different radar products in 4 countries (Denmark, Finland, the Netherlands and Sweden) has been analyzed. Special emphasis has been put on quantifying discrepancies between radar and gauges in times of heavy rain. A relatively good agreement was found in terms of temporal consistency (correlation coefficient between 0.7-0.9). However, the scatter

at sub-hourly time scales remains high (98-144% at 5-15 min). Moreover, all 6 radar products exhibited a clear pattern of underestimation. The multiplicative biases at 5-15 min were between 1.20-1.77, suggesting that radar underestimates rainfall rates by 17-44% compared with gauges. A substantial part of the bias (i.e., 10-30% according to areal-reduction factors) is likely due to differences in sampling volumes. However, this remains hard to quantify precisely in the absence of dense rain gauge networks. An alternative bias model that accounts for the differences in mean and variance between radar and gauge

measurements suggested that the actual bias affecting radar rainfall estimates could be as low as 10%. Moreover, higher resolution radar products seemed to agree better with gauges, which is encouraging. At the same time, these conclusions

strongly rely on the assumption that errors are log-normally distributed and independent of intensity, which, as we have seen in this study, is likely not to be true during the peaks.

Based on our analysis, the main issue affecting current operational radar rainfall estimates is the fact that the multiplicative bias increases with rainfall intensity. The most likely reason for this conditional bias is the use of a fixed Marshall-Palmer Z-R relationship to convert reflectivity to rainfall rates, which does not account for the changes in raindrop size distributions during heavy convective precipitation events. One way to mitigate the conditional bias with intensity, as demonstrated by the Finnish OSAPOL project, is to rely on differential phase shift Kdp instead of reflectivity. Another possibility is to use a fixed Z-R relationship but to perform frequent bias adjustments with the help of rain gauges (as demonstrated by the Dutch C-band product). Here, the temporal resolution of the gauge data appears to play crucial role in controlling the magnitude of the conditional bias, with daily and monthly corrections resulting in an increase of the bias of approximately 2% per $mmh^{-1}$ and hourly adjustments resulting in an increase of about 1% per $mmh^{-1}$. Nevertheless, even the hourly adjustments appeared to be insufficient for radar to adequately capture the peaks. Regardless of how rainfall rates were estimated, median peak intensity biases systematically exceeded the average G/R ratios, reaching values of 1.8-3.0 (i.e., radar underestimates by 44-67%). Occasionally, the peak intensity bias even exceeded 80% (factor of 5). We believe that sub-hourly bias adjustments might help further reduce the bias affecting the peaks. However, this only applies to the peaks and is not recommended for low to moderate rainfall intensities due to the large uncertainty affecting rain gauge measurements. Future research should focus on finding better ways to dynamically adjust radar data with the help of rain gauge measurements at different temporal resolutions depending on event dynamics, amounts and intensities.

Overall, the X-band data for Denmark showed promising results, outperforming all other C-band products in terms of accuracy and correlation, thereby demonstrating the value of high-resolution rainfall observations for urban hydrology. However, due to the shorter data record, only 10 events over 2 years could be considered. The polarimetric estimates from the Finnish OSAPOL project also showed promising performance, which is remarkable considering the fact that they were not adjusted by any gauges. However, it should also be pointed out that for now, the overall performance of the OSAPOL remains similar to that of the Dutch C-band product with fixed Z-R relationship and hourly bias correction. Interestingly, the distance between the radar and the gauges did not appear to have a strong effect on peak intensity bias. We explain this by the fact that range-dependent biases tend to be small compared with the large spatial variability of rain at the event scale. Therefore, range effects are masked by other errors and only become visible when the radar data are aggregated over the course of several days or months.

Another important finding of this paper was that the largest bias between radar and gauges in terms of peak intensities does not necessarily occur at the highest temporal sampling resolution. Depending on the autocorrelation structure of the errors and the resolution of the rain gauge data used for the adjustments, multiplicative biases may amplify over time instead of converging to the mean value. This mostly happens at the sub-hourly time scales and roughly affects 40-50% of all events in single-radar products and 15-30% in composite products. Most of these cases were characterized by a succession of multiple rainfall peaks or alternatively, one very intense peak of 15-30 min during which radar strongly underestimated the intensity for 2 or more consecutive time steps. The strong dependence of the error structure in radar data depending on aggregation time

scale still represents a major challenge as it limits our ability to accurately characterize rainfall extremes and uncertainties in hydrological models across scales (Bruni et al., 2015). One way to partially mitigate this effect is to combine measurements from multiple radars. However, more research is necessary to precisely quantify this part of the error.

Finally, like with any statistical analysis, there are a few important limitations that need to be mentioned. The first is that little focus has been given to the analysis of the rain gauge data themselves. In reality, gauges also suffer from measurement uncertainties and errors, the most common being an underestimation of rainfall rates in times of heavy precipitation due to calibration issues and wind effects. No attempt has been made to correct for these additional biases nor to distinguish between gauge and radar-induced errors. Since the gauge data are likely to be underestimated as well, the actual bias between the two

sensors might be larger than suspected. The second issue is the relatively short length of the observational record (10-15 years) which meant that only a small number of extreme rain events could be considered. Moreover, it is worth mentioning that some of the events in the database actually occurred on the same day but were captured by different gauges at different locations. The derived statistics might therefore be biased towards characterizing the performance of the radar during these days instead of the average performance over a large number of independent events. Another issue is the lack of a common denominator

for comparing the radar products. Future studies involving identical radar systems and different levels of processing (e.g., by switching on/off individual correction schemes) would be useful to get a better understanding of the strengths and weaknesses of individual retrieval techniques within a more controlled setting. Despite all these limitations, the present study already provided some important insight into the major issues affecting radar-rainfall estimates in times of heavy rain. Also, several useful strategies for mitigating errors and reducing biases were identified. Future research should focus on analyzing more

radar products and identifying the most promising strategies for improving performance in each country.

*Acknowledgements.* The authors acknowledge funding by the EU within the framework of the ERA-NET Cofund WaterWorks2014 project MUFFIN (Multiscale Flood Forecasting: From Local Tailored Systems to a Pan-European Service). This ERA-NET is an integral part of the 2015 Joint Activities developed by the Water Challenges for a Changing World Joint Programme Initiative (Water JPI). The first author acknowledges funding by the Netherlands Organisation for Scientific Research NWO (project code ALWWW.2014.3). The Finnish partners

acknowledge funding by the Maa- ja vesitekniikan tuki ry. foundation. The Optimal Rain Products with Dual-Pol Doppler Weather Radar (OSAPOL) project was funded by the European Regional Development Fund and Business Finland. The authors would like to thank the Danish, Finnish, Swedish and Dutch Meteorological Institutes (i.e., DMI, FMI, SMHI and KNMI) for collecting and distributing the radar and gauge data used in this study.

*Data availability.* The Dutch radar products are available for free in HDF5 format through the FTP of KNMI or in netCDF4 format via the

710 Climate4Impact website. The Danish, Swedish and Finnish products are not open yet but can be made available for research purposes upon request to the authors.

*Author contributions.* Marc Schleiss coordinated the experiments, developed the theoretical formalism, performed the analyses and wrote the manuscript. Jonas Olsson and Peter Berg compiled the Swedish radar and BALTRAD datasets with support from Denica Bozhinova. Tero Niemi and Teemu Kokkonen produced the Finnish radar and gauge datasets with support from Seppo Pulkkinen. Søren Thorndahl, Rasmus Nielsen and Jesper Ellerbæk Nielsen produced the Danish C-band and X-band radar datasets. All authors provided critical feedback and helped shape the research, analysis and manuscript.

*Competing interests.* The authors declare that they have no competing interests.

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

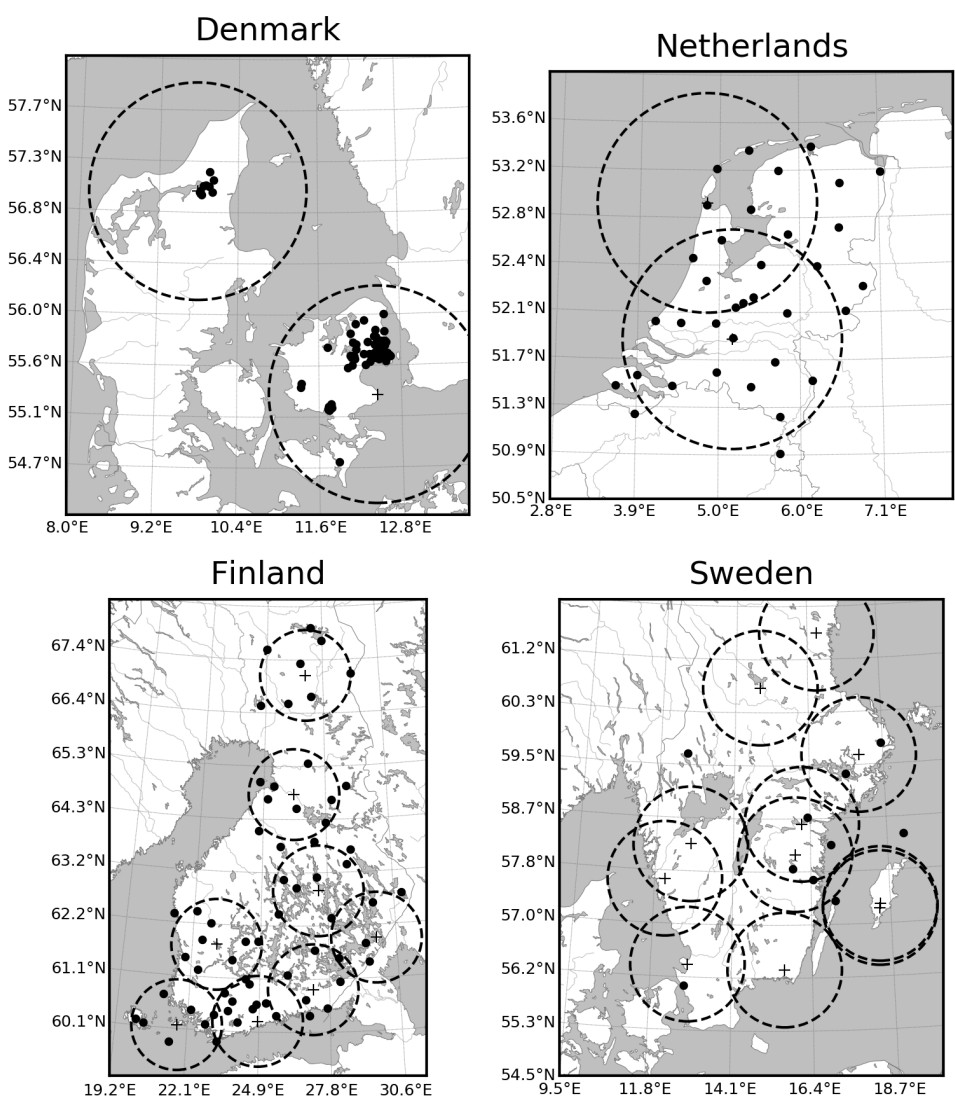

**Figure 1.** The four considered study areas in Denmark, the Netherlands, Finland and Sweden with the used rain gauges (black dots) and the location of the C-band radars marked by black crosses. The dashed lines denote circles of 100 km radius around each radar. Due to maintenance and relocations, not all the radars were operating at the same time.

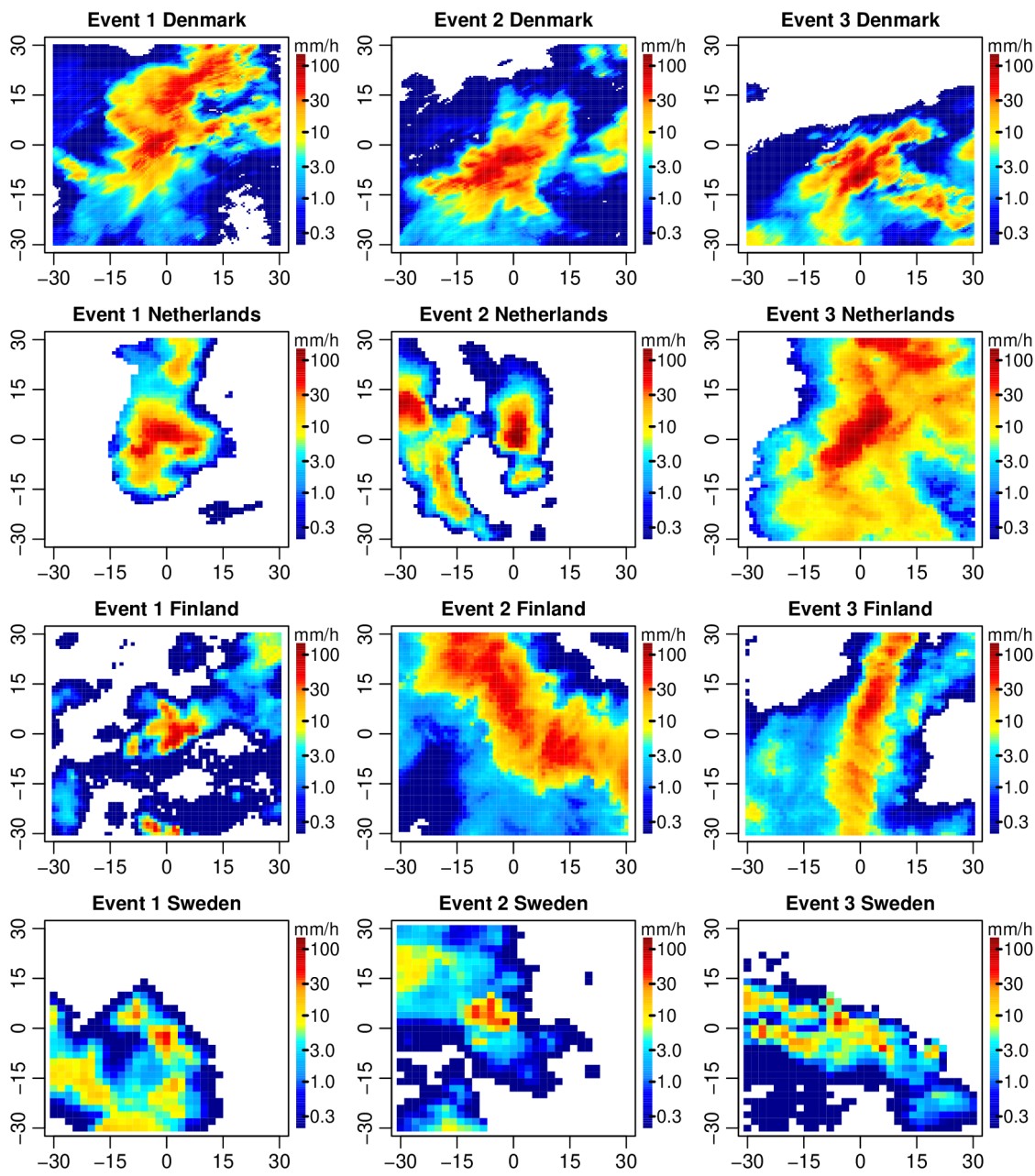

**Figure 2.** Snapshots of the radar rainfall estimates (in mmh$^{-1}$) at the time of peak intensity for the 3 most intense events in each country. Each map is a square of size 60×60 km$^2$ with the gauge located in the center of the domain.

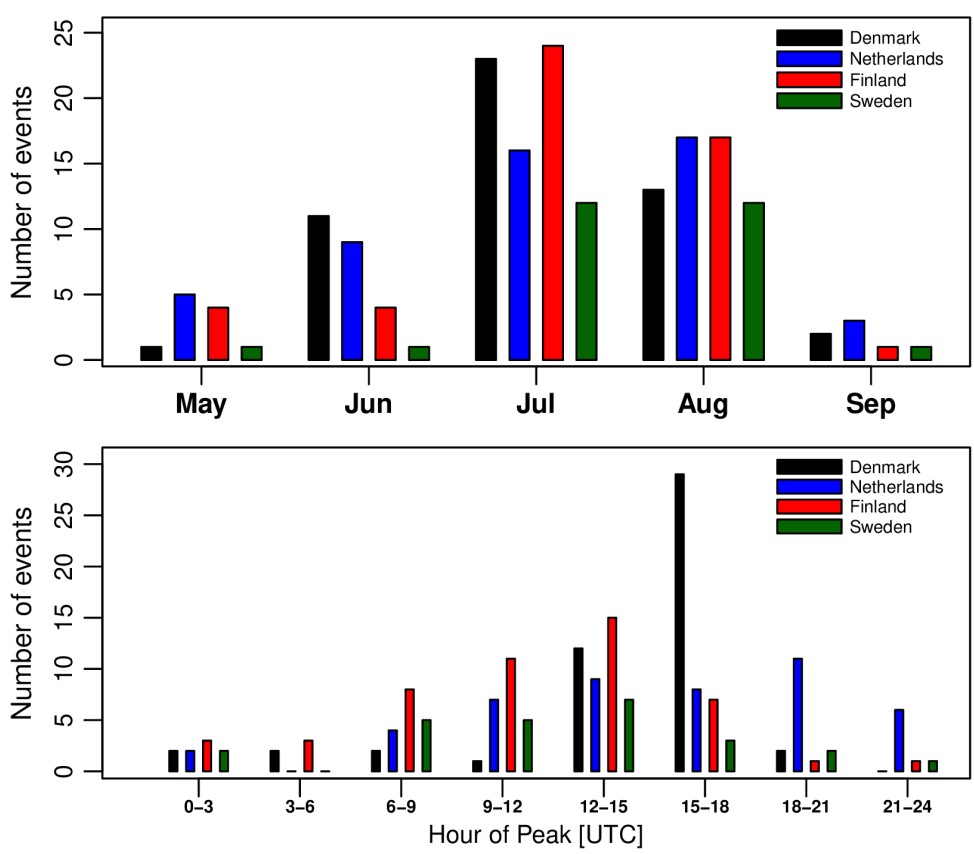

**Figure 3.** Distribution of the 50 top events over the month (top panel) and hour of the day (bottom panel).

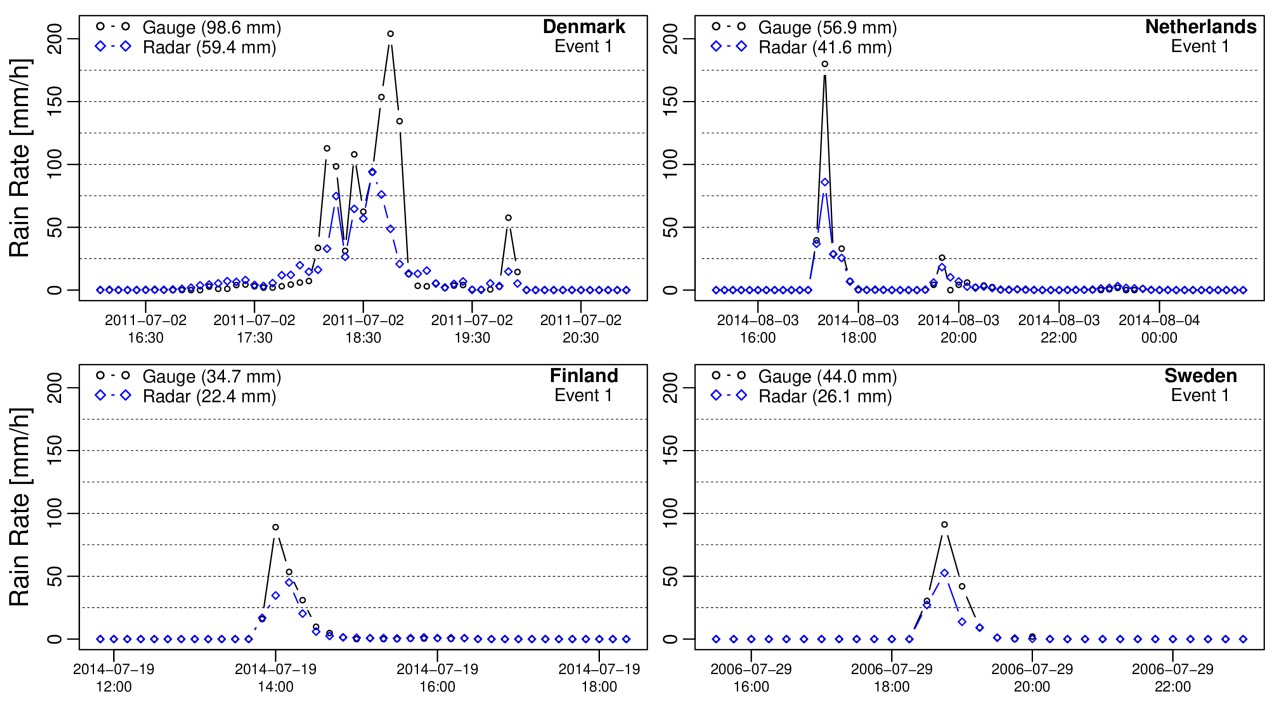

**Figure 4.** Time series of radar and gauge intensities (in mmh$^{-1}$) for the most intense event in each country.

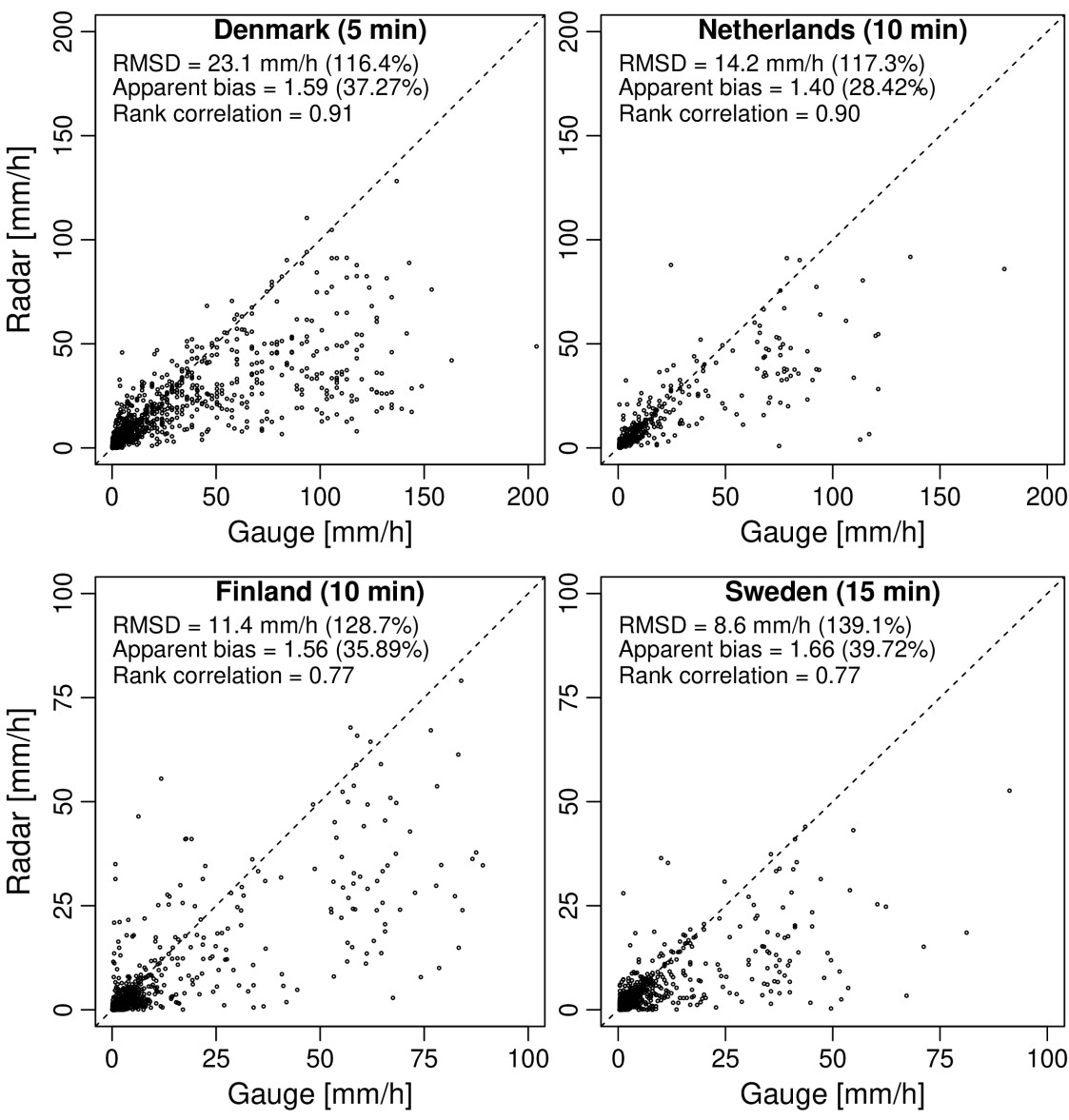

**Figure 5.** Radar versus gauge intensities (in mmh$^{-1}$) at the highest available temporal resolution for each country (all 50 events combined). The dashed line represents the diagonal.

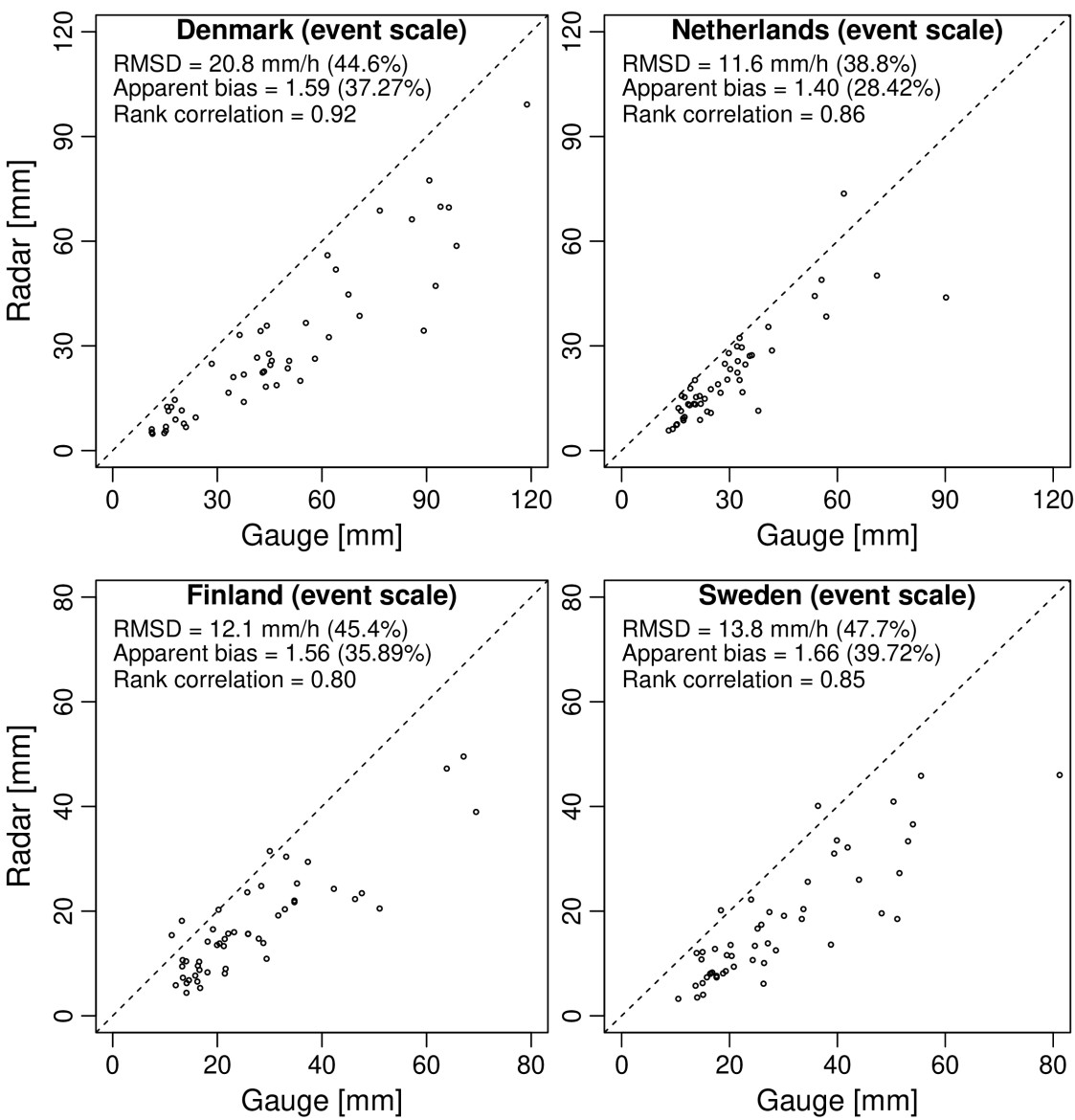

**Figure 6.** Radar versus gauge accumulations (in mm) at the event scale for each country (i.e., one dot per event). The dashed line represents the diagonal.

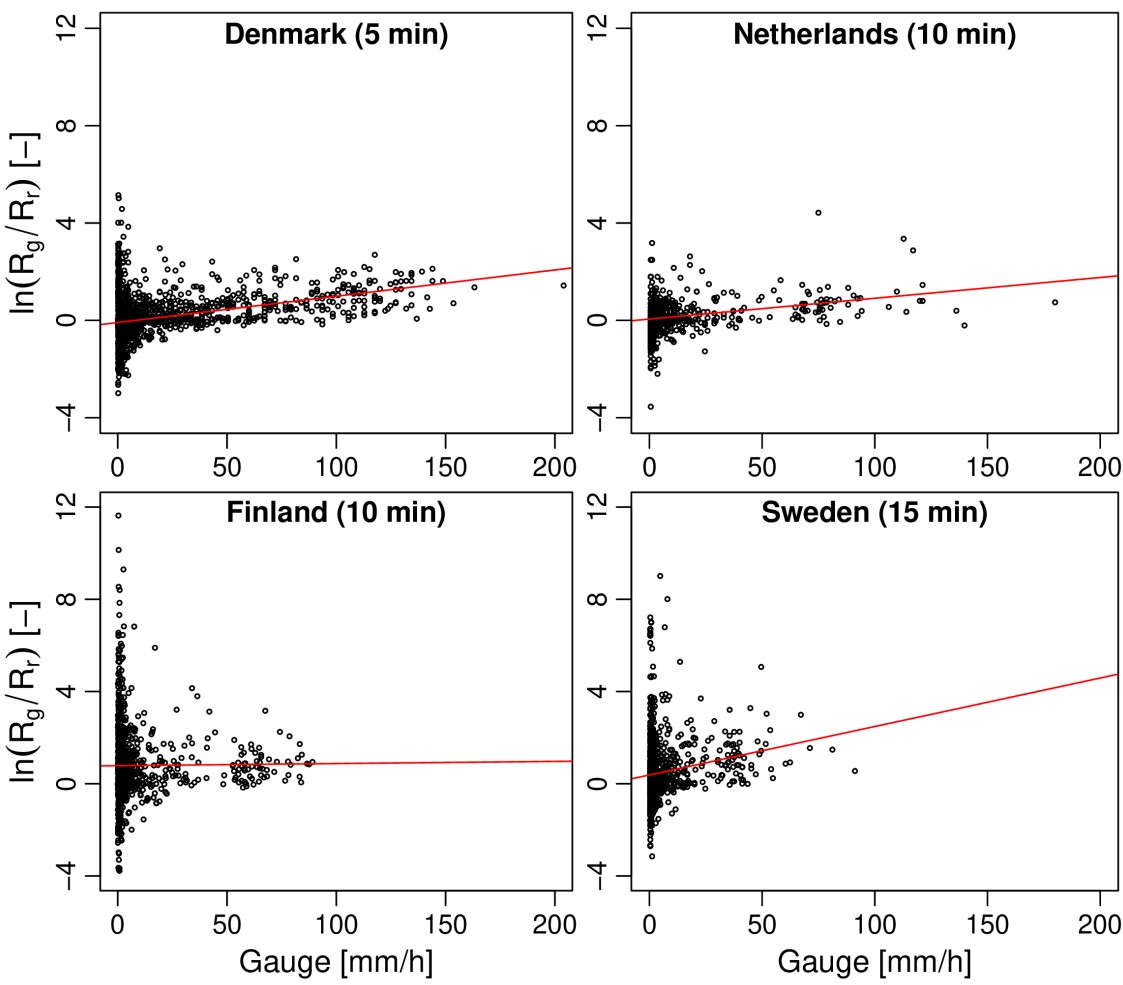

**Figure 7.** Log ratio of gauge over radar values as a function of rain gauge intensity (in mmh$^{-1}$) for each country. The red lines represent the fitted linear regression models.

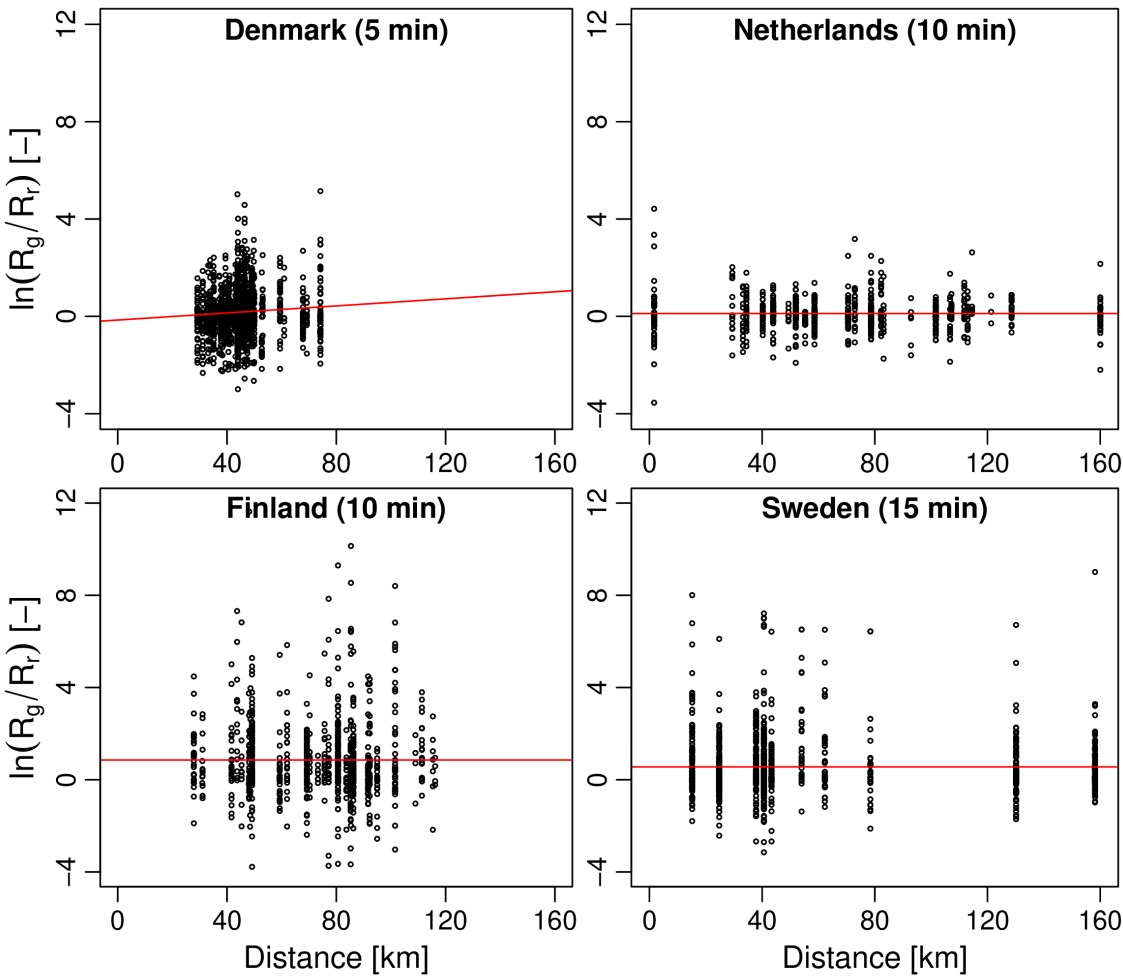

**Figure 8.** Log ratio of gauge over radar values as a function of the distance to the nearest radar. The red line represents the fitted linear regression model.

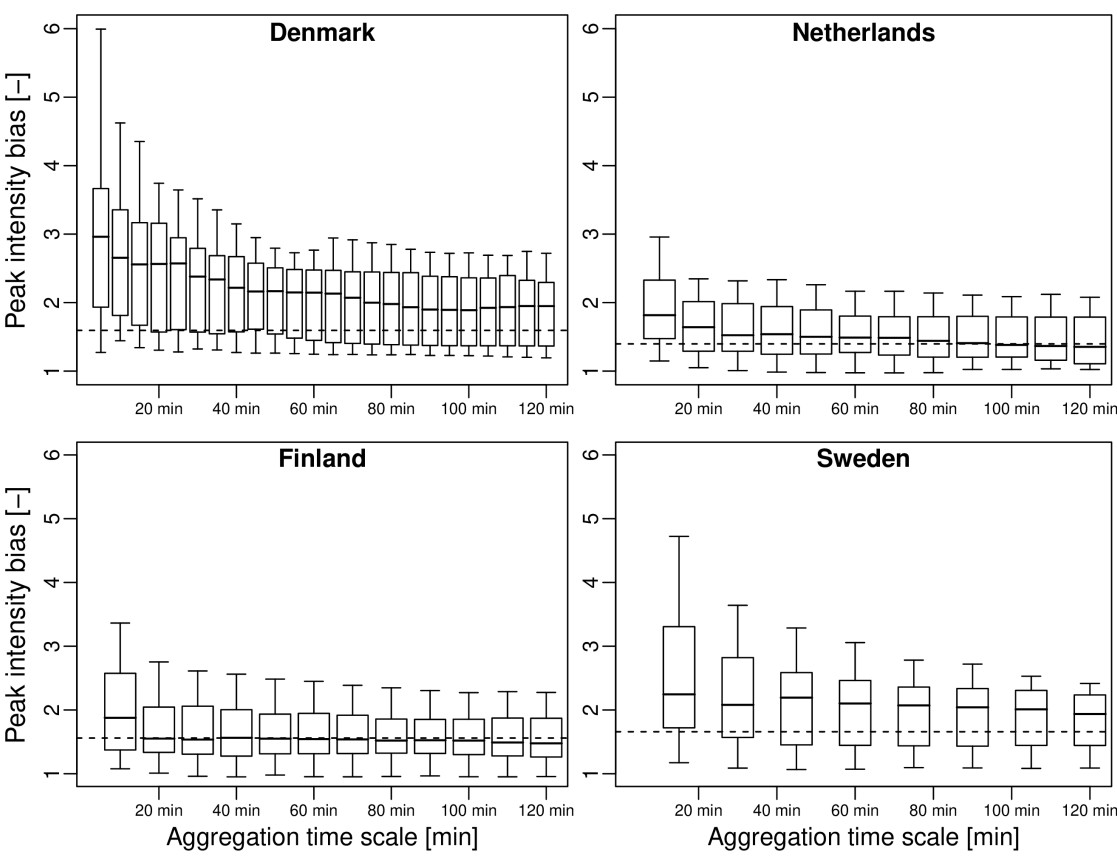

**Figure 9.** Boxplots of peak intensity bias versus aggregation time scale. Each boxplot represents the 10%, 25%, 50%, 75% and 90% quantiles for the 50 top events in each country. The horizontal lines denote the average multiplicative biases (G/R ratio).

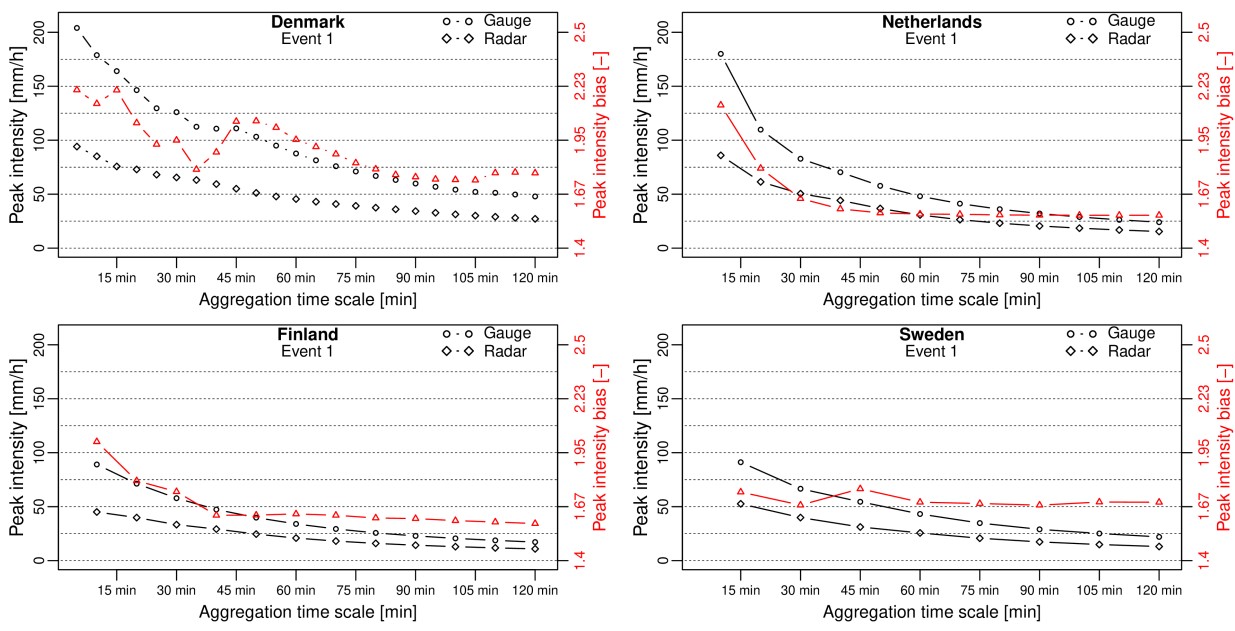

**Figure 10.** Peak rainfall intensities measured by radar and gauges as a function of the aggregation time scale for the top 1 event in each country. The red triangles show the peak intensity bias between radar and gauges (axis on the right).

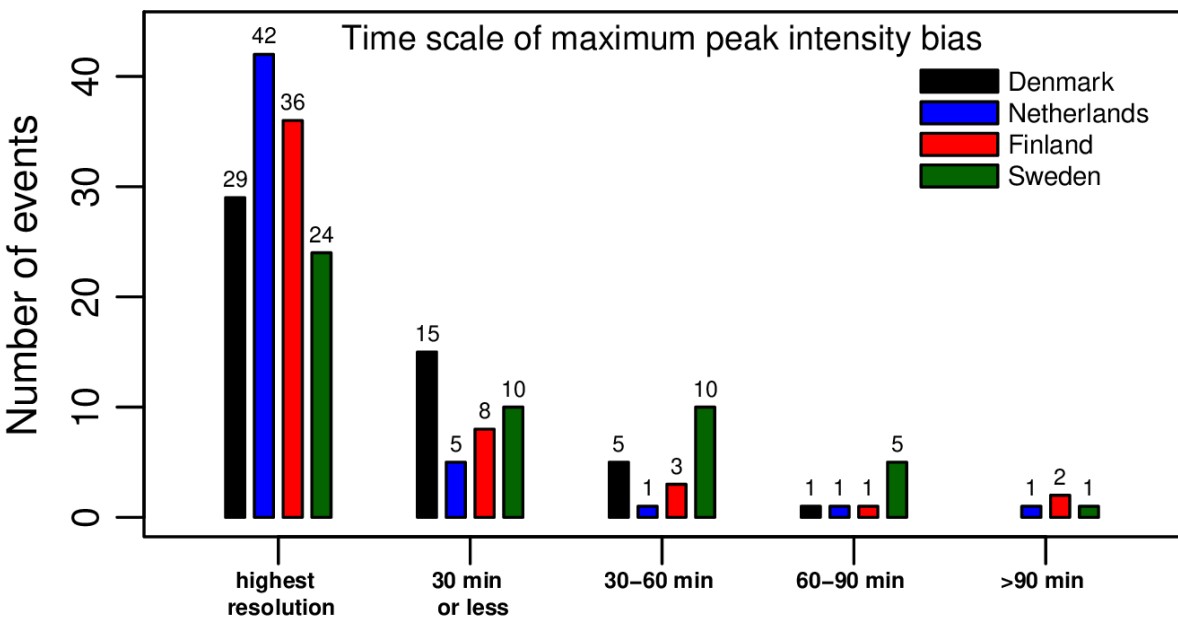

**Figure 11.** Aggregation time scale at which the maximum peak intensity bias between gauge and radar occurred.

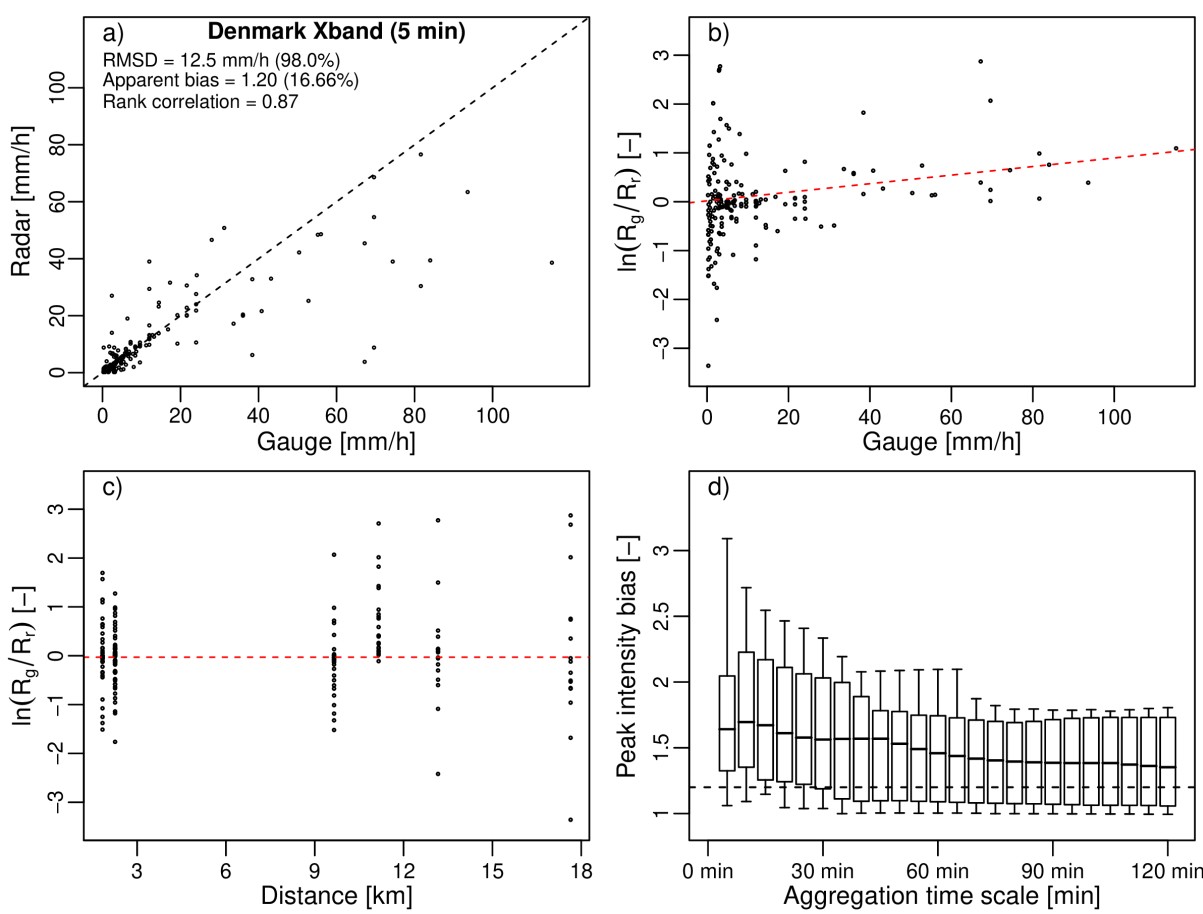

**Figure 12.** Performance metrics for the Danish X-band radar system (top 10 events).

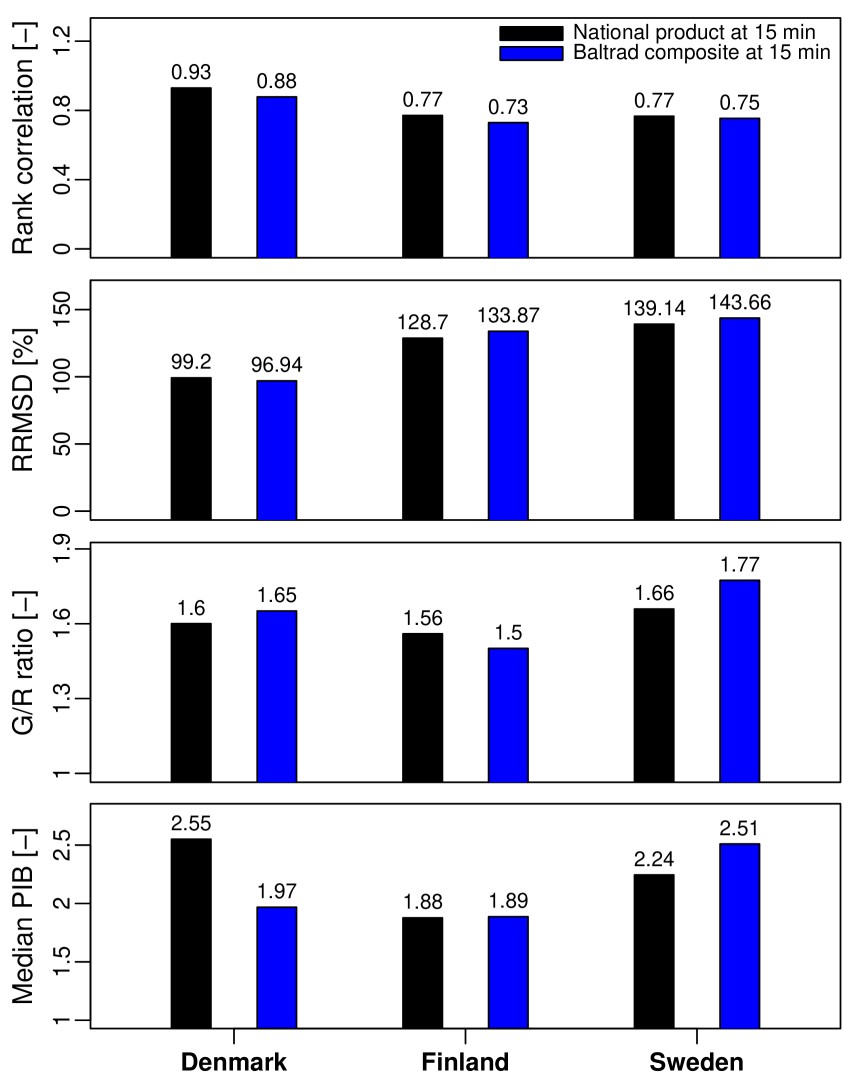

**Figure 13.** Rank correlation, relative root mean square difference, G/R ratio and peak intensity bias (at 15 min resolution) of the national radar products and the BALTRAD composite.

**Table 1.** Rain gauge datasets used to determine the top 50 rainfall events for each country. The time periods were chosen based on radar data availability.

|  | Denmark | Netherlands | Finland | Sweden |
|---|---|---|---|---|
| Number of available gauges | 66 | 35 | 64 | 10 |
| Gauges used for top 50 events | 50 | 31 | 50 | 5 |
| Time period | 2003–2016 | 2008-2018 | 2013-2016 | 2000–2018 |
| Gauge sampling resolution | 5 min | 10 min | 10 min | 15 min |

**Table 2.** Radar products used in this study.

| Country | Radar type(s) | Resolution | Method | Bias correction |
|---------|---------------|------------|--------|-----------------|
| Denmark | 1 single-pol C-band | 500×500 m, 5 min | Z-R | yes |
| Netherlands | 2 single-pol C-band | 1×1 km, 5 min | Z-R | yes |
| Finland | 9 dual-pol C-band | 1×1 km, 5 min | Z-R and Kdp | no |
| Sweden | 12 single-pol C-band | 2×2 km, 15 min | Z-R | yes |
| Denmark | 1 dual-pol X-band | 100×100 m, 1 min | Z-R | yes |
| Baltic region | C-band (BALTRAD) | 2×2 km, 15 min | Z-R | yes |

**Table 3.** Summary statistics for the highest aggregation time scale (all 50 events combined). Average intensity for gauges and radar $\mu_g$ and $\mu_r$, standard deviations $\sigma_g$ and $\sigma_r$, G/R ratio, coefficient of variation, scale parameter $\sigma_\varepsilon$ and model bias $\beta$.

| Country | $\mu_g$ | $\mu_r$ | $\sigma_g$ | $\sigma_r$ | G/R | $\frac{\mathrm{CV}_g}{\mathrm{CV}_r}$ | $\sigma_\varepsilon$ | $\beta$ |
|---|---|---|---|---|---|---|---|---|
| | mmh$^{-1}$ | mmh$^{-1}$ | mm$h^{-1}$ | mm$h^{-1}$ | [-] | [-] | [-] | [-] |
| Denmark (500 m, 5 min) | 19.8 | 12.4 | 32.7 | 17.6 | 1.59 | 1.17 | 0.93 | 1.04 |
| Netherlands (1 km, 10 min) | 12.1 | 8.6 | 23.7 | 15.5 | 1.40 | 1.09 | 0.89 | 0.94 |
| Finland (1 km, 10 min) | 8.8 | 5.7 | 17.2 | 11.1 | 1.56 | 1.00 | 0.83 | 1.11 |
| Sweden (2 km, 15 min) | 6.2 | 3.7 | 11.4 | 6.2 | 1.66 | 1.11 | 0.90 | 1.11 |

**Table 4.** Summary statistics for the highest aggregation time scale (all 50 events combined). G/R ratio, G/R ratio corrected for areal reduction factor ARF, model bias $\beta$ assuming log-normal distribution and relative increase in $\beta$ with respect to intensity and range.

| Country | G/R | G/R corrected | model bias | relative increase in $\beta$ | relative increase in $\beta$ |
|---|---|---|---|---|---|
| | [-] | for ARF [-] | $\beta$ [-] | with intensity [(mm/h)$^{-1}$] | with range [km$^{-1}$] |
| Denmark (500 m, 5 min) | 1.59 | 1.39 | 1.04 | 1.09% | 0.73% |
| Netherlands (1 km, 10 min) | 1.40 | 1.14 | 0.94 | 0.86% | 0 |
| Finland (1 km, 10 min) | 1.56 | 1.27 | 1.11 | 0.09% | 0 |
| Sweden (2 km, 15 min) | 1.66 | 1.17 | 1.11 | 2.12% | 0 |

# 1 Appendix: Top 50 events for each country

**Table A1.** Top 50 events for Denmark

| Event | Starting Time [UTC] | Gauge | Duration | Amount [mm] | Peak [mmh$^{-1}$] |
|---|---|---|---|---|---|
| 1 | 2011-07-02 17:05 | 5805 | 2h50min | 98.6 | 204.0 |
| 2 | 2011-07-02 17:20 | 5725 | 2h10min | 92.6 | 163.2 |
| 3 | 2011-07-02 17:10 | 5685 | 2h25min | 89.2 | 148.8 |
| 4 | 2013-08-10 17:25 | 5675 | 30min | 15.2 | 144.0 |
| 5 | 2006-08-15 05:55 | 5901 | 11h45min | 20.4 | 144.0 |
| 6 | 2011-07-02 17:10 | 5730 | 2h25min | 94.0 | 142.8 |
| 7 | 2011-07-02 16:55 | 5740 | 2h50min | 118.8 | 141.6 |
| 8 | 2016-07-25 16:30 | 5590 | 35min | 23.8 | 139.2 |
| 9 | 2011-07-02 17:00 | 5785 | 2h50min | 96.4 | 136.8 |
| 10 | 2011-07-02 17:15 | 5675 | 2h15min | 37.6 | 134.4 |
| 11 | 2007-08-11 13:05 | 5790 | 2h35min | 67.6 | 134.4 |
| 12 | 2007-08-11 14:50 | 5650 | 1h35min | 58.0 | 134.4 |
| 13 | 2007-08-11 13:50 | 5705 | 2h25min | 42.4 | 134.4 |
| 14 | 2011-07-02 17:10 | 5790 | 2h55min | 90.8 | 132.0 |
| 15 | 2011-07-02 15:45 | 5745 | 3h30min | 76.6 | 129.6 |
| 16 | 2005-08-07 09:15 | 5755 | 8h35min | 53.8 | 129.6 |
| 17 | 2011-07-02 18:15 | 5665 | 2h5min | 44.0 | 127.2 |
| 18 | 2016-06-23 18:45 | 5675 | 9h25min | 47.0 | 127.2 |
| 19 | 2007-08-11 13:45 | 5771 | 2h5min | 37.6 | 127.2 |
| 20 | 2011-07-02 17:05 | 5810 | 3h | 55.4 | 127.2 |
| 21 | 2007-06-23 09:15 | 5655 | 6h5min | 38.8 | 122.4 |
| 22 | 2007-06-23 09:30 | 5670 | 6h | 30.2 | 122.4 |
| 23 | 2011-07-02 17:20 | 5715 | 2h20min | 70.8 | 120.0 |
| 24 | 2011-07-02 17:25 | 5710 | 2h20min | 64.0 | 120.0 |
| 25 | 2011-07-02 17:20 | 5795 | 2h20min | 61.6 | 120.0 |
| 26 | 2011-08-08 13:05 | 5585 | 3h10min | 18.0 | 117.6 |
| 27 | 2011-07-02 17:20 | 5804 | 2h35min | 85.8 | 117.6 |
| 28 | 2013-08-10 10:20 | 5670 | 7h30min | 16.8 | 117.6 |
| 29 | 2016-06-23 18:30 | 5915 | 9h30min | 45.6 | 115.2 |
| 30 | 2008-06-27 09:25 | 5620 | 9h10min | 21.0 | 112.8 |
| 31 | 2011-07-02 17:25 | 5655 | 2h10min | 43.4 | 112.8 |
| 32 | 2007-08-11 13:50 | 5710 | 1h10min | 34.6 | 112.8 |
| 33 | 2005-07-30 08:10 | 5570 | 5h10min | 28.4 | 110.4 |
| 34 | 2013-08-10 17:20 | 5690 | 10min | 11.2 | 108.0 |
| 35 | 2009-07-20 09:20 | 5570 | 8h30min | 15.4 | 108.0 |
| 36 | 2015-09-04 06:40 | 5685 | 1h25min | 36.4 | 108.0 |
| 37 | 2011-07-02 17:20 | 5694 | 2h15min | 62.0 | 108.0 |
| 38 | 2016-06-23 18:30 | 5905 | 7h20min | 44.8 | 108.0 |
| 39 | 2011-08-09 19:00 | 5675 | 20min | 11.4 | 105.6 |
| 40 | 2015-09-04 06:05 | 5690 | 2h | 44.2 | 105.6 |
| 41 | 2011-07-02 17:20 | 5660 | 2h15min | 50.2 | 105.6 |
| 42 | 2016-06-23 18:20 | 5925 | 9h40min | 50.6 | 103.6 |
| 43 | 2011-05-22 14:50 | 5740 | 2h50min | 19.8 | 103.2 |
| 44 | 2007-08-10 18:20 | 5855 | 10min | 14.8 | 103.2 |
| 45 | 2016-06-23 18:30 | 5930 | 9h40min | 43.0 | 103.2 |
| 46 | 2008-06-27 09:20 | 5633 | 1h10min | 11.2 | 100.8 |
| 47 | 2016-06-23 18:30 | 5901 | 7h20min | 41.4 | 100.8 |
| 48 | 2011-07-02 18:20 | 5650 | 1h15min | 45.2 | 98.4 |
| 49 | 2011-07-02 18:55 | 5825 | 1h5min | 33.2 | 98.4 |
| 50 | 2014-06-20 03:50 | 5580 | 5h10min | 15.6 | 96.8 |

**Table A2.** Top 50 events for the Netherlands

| Event | Starting Time [UTC] | Gauge | Duration | Amount [mm] | Peak [mmh$^{-1}$] |
|---|---|---|---|---|---|
| 1 | 2014-08-03 17:10 | 380 | 6h30min | 56.9 | 180.0 |
| 2 | 2014-07-28 11:30 | 275 | 3h | 61.8 | 139.8 |
| 3 | 2011-06-28 18:20 | 356 | 6h | 90.2 | 136.2 |
| 4 | 2016-06-23 01:10 | 260 | 1h | 36.2 | 121.2 |
| 5 | 2015-08-30 22:20 | 283 | 3h50min | 30.2 | 120.0 |
| 6 | 2013-08-19 11:20 | 286 | 2h10min | 29.8 | 114.0 |
| 7 | 2015-08-30 19:40 | 356 | 6h20min | 55.6 | 112.8 |
| 8 | 2012-05-20 14:20 | 375 | 4h30min | 21.8 | 109.8 |
| 9 | 2013-07-26 12:50 | 286 | 30min | 22.0 | 106.2 |
| 10 | 2016-09-15 21:20 | 375 | 1h30min | 18.9 | 94.2 |
| 11 | 2011-06-28 19:50 | 273 | 11h40min | 25.1 | 93.6 |
| 12 | 2012-08-15 19:40 | 370 | 1h | 15.4 | 92.4 |
| 13 | 2011-08-22 23:40 | 375 | 12h | 33.4 | 92.4 |
| 14 | 2011-08-18 16:30 | 391 | 4h10min | 29.4 | 92.4 |
| 15 | 2016-06-23 20:20 | 380 | 3h30min | 27.5 | 90.6 |
| 16 | 2015-08-31 14:30 | 270 | 2h20min | 32.2 | 88.2 |
| 17 | 2009-07-03 14:10 | 391 | 2h10min | 38.0 | 88.2 |
| 18 | 2013-08-05 23:00 | 280 | 30min | 14.2 | 84.0 |
| 19 | 2012-06-21 20:00 | 290 | 3h10min | 17.2 | 82.2 |
| 20 | 2009-07-21 16:50 | 269 | 3h | 17.2 | 80.4 |
| 21 | 2016-06-15 10:50 | 277 | 7h30min | 34.5 | 80.4 |
| 22 | 2008-08-07 07:10 | 240 | 7h10min | 32.9 | 79.2 |
| 23 | 2008-07-26 18:10 | 270 | 8h10min | 26.8 | 78.6 |
| 24 | 2015-07-05 09:50 | 270 | 6h30min | 15.4 | 78.6 |
| 25 | 2016-06-23 | 344 | 10h10min | 32.8 | 78.6 |
| 26 | 2014-07-28 02:20 | 257 | 10h20min | 71.3 | 77.4 |
| 27 | 2009-07-14 12:20 | 286 | 3h20min | 17.5 | 77.4 |
| 28 | 2012-08-05 13:10 | 323 | 6h40min | 18.5 | 77.4 |
| 29 | 2009-05-25 20:50 | 260 | 6h30min | 23.8 | 76.8 |
| 30 | 2012-05-10 14:40 | 375 | 3h50min | 15.3 | 76.2 |
| 31 | 2014-07-10 23:20 | 269 | 50min | 20.7 | 75.6 |
| 32 | 2008-07-06 08:00 | 277 | 30min | 20.1 | 75.6 |
| 33 | 2009-06-09 10:50 | 319 | 8h20min | 24.8 | 75.6 |
| 34 | 2014-07-10 21:10 | 391 | 20min | 20.4 | 75.6 |
| 35 | 2008-09-11 23:50 | 265 | 16h40min | 41.8 | 74.4 |
| 36 | 2011-06-05 16:10 | 286 | 1h30min | 19.1 | 73.8 |
| 37 | 2015-08-24 15:00 | 269 | 3h40min | 13.3 | 70.8 |
| 38 | 2012-05-20 21:30 | 278 | 30min | 15.8 | 70.2 |
| 39 | 2013-07-27 21:40 | 350 | 2h10min | 33.6 | 70.2 |
| 40 | 2011-08-03 14:00 | 278 | 7h50min | 40.8 | 69.0 |
| 41 | 2011-08-23 10:40 | 283 | 1h30min | 16.5 | 69.0 |
| 42 | 2008-08-12 23:40 | 257 | 12h20min | 23.1 | 68.4 |
| 43 | 2010-07-14 15:50 | 377 | 1h30min | 16.7 | 68.4 |
| 44 | 2014-07-27 22:00 | 240 | 14h20min | 53.7 | 67.8 |
| 45 | 2009-05-15 05:00 | 273 | 16h20min | 28.8 | 67.8 |
| 46 | 2012-08-04 14:40 | 273 | 4h10min | 17.5 | 67.8 |
| 47 | 2013-07-27 23:50 | 278 | 50min | 20.5 | 67.8 |
| 48 | 2009-07-03 14:30 | 290 | 4h10min | 32.1 | 66.0 |
| 49 | 2015-08-14 18:10 | 310 | 4h | 21.7 | 66.0 |
| 50 | 2011-09-06 10:20 | 257 | 11h20min | 33.1 | 64.8 |

**Table A3.** Top 50 events for Finland

| Event | Starting Time [UTC] | Gauge | Duration | Amount [mm] | Peak [mmh$^{-1}$] |
|---|---|---|---|---|---|
| 1 | 2014-07-19 13:50 | 101787 | 2h30min | 34.7 | 89.1 |
| 2 | 2014-07-31 09:00 | 101103 | 1h20min | 18.1 | 87.5 |
| 3 | 2014-07-30 15:50 | 101289 | 19h20min | 34.8 | 86.6 |
| 4 | 2014-05-25 16:40 | 101555 | 29h50min | 31.6 | 84.2 |
| 5 | 2014-07-31 11:10 | 101690 | 3h00min | 51.0 | 83.9 |
| 6 | 2014-07-18 08:40 | 101799 | 2h00min | 25.7 | 83.2 |
| 7 | 2013-08-07 10:10 | 100951 | 15h | 25.9 | 82.4 |
| 8 | 2014-07-19 09:50 | 101194 | 50min | 14.6 | 79.1 |
| 9 | 2014-05-25 09:50 | 101339 | 26h | 48.4 | 78.6 |
| 10 | 2014-07-31 11:00 | 101787 | 4h | 28.4 | 78.1 |
| 11 | 2015-07-22 09:00 | 101603 | 2h30min | 29.4 | 77.9 |
| 12 | 2014-07-09 14:40 | 101800 | 20min | 22.1 | 76.6 |
| 13 | 2014-08-13 21:40 | 100908 | 6h50min | 28.9 | 74.2 |
| 14 | 2014-08-09 14:40 | 101826 | 30min | 16.3 | 72.8 |
| 15 | 2014-08-11 22:50 | 100953 | 3h20min | 37.3 | 71.6 |
| 16 | 2013-08-10 13:50 | 100917 | 40min | 14.1 | 69.2 |
| 17 | 2016-07-31 17:20 | 101572 | 2h10min | 21.2 | 68.3 |
| 18 | 2016-08-06 16:40 | 101338 | 1h | 35.2 | 68.2 |
| 19 | 2016-07-31 09:40 | 101555 | 11h20min | 27.9 | 67.5 |
| 20 | 2016-07-03 12:30 | 101603 | 7h30min | 67.1 | 66.9 |
| 21 | 2016-06-30 10:10 | 126736 | 25h50min | 63.9 | 66.2 |
| 22 | 2014-08-12 23:10 | 100955 | 8h | 20.1 | 65.6 |
| 23 | 2014-08-11 07:00 | 101726 | 4h30min | 13.5 | 65.6 |
| 24 | 2016-07-25 09:00 | 101743 | 6h20min | 25.9 | 65.6 |
| 25 | 2014-07-14 11:50 | 101339 | 1h30min | 23.2 | 65.0 |
| 26 | 2015-08-30 17:10 | 100953 | 20min | 15.8 | 65.0 |
| 27 | 2016-07-12 05:10 | 101537 | 3h10min | 21.4 | 64.7 |
| 28 | 2014-08-22 12:20 | 101805 | 2h | 16.3 | 63.6 |
| 29 | 2015-07-08 14:00 | 101537 | 25h10min | 46.3 | 62.9 |
| 30 | 2013-06-27 10:20 | 101338 | 8h30min | 33.2 | 62.1 |
| 31 | 2014-06-06 13:00 | 101690 | 6h30min | 16.7 | 61.4 |
| 32 | 2013-09-01 06:10 | 101272 | 9h30min | 33.0 | 61.2 |
| 33 | 2016-07-31 06:40 | 100974 | 3h40min | 21.6 | 61.0 |
| 34 | 2013-08-15 14:00 | 101124 | 50min | 14.0 | 60.5 |
| 35 | 2014-05-19 18:40 | 101537 | 4h10min | 21.4 | 59.6 |
| 36 | 2015-08-08 16:50 | 101632 | 2h30min | 11.3 | 58.9 |
| 37 | 2013-08-31 11:30 | 100955 | 3h20min | 30.0 | 58.7 |
| 38 | 2016-07-11 14:30 | 103794 | 11h30min | 14.1 | 58.4 |
| 39 | 2014-07-14 13:00 | 101555 | 2h10min | 20.2 | 58.1 |
| 40 | 2016-07-31 06:20 | 101632 | 6h30min | 16.5 | 58.1 |
| 41 | 2016-08-04 11:10 | 101194 | 7h | 18.1 | 58.0 |
| 42 | 2016-07-27 14:50 | 101950 | 20min | 13.2 | 57.3 |
| 43 | 2014-08-13 16:50 | 100967 | 3h40min | 12.1 | 56.8 |
| 44 | 2014-08-11 08:30 | 126736 | 3h20min | 13.4 | 56.7 |
| 45 | 2015-07-16 12:20 | 101103 | 24h30min | 69.5 | 56.6 |
| 46 | 2016-07-27 04:00 | 101805 | 5h20min | 16.6 | 55.5 |
| 47 | 2016-07-14 10:10 | 101933 | 1h | 20.4 | 55.2 |
| 48 | 2014-05-19 13:40 | 100967 | 20min | 13.3 | 55.1 |
| 49 | 2014-08-11 23:40 | 101603 | 12h10min | 42.4 | 53.9 |
| 50 | 2013-06-27 11:00 | 101150 | 5h10min | 19.2 | 53.2 |

**Table A4.** Top 50 events for Sweden

| Event | Starting Time [UTC] | Gauge | Duration | Amount [mm] | Peak [mmh$^{-1}$] |
|---|---|---|---|---|---|
| 1 | 2006-07-29 18:30 | 92410 | 1h30min | 44.0 | 91.2 |
| 2 | 2013-07-26 07:30 | 87140 | 3h45min | 48.2 | 81.2 |
| 3 | 2008-07-21 03:15 | 98490 | 7h45min | 51.5 | 71.2 |
| 4 | 2010-08-17 04:15 | 76420 | 8h15min | 26.3 | 67.2 |
| 5 | 2001-08-26 18:00 | 97280 | 19h15min | 54.0 | 62.4 |
| 6 | 2008-07-05 14:15 | 92410 | 1h | 16.8 | 60.4 |
| 7 | 2014-08-03 01:00 | 87140 | 1h30min | 28.6 | 54.8 |
| 8 | 2008-07-05 20:30 | 75520 | 37h45min | 53.1 | 53.6 |
| 9 | 2001-08-26 15:15 | 86420 | 19h30min | 38.8 | 52.0 |
| 10 | 2007-09-10 15:30 | 89230 | 17h15min | 51.1 | 51.6 |
| 11 | 2015-07-14 18:45 | 75520 | 3h | 25.9 | 49.6 |
| 12 | 2014-08-11 07:15 | 89230 | 2h30min | 26.4 | 49.6 |
| 13 | 2012-08-07 16:45 | 97280 | 5h45min | 16.5 | 48.8 |
| 14 | 2011-08-10 11:00 | 97280 | 2h45min | 33.4 | 48.0 |
| 15 | 2012-08-08 20:00 | 89230 | 9h45min | 39.9 | 47.2 |
| 16 | 2011-07-23 02:30 | 92410 | 1h | 18.8 | 45.2 |
| 17 | 2012-07-20 18:15 | 98490 | 11h45min | 24.7 | 45.2 |
| 18 | 2018-08-05 13:15 | 98490 | 3h45min | 15.1 | 44.8 |
| 19 | 2006-08-22 15:45 | 62040 | 21h | 50.4 | 41.6 |
| 20 | 2006-08-20 05:30 | 62040 | 14h15min | 27.4 | 41.2 |
| 21 | 2013-08-13 07:45 | 62040 | 35h15min | 81.2 | 41.2 |
| 22 | 2009-05-20 12:00 | 76420 | 7h30min | 17.6 | 41.2 |
| 23 | 2010-07-29 09:45 | 97280 | 8h15min | 36.4 | 40.8 |
| 24 | 2001-08-06 12:45 | 98490 | 3h | 17.3 | 40.4 |
| 25 | 2011-07-22 20:15 | 86420 | 8h45min | 13.7 | 40.0 |
| 26 | 2006-09-03 04:15 | 97280 | 4h45min | 19.5 | 40.0 |
| 27 | 2010-08-17 14:15 | 86420 | 2h45min | 20.4 | 39.6 |
| 28 | 2011-08-18 11:00 | 98490 | 4h45min | 10.5 | 39.6 |
| 29 | 2016-07-26 13:15 | 87140 | 45min | 17.6 | 38.8 |
| 30 | 2012-05-31 08:30 | 97280 | 10h45min | 20.8 | 38.8 |
| 31 | 2008-08-07 17:45 | 97280 | 16h15min | 34.5 | 38.4 |
| 32 | 2018-08-24 12:15 | 77210 | 3h15min | 18.4 | 37.6 |
| 33 | 2011-06-23 00:45 | 86420 | 8h | 39.4 | 37.6 |
| 34 | 2009-07-30 14:00 | 92410 | 2h30min | 24.3 | 37.6 |
| 35 | 2007-08-10 06:45 | 98490 | 5h45min | 20.2 | 37.6 |
| 36 | 2018-08-14 01:45 | 75520 | 18h30min | 55.5 | 37.2 |
| 37 | 2008-07-12 09:15 | 92410 | 3h30min | 19.3 | 37.2 |
| 38 | 2014-07-28 12:15 | 76420 | 2h15min | 15.0 | 36.8 |
| 39 | 2010-07-17 15:45 | 89230 | 5h | 13.9 | 36.8 |
| 40 | 2008-06-30 06:45 | 98490 | 5h45min | 14.8 | 36.8 |
| 41 | 2008-08-02 09:15 | 97280 | 13h30min | 33.7 | 36.4 |
| 42 | 2010-08-23 21:15 | 87140 | 4h | 24.0 | 35.6 |
| 43 | 2006-08-03 00:15 | 89230 | 5h | 41.9 | 35.6 |
| 44 | 2001-08-10 02:15 | 92410 | 26h45min | 27.1 | 35.6 |
| 45 | 2010-08-19 11:45 | 77210 | 5h45min | 25.2 | 35.2 |
| 46 | 2015-07-13 08:00 | 75520 | 22h15min | 30.1 | 34.8 |
| 47 | 2005-05-04 16:00 | 86420 | 1h | 14.0 | 34.8 |
| 48 | 2014-07-28 06:45 | 89230 | 1h30min | 15.8 | 34.8 |
| 49 | 2012-06-11 10:15 | 97280 | 2h | 16.4 | 34.8 |
| 50 | 2010-08-09 06:45 | 76420 | 8h | 15.0 | 34.0 |

**Table A5.** Top 10 events for Danish X-band product

| Event | Starting Time [UTC] | Gauge | Duration | Amount [mm] | Peak [mmh$^{-1}$] |
|---|---|---|---|---|---|
| 1 | 2017-08-01 18:15 | 5058 | 7h10min | 15.6 | 115.2 |
| 2 | 2016-07-25 13:35 | 5049 | 5h10min | 25.0 | 93.6 |
| 3 | 2016-07-25 13:55 | 5045 | 4h20min | 26.4 | 84.0 |
| 4 | 2017-08-01 18:20 | 5057 | 4h10min | 15.6 | 81.6 |
| 5 | 2017-08-15 18:15 | 5057 | 2h5min | 31.8 | 81.6 |
| 6 | 2017-08-15 18:15 | 5058 | 2h | 27.6 | 74.4 |
| 7 | 2017-06-16 01:15 | 5052 | 5min | 8.8 | 69.6 |
| 8 | 2017-08-18 12:50 | 5054 | 9h15min | 15.8 | 69.6 |
| 9 | 2017-06-15 21:45 | 5057 | 3h40min | 13.2 | 69.6 |
| 10 | 2016-06-16 15:50 | 5052 | 2h10min | 16.2 | 67.2 |