# Peer review of "The accuracy of weather radar in heavy rain: a comparative study for Denmark, the Netherlands, Finland and Sweden"

_Hydrology and Earth System Sciences, 2019_

## Referee Comment (RC1) · Anonymous Referee #1 · 18 Sep 2019

**Ref**: HESS_2019_427_review

**Title**: The accuracy of weather radar in heavy rain: a comparative study for Denmark, the Netherlands, Finland and Sweden.

**Authors**: Marc Schleiss*, Jonas Olsson, Peter Berg, Tero Niemi, Teemu Kokkonen, Søren Thorndahl, Rasmus Nielsen, Jesper Ellerbæk Nielsen, Denica Bozhinova, and Seppo Pulkkinen

**General comments**

This manuscript presents rainfall comparisons estimated from radar and rain gauge observations (in Denmark, the Netherlands, Finland, and Sweden for 50 top heavy rainfall events occurred in each country). Biases and some discrepancies were analyzed with respect to different temporal and spatial scales and rainfall (and peak) intensity to assess the performance of radar products capturing heavy rainfalls. Because of the differences in radar hardware, radar data processing, and the collected events from each country, the interpretation of the comparison is not straightforward and challenging but the authors made a worthwhile multinational effort to document the differences.

The topic fits to the scope of the journal's special issue, but the manuscript needs to be better clarified for its publication. Please see the comments below. Line numbers are indicated with "L"

Recommendation: Major revision required

**Major comments**

1. (Abstract) L15-19: Throughout the manuscript, supporting materials for urban hydrology and mitigations of attenuation are not presented. Revise this part and reflect what has been presented.
2. The link with hydrology or urban flooding/forecast:
   a. One of the objectives of this study is explicitly written to better understand the link between rainfall and urban flooding (L7-9) or/and the use of radar in hydrology and flood forecasting (L84-85). However, very few discussions were presented in this aspect. Add either more supporting martials for flooding parts (link with the presented work) or clarify better the objective of the presented work.
   b. Hydrological model (L171, L205, L397, L472, L490) has been mentioned in several sections without reference cited and the statements are rather generally made, which requires improvement in either writing or strengthening the explanation with more supporting materials (particularly for the statement made in the conclusion).
3. Better clarification and more supporting materials are required in results and conclusions (see the minor comments 16-37).

**Minor comments**

1. L10-L11: Clarify better "the top 50 events", "overall agreement", "the peaks" of what.
2. L44: need clarification of "accuracy" (of what).
3. L46-47: This term "higher-level" composite is less objective and vague. Rephrase it.

4. L59-60: ", the longest…15-20 years at best." Is it the case for world-wide or those countries presented in the manuscript?

5. L76-78: "Often…the results" This is not clearly written in the context. Specify better. Also, adding more backgrounds/references to support strong needs in multinational assessment and comparisons will be necessary. At least, in Europe, there has been an effort made with BALTRAD products (Michelson et al. 2018, referenced already in the manuscript but in later chapter) and with the OPERA products (e.g., Saltikoff et al. 2019, Park et al 2019), which can be referred in the introduction.

    Saltikoff, E.; Haase, G.; Delobbe, L.; Gaussiat, N.; Martet, M.; Idziorek, D.; Leijnse, H.; Novák, P.; Lukach, M.; Stephan, K. OPERA the Radar Project. Atmosphere 2019, 10, 320
    Park, S., Berenguer, M., Semper-Torres, D., 2019: Long-term analysis of gauge-adjusted radar rainfall accumulations at European scale. J. Hydrol., 573, 768-777. doi:10.1016/j.jhydrol.2019.03.093

6. Table 2: Clarify the data resolution original vs. used for the comparison, e.g., in the text Line 128, Danish data has been interpolated to 1 min. In Table 3, is the comparison also done 5 km not 1 km?

7. L153-154: reference missing for the operational product.

8. L164: "Polar radar measurements". Describe better, it seems a jargon, meaning radar measurement done at polar grid.

9. L170: After applying HIPRAD, the temporal/spatial resolution of the data remains the same shown in Table 2?

10. L178, "Aalborg" add country name and indicate the coverage of this radar in Fig 1.

11. L188: what is "tas BALTRAD"?

12. L206-208: Add reference

13. L290: "the HIPRAD" here, isn't it BALTRAD?

14. L249: "the highest available temporal" This term is used several times later, but isn't it the same as gauge sampling resolution (shown in table 1)? Is there any reason for such term? If so, explain better.

15. L 249: "Top event" → Event 1 (fig. 2), where these gauges located in Fig 1?

16. L253-254: Some results presented were already gauge adjusted and one (Finland) not. It is not clear to compare these numbers from literature examples (which is not clearly mentioned either if they (literature examples) were also derived before the adjustment or after?). Is it necessary?

17. L258: "The third rainfall peak" indicate here figure 4 (perhaps better with 4a indicating Denmark).

18. L264-265: "the relatively large peak intensity biases of 2.17, 2.09, 1.98 and 1.73 for Denmark, Finland, the Netherlands and Sweden…confirms this hypothesis" if the hypothesis refers the previous sentence, the bias for Netherlands should be larger than that of Finland because the peak intensity is higher for NL than for Finland (L256), isn't it?

19. L272 "at these scales" and L275 "such small scales". What does it mean? Is it related to storm scale? Or do you mean that the comparison was done with the instantaneous and point estimates (that affects representativeness error)?

20. L283: This is redundantly written (merge with L280-282)

21. L300-301: Are these numbers MB after the ARFs reduction applied? is it also shown in Table 3?

22. L302-302: Is the statement made before applying the ARFs? Clarify better. After ARFs, Swedish result shows the best, doesn't it?

23. L306-307: This does not support any argument and redundantly written in L300. Rephrase or remove it.
24. L324, L405: "deeper analysis" Avoid "deeper" (somewhat subjective word) and revise the sentence.
25. L325: "temporal aggregation time scale" → aggregation time scale (isn't it the same as shown in Figures 8-10?)
26. L338-339: "Furthermore, the quality….an important role". Add supporting explanation.
27. L359: It is not clear in Table 3 that the Danish products are the best in terms of RRMSE and CC. Revise this part.
28. L363-364: "However, a closer analysis….only 0.2", what does it mean?
29. L375-376: Clarify what is "viewpoints". Apart from the statement, how the attenuation and VPR correction applied to the group 2 data (Yes for Danish C band data, not explicitly indicated for the Swedish) were performed?
30. L379: "a coarser scale" in time or/and space?
31. L397-399: add reference. Is there any example run for the presented event?
32. L418: "the same order…than for…" → the same order…as for
33. L421-L422: This statement needs better supporting explanation, e.g., what dual-polarization capabilities was used in the processing of the data?
34. L469-470: "Bias correction…on peak intensity bias". Is this conclusion derived from all the presented cases for four countries? There are some explanations for the Dutch product (L348-349), but not easy to find for the others. For Finland, the presented examples are not even bias corrected, so it is not clear what the authors mean.
35. L471-472: Throughout the manuscript, "the importance of high-resolution radar observations in hydrological study" is hardly demonstrated/literature-reviewed with respect to the high-resolution radar products, which makes such conclusive statements weak. Add more solid outputs or references.
36. L488-489: Add references.
37. L489-490: Add references or strengthen supporting material for the referred rainfall uncertainties in hydrological models (e.g., some examples among any of the events -50 events*4 countries as a part of discussion or more explanation in L397-399).

---

## Referee Comment (RC2) · Anonymous Referee #2 · 24 Sep 2019

This paper compares the accuracy of weather radar rainfall using data from different countries (Denmark, Netherlands, Finland and Sweden). The study focuses on the top 50 heavy rainfall events which are more relevant for urban hydrology. The results showed that 1) radar underestimates rainfall rates; 2) radar products with higher spatial/temporal resolutions agree better with observations; 3) the combination of radar measurements from overlapping radars can improve rainfall rates. Although the results are interesting for the scientific community, there are a number of issues that the authors need to address before the paper is accepted for publication:

1) Raingauge data quality. The raingauge measurements used to validate the radar

observations come from different operational agencies. It is obvious that the quality of the gauge measurements is not going to be the same among the different agencies and therefore this could impact your results. There is no discussion about this in the paper.

2) Raingauge network density (Fig 1). It seems that the gauge network density is playing an important role in your results and there is little discussion on this. For Denmark the gauges are mainly clustered in a particular area (around 40-60km from radar site), for Finland the gauges are further away (beyond 50km) and cover different radars, for Sweden I can only see 4-5 gauges, whereas for the Netherlands all the gauges are more or less evenly distributed between 0-100km in range from the radar sites. This again will have important consequences in your results. For instance, VPR corrections will be important at far ranges. Attenuation due to heavy rain will also play a role. I will expect the radar rainfall error to increase with range and so the results will be better (or worse) depending on the location of the raingauge network.

3) Radar data quality. Every operational agency applies different corrections to the radar data. These corrections are extremely important and can help to explain some of the results. However, there is very little detail in the paper on the actual processing steps performed by each operational agency. Some corrections are discussed, but what about corrections for attenuation, VPR, partial beam blockage, etc for some of the countries. How do you ensure that the radar data have good data quality in both rain/no rain conditions? How does the operational agencies monitor the calibration of their radars (I do not mean comparisons with raingauge observations)? Do the bias corrections include the same (or some) of the gauges that you used for your validation? If so, what are the implications? I think this section deserves a more detailed summary.

4) The radars have different spatial/temporal resolutions. This is obviously a challenge when comparing the accuracy across different operational agencies. Would not be better to accumulate to the same spatial/temporal resolution (e.g. 2x2km, 15min) in order to have a fair assessment of the results? It seems to me that the different spatial

resolutions have important implications in your comparisons.

5) The use of ARF can help to explain the discrepancies, but I suggest to compare with the method proposed by Ciach and Krajewski (1999) which actually uses the spatial correlation of the rainfall field within the radar grid resolution to separate (or explain) the variance due to the fact that gauges represent a point whereas radar rainfall is an areal measurement from the total variance (see also Bringi et al, 2011).

6) Although the focus of heavy rainfall is important, what about the accuracy of radar rainfall for more conventional events (implications of the different corrections for radar errors) or in no rain conditions (e.g. implications of using robust clutter schemes, etc)? Are the results still consistent with those observed during heavy rainfall?

Other comments:

Fig 1. x/y labels? is that lat/lon?

Line 80. There is a reference that it is worth to look at related to the impact of spatial/temporal resolution in hydrodynamic modelling (Ochoa-Rodriguez et al, 2015).

Line 155. A lot of statements not justified: "Erroneous echoes and non-meteorological targets are removed using four different techniques. The algorithm used for correcting the vertical profile of reflectivity (VPR) is the same as in the operational product."

Line 160. "BRDC"?

Fig 5. did you accumulate to 1h? or it is 5min,15min ...and so on?

Table 2. can you include more radar specs? e.g. beamwidth, scanning rate, radome type, pulse width, etc that can affect the measurements.

Line 420. "The total accumulated rainfall amounts per event (i.e., 10-30 mm) were lower though, suggesting that the events sampled by the X-band system were rather short and localized." For x-band radars, sometimes the radar signal might be lost due to attenuation in heavy rain and without signal there is no way to apply any correction. Is

this the reason for the lower rainfall amounts? i think signal lost due to rain attenuation at X-band has to be carefully taken in to account.

References

Bringi, et al., 2011. Rainfall estimation with an operational polarimetric C-band radar in the United Kingdom: comparison with a gauge network and error analysis. Journal of Hydrometeorology, 12(5), pp.935-954.

Ciach, G. J., and W. F. Krajewski, 1999: On the estimation of radar rainfall error variance. Adv. Water Resour., 22, 585–595.

Ochoa-Rodriguez, S.,et al. 2015. Impact of spatial and temporal resolution of rainfall inputs on urban hydrodynamic modelling outputs: A multi-catchment investigation. Journal of Hydrology, 531, pp.389-407.

---

## Referee Comment (RC3) · Witold Krajewski (Referee) · 28 Sep 2019

It is a very nice simple-minded but important paper. We need results such as those reported in the paper to monitor our progress in variety of hydrologic problems. Radar-rainfall estimation is one of many of such problems in hydrology.

I have very few comments to suggest to improve the paper. Here they come:

1. The authors say little about the type of rain gauges used in the studies. "Automated" does not define the type and the type has implications for the expected errors (sampling). I suggest including the reference by Ciach (2003) if some of the gauges are tip-

[Figure]

ping buckets. 2. In the Conclusions, the authors say: "On average, the radar products with higher spatial resolutions were in better agreement with the gauges, thereby confirming the importance of high-resolution radar observations in hydrological studies." There are problems with this statement. First, it has been shown by several studies in the past that rain gauges have representativeness errors. The larger the area, the larger the error. Ciach and Krajewski (1999a,b) have established a framework on this that was followed my many subsequent studies. Therefore, it is expected that radar products with coarser resolution will show poorer agreement with rain gauges data. This says nothing regarding importance of high-resolution radar observations in hydrologic studies. In fact, for many applications the resolution is not the most important aspect of the radar-rainfall product. 3. The quality of the figures should be improved.

Figure 1. With wide spread of GIS technology, I would expect much better quality maps. At the very least distinguish land from sea. Is the same project used? Make the gauge locations solid dots so that they are better visible.

Figure 2. Don't repeat the dimension for each panel, the information is in the caption. The color scale is the same for all panels. Don't repeat it. By eliminating the legend and the axis description you gain space for the panels to be larger.

Figure 4. Just overall poor quality (aesthetically). Also, I do not thing that this figure adds much. They show just single event out of so many. I recommend removing it.

Figure 5 and 6. I recommend making all panels with the same scale range. This way you can remove the labels between the panels, make the panels larger, and make the dot larger.

Figure 7. Since you are using color in other figures, you can add color to this one. For example, you could use two shades of the color assigned to different countries to distinguish gauges and radar. This way you can remove the repetitive labels that clutter the figure.

[Figure]

Figure 8. Use the colors assigned to the countries to draw nice solid lines. You can add subtle light gray horizontal grid to the panels. Change the y-axis scale range to simplify the numbers, e.g. for the correlation you can use 0.5-1.0 range with horizontal lines only. The principle to follow here is to minimize the amount of ink for the same information content.

Figure 9. Very busy. You can de-clutter by simple removing the labels between the panels since both axes range is the same for all panels (good!)

Figure 10. Same as above. Did you explain the red bump for Denmark at 45 minutes scale?

Figure 11. I recommend remove the whole story of the X-band radar. Including it seems forced. That's not what the paper is all about. Write another study about the X-band radar performance.

Suggested references to add:

Ciach, G.J. and W.F. Krajewski, Radar-rain gauge comparisons under observational uncertainties, Journal of Applied Meteorology, 38(10), 1519–1525, 1999. Ciach, G.J. and W.F. Krajewski, On the estimation of radar rainfall error variance, Advances in Water Resources, 22(6), 585–595, 1999. Ciach, G. J., Local random errors in tipping-bucket rain gauge measurements. Journal of Atmospheric and Oceanic Technology, 20, 752–759, 2003.
* * *

---

## Author Comment (AC1) · 6 Nov 2019

**Referee 1:**

**Major comments:**

1. (Abstract) L15-19: Throughout the manuscript, supporting materials for urban hydrology and mitigations of attenuation are not presented. Revise this part and reflect what has been presented.

Response: We will add more supporting material and more in-depth discussion about the link between radar rainfall measurements and their use in urban hydrology. The discussion will be based on the following list of papers:

- Aronica, G., Freni, G., Oliveri, E., 2005. Uncertainty analysis of the influence of rainfall time resolution in the modelling of urban drainage systems. Hydrol. Process. 19, 1055–1071. https://doi.org/10.1002/hyp.5645
- Bruni, G., Reinoso, R., van de Giesen, N.C., Clemens, F.H.L.R., ten Veldhuis, J.A.E., 2015. On the sensitivity of urban hydrodynamic modelling to rainfall spatial and temporal resolution. Hydrol. Earth Syst. Sci. 19, 691–709. https://doi.org/10.5194/hess-19-691-2015
- Courty, L.G., Rico-Ramirez, M.Á., Pedrozo-Acuña, A., 2018. The Significance of the Spatial Variability of Rainfall on the Numerical Simulation of Urban Floods. Water 10, 207. https://doi.org/10.3390/w10020207
- Cristiano, E., ten Veldhius, M., van de Giesen, N., 2017. Spatial and temporal variability of rainfall and their effects on hydrological response in urban areas – a review. Hydrol. Earth Syst. Sci. 21, 3859–3878. https://doi.org/10.5194/hess-21-3859-2017
- He, X., Sonnenborg, T.O., Refsgaard, J.C., Vejen, F., Jensen, K.H., 2013. Evaluation of the value of radar QPE data and rain gauge data for hydrological modeling, Water Resources Research, 49 (9), pp. 5989-6005, https://doi.org/10.1002/wrcr.20471
- Löwe, R., Thorndahl, S., Mikkelsen, P.S., Rasmussen, M.R. and Madsen, H (2014), Probabilistic online runoff forecasting for urban catchments using inputs from rain gauges as well as statically and dynamically adjusted weather radar, Journal of Hydrology, Volume 512, 397-407, http://dx.doi.org/10.1016/j.jhydrol.2014.03.027
- Niemi, T.J., Warsta, L., Taka, M., Hickman, B., Pulkkinen, S., Krebs, G., Moisseev, D.N., Koivusalo, H., Kokkonen, T., 2017. Applicability of open rainfall data to event-scale urban rainfall-runoff modelling. J. Hydrol. 547, 143–155. https://doi.org/10.1016/j.jhydrol.2017.01.056
- Ochoa-Rodriguez, S., Wang, L.-P., Gires, A., Pina, R.D., Reinoso-Rondinel, R., Bruni, G., Ichiba, A., Gaitan, S., Cristiano, E., van Assel, J., Kroll, S., Murlà-Tuyls, D., Tisserand, B., Schertzer, D., Tchiguirinskaia, I., Onof, C., Willems, P., ten Veldhuis, M.-C., 2015. Impact of spatial and temporal resolution of rainfall inputs on urban hydrodynamic modelling outputs: A multi-catchment investigation. J. Hydrol. 531, Part 2, 389–407. https://doi.org/10.1016/j.jhydrol.2015.05.035
- Rafieeinasab, A., Norouzi, A., Kim, S., Habibi, H., Nazari, B., Seo, D.-J., Lee, H., Cosgrove, B., Cui, Z., 2015. Toward high-resolution flash flood prediction in large urban areas – Analysis of sensitivity to spatiotemporal resolution of rainfall input and hydrologic modeling. J. Hydrol. 531, Part 2, 370–388. https://doi.org/10.1016/j.jhydrol.2015.08.045
- Rico-Ramirez, M.A., Liguori, S., Schellart, A.N.A., 2015. Quantifying radar-rainfall uncertainties in urban drainage flow modelling. J. Hydrol. 528, 17–28. https://doi.org/10.1016/j.jhydrol.2015.05.057

- Thorndahl, S. Nielsen, J.E. and Jensen, D.G. (2016) Urban pluvial flood prediction: evaluating radar rainfall nowcasts and numerical weather prediction models as inputs. Water Science and Technology 74 (11) pp: 2599-2610 http://dx.doi.org/doi:10.2166/wst.2016.474
- Wright, D.B., Smith, J.A., Baeck, M.L., 2014. Flood frequency analysis using radar rainfall fields and stochastic storm transposition. Water Resour. Res. 50, 1592–1615. https://doi.org/10.1002/2013WR014224
- Yoon, S.-S., Lee, B., 2017. Effects of Using High-Density Rain Gauge Networks and Weather Radar Data on Urban Hydrological Analyses. Water 9, 931. https://doi.org/10.3390/w9120931
- Zhou, Z., Smith, J.A., Yang, L., Baeck, M.L., Chaney, M., Ten Veldhuis, M.-C., Deng, H., Liu, S., 2017. The complexities of urban flood response: Flood frequency analyses for the Charlotte Metropolitan Region. Water Resour. Res. 53, 7401–7425. https://doi.org/10.1002/2016WR019997

2a. The link with hydrology or urban flooding/forecast:
One of the objectives of this study is to better understand the link between rainfall and urban flooding (L7-9) or/and the use of radar in hydrology and flood forecasting (L84-85). However, very few discussions were presented in this aspect. Add either more supporting materials for flooding parts (link with the presented work) or clarify better the objective of the presented work.

Response: More in-depth discussion about these issues will be added during revision (both in the introduction and the results). See response to major comment 1 for more details.

2b. Hydrological model (L171, L205, L397, L472, L490) has been mentioned in several sections without reference cited and the statements are rather generally made, which requires improvement in either writing or strengthening the explanation with more supporting materials (particularly for the statement made in the conclusion).

Response: This will be made more explicit during revision.

3. Better clarification and more supporting materials are required in results and conclusions (see the minor comments 16-37).

Response: The results and conclusion sections will be rewritten taking into account all referee comments. In particular, we will try to make a better and more clear distinction between representativeness errors (areal vs point) and overall accuracy of the radar products (as suggested by referee 3) and to better highlight the link to hydrological response.

**Minor Comments:**

1. L10-L11: Clarify better "the top 50 events", "overall agreement", "the peaks" of what.

Response: The sentences will be clarified during revision.

2. L44: need clarification of "accuracy" (of what).

Response: This will be clarified during revision. In the context of this paper, accuracy primarily relates to bias and root mean square error.

3. L46-47: This term "higher-level" composite is less objective and vague. Rephrase it.

Response: OK

4. L59-60: ", the longest...15-20 years at best." Is it the case for world-wide or those countries presented in the manuscript?

Response: To the best of our knowledge, we believe that this is the case worldwide.

5. L76-78: "Often...the results" This is not clearly written in the context. Specify better. Also, adding more backgrounds/references to support strong needs in multinational assessment and comparisons will be necessary. At least, in Europe, there has been an effort made with BALTRAD products (Michelson et al. 2018, referenced already in the manuscript but in later chapter) and with the OPERA products (e.g., Saltikoff et al. 2019, Park et al 2019), which can be referred in the introduction.

Response: More background information about BALTRAD and more details about past international efforts for assessing and exchanging radar data will be added during revision (with references).

6. Table 2: Clarify the data resolution original vs. used for the comparison, e.g., in the text Line 128, Danish data has been interpolated to 1 min. In Table 3, is the comparison done at 5 min and at 1 min?

Response: The comparisons for Denmark were done at the 5 min resolution to match the resolution of the radar and gauge data. Although 1-min gauge data could be used in theory (using advection interpolation), this is not recommended here as this would add additional uncertainty due to interpolation. Also, the sampling uncertainty in rain gauge measurements at such short timescales would be very large.

7. L153-154: reference missing for the operational product.

Response: The following reference will be added during revision:

- Koistinen, J. and Pohjola, H., 2014. Estimation of Ground-Level Reflectivity Factor in Operational Weather Radar Networks Using VPR-Based Correction Ensembles, J. Appl. Meteor. Climatol. 53, 2394–2411, https://doi.org/10.1175/JAMC-D-13-0343.1

8. L164: "Polar radar measurements". Describe better, it seems a jargon, meaning radar measurement done at polar grid.

Response: Yes, the measurements are made over a polar grid and projected afterwards. The sentence will be clarified during revision.

9. L170: After applying HIPRAD, the temporal/spatial resolution of the data remains the same as shown in Table 2?

Response: Yes, the output has the same spatial and temporal resolution.

10. L178, "Aalborg" add country name and indicate the coverage of this radar in Fig1.

Response: Sure, no problem.

11. L188: what is "tas BALTRAD"?

Response: this is the official name of the product we used. But since this is confusing and probably not helpful for the reader, the name will be shortened to "BALTRAD" during revision.

12. L206-208: Add reference

Response: OK

13. L290: "the HIPRAD" here, isn't it BALTRAD?

Response: No, these are two different products.

14. L249: "the highest available temporal" This term is used several times later, but isn't it the same as gauge sampling resolution (shown in table 1)? Is there any reason for such term? If so, explain better.

Response: The highest available temporal resolution refers to the highest common time resolution at which both radar and gauge data are available. This will be clarified during revision.

15. L 249: "Top event" → Event 1 (fig. 2), where are these gauges located in Fig 1?

Response: The location of the gauges will be highlighted in the figure and their distances to the closest radar will be mentioned in the text.

16. L253-254: Some results presented were already gauge adjusted and one (Finland) not. It is not clear to compare these numbers from literature examples (which is not clearly mentioned either if they were also derived before the adjustment or after?). Is it necessary?

Response: Yes, we believe that such comparisons are useful. At the same time, we agree with the referee that this can be rather misleading if done improperly. We will clarify this during revision.

17. L258: "The third rainfall peak" indicate here figure 4 (perhaps better with 4a indicating Denmark).

Response: OK

18. L264-265: "the relatively large peak intensity biases of 2.17, 2.09, 1.98 and 1.73 for Denmark, Finland, the Netherlands and Sweden...confirms this hypothesis" if the hypothesis refers the previous sentence, the bias for Netherlands should be larger than that of Finland because the peak intensity is higher for NL than for Finland (L256), isn't it?

Response: The sentence will be changed to: "Clearly, the error structure between radar and gauges appears to fluctuate over time, with large deviations from the mean multiplicative bias in times of high rainfall intensities. As a result, the peak intensity biases (i.e., 2.17, 2.09, 1.98 and 1.73 for Denmark, Finland, the Netherlands and Sweden respectively) are systematically larger than the average bias over the whole event (i.e., 1.66, 1.37, 1.55 and 1.69).

19. L272 "at these scales" and L275 "such small scales". What does it mean? Is it related to storm scale? Or do you mean that the comparison was done with the instantaneous and point estimates (that affects representativeness error)?

Response: It means that the measurements are compared at high temporal resolutions (i.e., 5, 10 or 15 minutes depending on the radar product). At these scales, sampling effects can have a rather large impact on traditional error metrics such as bias and rmse. We will clarify this during revision.

20. L283: This is redundantly written (merge with L280-282)

Response: The lines will be merged.

21. L300-301: Are these numbers MB after the ARFs reduction applied? is it also shown in Table 3?

Response: Yes, the numbers on L300-301 are before/after ARF. Table 3 gives the value of the bias that can be explained by the ARF.

22. L302-302: Is the statement made before applying the ARFs? Clarify better. After ARFs, Swedish result shows the best, doesn't it?

Response: Yes, the Swedish product has the highest ARF. But if we take into account the part due to measurement support, its average multiplicative is the lowest of all. We understand that the way this is currently formulated in the paper might be confusing and will clarify this during revision (following up on a similar comment made by referee 3).

23. L306-307: This does not support any argument and redundantly written in L300. Rephrase or remove it.

Response: OK

24. L324, L405: "deeper analysis" Avoid "deeper" (somewhat subjective word) and revise the sentence.

Response: OK

25. L325: "temporal aggregation time scale" -> aggregation time scale (isn't it the same as shown in Figures 8-10?)

Response: Yes, the formulation was not ideal. The sentence will be changed during revision.

26. L338-339: "Furthermore, the quality....an important role". Add supporting explanation.

Response: We will provide more details on the radar and gauge data. See previous comments.

27. L359: It is not clear in Table 3 that the Danish products are the best in terms of RRMSE and CC. Revise this part.

Response: OK

28. L363-364: "However, a closer analysis....only 0.2", what does it mean?

Response: It means that we performed a correlation analysis and found a value of 0.2 between PIB and peak intensity (in mm/h). The formulation of the sentence will be improved during revision and replaced by: "However, the rank correlation coefficient between the PIB and peak intensity is only 0.20. Therefore, intensity is likely not the dominant factor at play here."

29. L375-376: Clarify what is "viewpoints". Apart from the statement, how the attenuation and VPR correction applied to the group 2 data (Yes for Danish C band data, not explicitly indicated for the Swedish) were performed?

Response: More details about the geometry of the rain gauge network and the distances of the gauges to the radars will be added during revision (see response to major comment 2, referee 2).

30. L379: "a coarser scale" in time or/and space?

Response: In time.

31. L397-399: add reference. Is there any example run for the presented event?

Response: No, there's no example run for this event. But we will provide references here to other papers cited in the introduction.

32. L418: "the same order...than for..." -> the same order...as for

Response: OK

33. L421-L422: This statement needs better supporting explanation, e.g., what dual-polarization capabilities was used in the processing of the data?

Response: This should now be clear thanks to the new information about the individual radar products. See response to major comment 3 of referee 2 for more details.

34. L469-470: "Bias correction...on peak intensity bias". Is this conclusion derived from all the presented cases for four countries? There are some explanations for the Dutch product (L348-349), but not easy to find for the others. For Finland, the presented examples are not even bias corrected, so it is not clear what the authors mean.

Response: This was mostly referring to the Dutch and Swedish products and will be rephrased.

35. L471-472: Throughout the manuscript, "the importance of high-resolution radar observations in hydrological study" is hardly demonstrated/literature-reviewed with respect to the high-resolution radar products, which makes such conclusive statements weak. Add more solid outputs or references.

Response: More references will be added in the introduction during revision (see major comment 1). In addition, we will try to clarify the importance of the observed biases in terms of predicting hydrologic response.

36. L488-489: Add references.

Response: OK, no problem.

37. L489-490: Add references or strengthen supporting material for the referred rainfall uncertainties in hydrological models (e.g., some examples among any of the events 50 events*4 countries as a part of discussion or more explanation in L397-399).

Response: OK. We will use some of the references mentioned in the response to comment 1

---

## Author Comment (AC2) · 6 Nov 2019

**Referee 2**

This paper compares the accuracy of weather radar rainfall using data from different countries (Denmark, Netherlands, Finland and Sweden). The study focuses on the top 50 heavy rainfall events which are more relevant for urban hydrology. The results showed that 1) radar underestimates rainfall rates; 2) radar products with higher spatial/temporal resolutions agree better with observations; 3) the combination of radar measurements from overlapping radars can improve rainfall rates. Although the results are interesting for the scientific community, there are a number of issues that the authors need to address before the paper is accepted for publication:

**Major comments:**

1) Rain gauge data quality. The rain gauge measurements used to validate the radar observations come from different operational agencies. It is obvious that the quality of the gauge measurements is not going to be the same among the different agencies and therefore this could impact your results. There is no discussion about this in the paper.

Response: Indeed, data quality plays a big role. To address this, more details about the type of rain gauges used in each country and their associated measurement errors/uncertainties will be added to the data section (see below). Systematic biases due to wind and calibration issues are important but unfortunately, we do not have enough information to reliably estimate them on an event-by-event basis. However, some typical values can be provided based on literature. Also, it is important to remind the reviewer that basic visual quality control has been performed on the gauge and radar data for each of the 50 events. Suspicious or obviously wrong measurements were discarded during this step. Most importantly, gauges are not considered as ground truth in this study. Rather, the goal is to describe the overall discrepancies between radar and gauge measurements, combining all sources of errors (i.e., gauges, radars, algorithms, humans) as well as differences in measurement scales.

Additional information about the rain gauges used in this study:

For Finland:
The gauges used in Finland are weight scales gauges of the type OTT Pluvio2 (https://www.ott.com). Observations are made according to World Meteorological Organization (WMO) regulations (WMO-No.8, CIMO Guide) with automatic quality control tests. Suspicious or erroneous values reported by the automatic tests are double checked manually. Measurements are based on weighing the mass of the liquid precipitation in the pluviometer, which is then converted into millimeters. A wind protector is used around the gauge. The opening of the gauge is placed at a height of 1.5 meters. For more details, see https://en.ilmatieteenlaitos.fi/weather-observations

For Denmark:
Some additional details about the Danish rain gauges can be found in Madsen et al. (2017) and the following 2 reports (in Danish):
https://www.dmi.dk/fileadmin/Rapporter/TR/tr06-15.pdf
https://www.dmi.dk/fileadmin/user_upload/Rapporter/TR/2016/DMI_Report_16_3.pdf
The full network is comprised of a combination of RIMCO tipping bucket gauges, OTT Pluvio2 weighing gauges and vibrating wire load sensors of type Geonor. But for this study, only the RIMCO tipping buckets were used.

For the Netherlands:
The automatic rain gauges in the Netherlands measure the precipitation depth using the displacement of a float in a reservoir (KNMI, 2000). The 10-min data from 2003-2017 used in this study have been validated internally by KNMI using a combination of automatic and manual quality control tests.

For Sweden:
Swedish gauges are of the type GEONOR ([www.geonor.no](www.geonor.no)) and consist of a bucket hanging from two chains and a metal thread that is kept in oscillation by an electromagnet. The frequency of the oscillations are transferred to a weight of the hydrometeors and is summed up. During the cold season, the bucket contains anti-freezing fluid which melts snow. The top pipe is also heated to remove larger chunks of snow and ice from clogging the opening. During the warm season an oil film is applied to keep evaporation at very low amounts. The sampling frequency is 15 min.

References:
- KNMI (2000), Handbook for the Meteorological Observation, 91–110 pp, De Bilt, Netherlands. [Available at http://projects.knmi.nl/hawa/pdf/Handbook_H01_H06.pdf]
- Madsen, H., Gregersen, I.B., Rosbjerg, D., Arnbjerg-Nielsen, K. (2017) Regional frequency analysis of short duration rainfall extremes using gridded daily rainfall data as co-variate, Water Science and Technology, 75 (8), pp. 1971-1981.

2) Rain gauge network density (Fig 1). It seems that the gauge network density is playing an important role in your results and there is little discussion on this. For Denmark the gauges are mainly clustered in a particular area (around 40-60km from radar site), for Finland the gauges are further away (beyond 50km) and cover different radars, for Sweden I can only see 4-5 gauges, whereas for the Netherlands all the gauges are more or less evenly distributed between 0-100km in range from the radar sites. This again will have important consequences in your results. For instance, VPR corrections will be important at far ranges. Attenuation due to heavy rain will also play a role. I will expect the radar rainfall error to increase with range and so the results will be better (or worse) depending on the location of the rain gauge network.

Response: Indeed, this was somewhat neglected during the analyses and discussion. To clarify this point, we will perform additional analysis and add more details about the spatial distribution of rain gauges, their distances to the radars and how much of the bias could be explained by these factors.

3) Radar data quality. Every operational agency applies different corrections to the radar data. These corrections are extremely important and can help to explain some of the results. However, there is very little detail in the paper on the actual processing steps performed by each operational agency. Some corrections are discussed, but what about corrections for attenuation, VPR, partial beam blockage, etc for some of the countries. How do you ensure that the radar data have good data quality in both rain/no rain conditions? How does the operational agencies monitor the calibration of their radars (I do not mean comparisons with rain gauge observations)? Do the bias corrections include the same (or some) of the gauges that you used for your validation? If so, what are the implications? I think this section deserves a more detailed summary.

Response: We agree this is a very important issue. But there are many factors at play and unfortunately, it is impossible to address all of them in this paper. To help with the interpretation, we will add as much information as possible about each radar product and how it was derived during revision.

For Finland:

The Finnish radar product is an experimental product from the FMI OSAPOL-project, which differs from the operational product used by the FMI mainly by making a better use of dual-polarization. The product is based on the data from years 2013-2016, during which the old single-polarization radars were being replaced by C-band dual-polarization Doppler radars. The product is therefore based on data from 4-8 dual-polarization radars depending on how many were available each year. The beam width of the radar measurements is 1 degree, the bin length is 500 m and the scanning is done in Pulse Pair Processing (PPP) mode. Doppler filtering is done first in the signal processing stage, and reflectivity measurements are calibrated based on solar signals (Holleman et al., 2010). These are followed by removal of non-meteorological targets using statistical clutter maps and fuzzy-logic-based HydroClass classification by Vaisala (Chandrasekar et al., 2013). Rainfall intensity is then estimated based on radar reflectivity (Z) and specific differential phase (Kdp) for each elevation angle. The reflectivity is attenuation-corrected (Gu et al., 2011), and the Kdp is estimated using the method described in Wang et al., 2009. For hydrometeors classified as precipitation, two alternative rain rate conversions are used. For heavy rain, i.e., Kdp>0.3 and Z>30 dBZ, the R(Kdp) relation given by $R=21Kdp^{0.72}$ (Leinonen et al., 2012) is used. For low to moderate intensities (i.e., Kdp<=0.3 or Z<=30 dBZ) and radar bins where HydroClass indicates non-liquid precipitation, a fixed Z(R) relation given by $Z = 223R^{1.53}$ (Leinonen et al., 2012) is used. A pseudo-CAPPI at 500 m height is produced from the rainfall intensity estimates using 4 lowest elevation angles and inverse distance-weighted interpolation with a Gaussian weight function. Finally, a composite VPR correction map (Koistinen et al., 2014) is applied to the resulting rainfall intensity fields that are produced at 1 km$^2$ and 5 min spatial and temporal resolution. The OSAPOL is the only product that is not gauge-adjusted.

For Denmark:

For the Danish C-band product, we don't have much more information than in the Thorndahl et al. (2014) paper. But we will ask the Danish Meteorological Institute DMI for more clarifications. In addition to that, two more references to He et al. (2013) and He et al. (2018) will be provided.

For the Netherlands:

The used product is a 10-year archive of 5~min precipitation depths at 1x1 km spatial resolution based on a composite of radar reflectivities from 2 C-band radars in De Bilt and Den Helder operated by the Royal Netherlands Meteorological Institute (KNMI). Note that the Netherlands recently upgraded their radars to dual-polarization. However, the dual-polarization rainfall estimates are not fully operational yet and all rainfall values used in this study were produced with the single-polarization algorithms. Also, the radar in De Bilt stopped contributing to the composite in the course of January 2017, at which point it was replaced by a new polarimetric radar in the nearby village of Herwijnen. For a detailed description of the processing chain, the reader is referred to Overeem et al. (2009). The radars used in this study were two single-polarization Selex (Gematronik) METEOR 360 AC Pulse radars with a wavelength of 5.2 cm, peak power of 365 kW, pulse repetition frequency of 250 Hz and 3-dB beamwidth of 1 degree. The scanning strategy consists of four azimuthal scans of 360 degrees at 4 elevation angles of 0.3, 1.1, 2.0, and 3.0 degrees. The data from these scans are combined into 5-min pseudo CAPPI according to the following procedure: for distances up to 60 km from the radar, only the highest elevation angle is used to reduce the risk of ground clutter and beam blockage. For distances of 15-80 km from the radar, the pseudo CAPPI is constructed by bilinear interpolation of the reflectivity values (in dBZ) of the nearest elevations below and above the 800-m height level. For distances of 80-200 km from the radar, only the reflectivity values of the lowest elevation angle are used, whereas it should be pointed out that the 800-m level only stays within the 3-dB beamwidth of the lowest elevation up to a range of about 150 km. Values beyond 200 km from the radar are ignored. Once the

pseudo CAPPI have been constructed, ground clutter and anomalous-propagation are removed using the procedure of Wessels and Beek-huis (1995), also described in Holleman and Beekhuis (2005). Spurious echoes within a radius of 15 km from the radar are mitigated based on the procedure described in Holleman (2007). A fixed Z-R relation of $Z = 200R^{1.6}$ is used to convert the reflectivities in the pseudo CAPPI to rainfall rates. During the conversion, reflectivity values are capped at 55 dBZ to suppress the influence of echoes induced by hail or strong residual clutter. Because of this, the maximum rainfall rate that can be estimated with this approach is 154 mm/h. Individual rainfall estimates from the two radars are then combined into one final composite using a weighting factor as a function of range r (km) from the radar (see Eq. 6 in Overeem et al. 2009). During the compositing, accumulations close to the radar are assigned lower weights to limit the impact of bright bands and spurious echoes. The composited rainfall rates are then adjusted for bias on an hourly basis using a network of 32 automatic rain gauges at 10 min resolution and 322 manual gauges at daily resolutions following the procedure of Holleman (2007). To improve spatial consistency, an additional bias correction at daily time scale (downscaled to hourly and 10 min scales) described in Overeem et al., (2009) is applied.

For Sweden:
The reference to Norin et al. (2015) is the only good source of information we could find. Some additional sentences will be added to the text specifying that the radars are being used for real-time operational production, and therefore prone to frequent changes and re-tuning. For example, the beam width of the radars has changed over time due to hardware upgrades and the scanning strategies and filters have been updated several times. Describing all these changes for each event and radar system quickly becomes unfeasible. Therefore, we prefer to adopt a more pragmatic approach by stating that both the technical aspects of the Swedish radar systems and their operation over the years (human and algorithm) will be assessed, with the assumption that single problematic events or radars will not have too large of an influence on the overall assessment.

References:
- Gu et al. 2011. Polarimetric Attenuation Correction in Heavy Rain at C Band, JAMC, 50, p. 39-58. https://doi.org/10.1175/2010JAMC2258.1
- He, X., Sonnenborg, T.O., Refsgaard, J.C., Vejen, F., Jensen, K.H., 2013. Evaluation of the value of radar QPE data and rain gauge data for hydrological modeling, Water Resources Research, 49 (9), pp. 5989-6005, https://doi.org/10.1002/wrcr.20471
- He, X., Koch, J., Zheng, C., Bøvith, T., Jensen, K.H, 2018. Comparison of simulated spatial patterns using rain gauge and polarimetric-radar-based precipitation data in catchment hydrological modeling, Journal of Hydrometeorology, 19 (8), pp. 1273-1288, https://doi.org/10.1175/JHM-D-17-0235.1
- Holleman, I., (2007): Bias adjustment and long-term verification of radar-based precipitation estimates. Meteor. Appl., 14, p.195–203, https://doi.org/10.1002/met.22
- Holleman, I. and Beekhuis, H. (2005): Review of the KNMI clutter removal scheme. Tech. Rep. TR-284, KNMI. [Available online at http://www.knmi.nl/publications/fulltexts/tr_clutter.pdf.]
- Holleman, I., Huuskonen, A., Kurri M. and Beekhuis, H. (2010): Operational Monitoring of Weather Radar Receiving Chain Using the Sun, Journal of Atmos. and Oceanic Tech., 27(1), p. 159-166, https://doi.org/10.1175/2009JTECHA1213.1
- Overeem, A., Holleman, I. And Buishand, A. (2009): Derivation of a 10-Year Radar-Based Climatology of Rainfall, JAMC, vol.48(), p.1448-1463, https://doi.org/10.1175/2009JAMC1954.1
- Wang. Y. and Chandrasekar, V., 2009. Algorithm for Estimation of the Specific Differential Phase, JTECH, 26(12), p. 2565-2578, https://doi.org/10.1175/2009JTECHA1358.1

- Wessels, H. R. A., and Beekhuis, J. H., 1995: Stepwise procedure for suppression of anomalous ground clutter. Proc. COST-75. Weather Radar Systems, International Seminar, Brussels, Belgium, COST, EUR 16013 EN, 270–277.

4) The radars have different spatial/temporal resolutions. This is obviously a challenge when comparing the accuracy across different operational agencies. Would not be better to accumulate to the same spatial/temporal resolution (e.g. 2x2km, 15min) in order to have a fair assessment of the results? It seems to me that the different spatial resolutions have important implications in your comparisons.

Response: Actually, because the Swedish product is at 15 min resolution and the Dutch product is at 10 min resolution, the smallest common resolution (assuming we don't want to interpolate) is 2x2 km and 30 minutes. This is rather coarse compared with the lifetime of convective cells and probably of lesser interest for most readers. We therefore think it is best to work at the highest possible resolution for the main parts of the analyses.

5) The use of ARF can help to explain the discrepancies, but I suggest to compare with the method proposed by Ciach and Krajewski (1999) which actually uses the spatial correlation of the rainfall field within the radar grid resolution to separate (or explain) the variance due to the fact that gauges represent a point whereas radar rainfall is an areal measurement from the total variance (see also Bringi et al, 2011).

Response: Thank you for the suggestion. We will compare our method to that of Ciach and Krajewski (1999) during revision to quantify how much of the bias could be explained by the spatial correlation of the rainfall fields. That being said, this is a rather delicate issue and may not necessarily result in more reliable estimates of ARFs as the latter are heavily influenced by radar data quality, the area chosen for analysis and the type of model used to represent small-scale variability. In the paper by Ciach and Krajweski (1999), an exponential function with "nugget" effect was used but other functional forms may lead to different estimates. Still, this is something that we will consider very carefully during revision.

6) Although the focus of heavy rainfall is important, what about the accuracy of radar rainfall for more conventional events (implications of the different corrections for radar errors) or in no rain conditions (e.g. implications of using robust clutter schemes, etc)? Are the results still consistent with those observed during heavy rainfall?

Response: Unfortunately, this is not feasible as only a small subset of the radar data archives has been processed so far (i.e. the top 100 events for each country, of which the 50 most intense after quality control were kept). However, it is worth pointing out that the 50 top events already contain a lot of "regular" time periods with low to moderate rainfall intensities. Therefore, a lot can be said already about the expected performance during conventional events. More information about this will be added during revision (e.g., by calculating the bias values after excluding the most intense time periods).

**Other comments:**

Fig 1. x/y labels? is that lat/lon?

Response: It's Lat/lon. The label will be added during revision.

Line 80. There is a reference that it is worth to look at related to the impact of spatial/temporal resolution in hydrodynamic modelling (Ochoa-Rodriguez et al, 2015).

Response: OK, thanks for the suggestion. We will add the reference.

Line 155. A lot of statements not justified: "Erroneous echoes and non-meteorological targets are removed using four different techniques. The algorithm used for correcting the vertical profile of reflectivity (VPR) is the same as in the operational product."

Response: Additional information about the radar products. See comment 3 of referee 2.

Line 160. "BRDC"?

Response: That's the name of the product. BRDC stands for BALTEX Radar Data Center, and BALTEX stands for Baltic Sea Experiment. A little note will be added in the text to explain this.

Fig 5. did you accumulate to 1h? or it is 5min,15min ...and so on?

Response: Fig 5 shows the results at 5, 10 and 15 min (as indicated)

Table 2. can you include more radar specs? e.g. beamwidth, scanning rate, radome type, pulse width, etc that can affect the measurements.

Response: Yes, some additional details will be added. See comment 3 of referee 2. However, it should be pointed out that these characteristics were not always constant over time due to hardware and software changes.

Line 420. "The total accumulated rainfall amounts per event (i.e., 10-30 mm) were lower though, suggesting that the events sampled by the X-band system were rather short and localized." For x-band radars, sometimes the radar signal might be lost due to attenuation in heavy rain and without signal there is no way to apply any correction. Is this the reason for the lower rainfall amounts? I think signal lost due to rain attenuation at X-band has to be carefully taken in to account.

Response: No, the signal was never lost during these events. We can't guarantee that the attenuation correction worked well but the relatively low bias and rmse suggests that there were no major issues. The lower rainfall amounts are simply due to the fact that there are only 2 years of data.

---

## Author Comment (AC3) · 6 Nov 2019

**Referee 3**

It is a very nice simple-minded but important paper. We need results such as those reported in the paper to monitor our progress in variety of hydrologic problems. Radar-rainfall estimation is one of many of such problems in hydrology. I have very few comments to suggest to improve the paper:

1. The authors say little about the type of rain gauges used in the studies. "Automated" does not define the type and the type has implications for the expected errors (sampling). I suggest including the reference by Ciach (2003) if some of the gauges are tipping buckets.

Response: Thank you for the suggestion. The reference to Ciach (2013) will be added and more details about the gauges will be provided. See comment 1, referee 2.

2. In the Conclusions, the authors say: "On average, the radar products with higher spatial resolutions were in better agreement with the gauges, thereby confirming the importance of high-resolution radar observations in hydrological studies." There are problems with this statement. First, it has been shown by several studies in the past that rain gauges have representativeness errors. The larger the area, the larger the error. Ciach and Krajewski (1999a,b) have established a framework on this that was followed my many subsequent studies. Therefore, it is expected that radar products with coarser resolution will show poorer agreement with rain gauges data. This says nothing regarding importance of high resolution radar observations in hydrologic studies. In fact, for many applications the resolution is not the most important aspect of the radar-rainfall product.

Response: The reviewer is right. A large part of the reported bias is due to representativeness errors. But this does not say much about the overall quality of the radar estimates. The revised version will contain more discussion about this issue and the problematic sentences will be reformulated to convey the right meaning. Note that gauges are not considered as ground truth in this study. Rather, the goal is to describe the overall discrepancies between radar and gauge measurements, combining all sources of errors (i.e., gauges, radars, algorithms, humans) as well as differences in measurement scales.

3. The quality of the figures should be improved.

Figure 1. With wide spread of GIS technology, I would expect much better quality maps. At the very least distinguish land from sea. Make the gauge locations solid dots so that they are better visible.

Response: A new fancier figure has been created (see attached file)

Figure 2. Don't repeat the dimension for each panel, the information is in the caption. The color scale is the same for all panels. Don't repeat it. By eliminating the legend and the axis description you gain space for the panels to be larger.

Response: OK, thanks for the suggestion.

Figure 4. Just overall poor quality (aesthetically). Also, I do not think that this figure adds much. They show just single event out of so many. I recommend removing it.

Response: In the authors' opinion, these figures are essential for understanding the time-dependent component of the error structure. We will improve the quality but keep the figure.

Figure 5 and 6. I recommend making all panels with the same scale range. This way you can remove the labels between the panels, make the panels larger, and make the dot larger.

Response: Thanks for the suggestion. But if we use the same scale for all panels, it becomes very hard to see the details for Sweden and Finland. A log-log scale also does not seem to be appropriate here. Therefore, we think it is best to keep the figure as it is now and add a small note in the caption to alert the reader about the different scales. The main idea here is to compare the correspondence between gauge and radar measurements in each country and not to compare event intensities between countries.

Figure 7. Since you are using color in other figures, you can add color to this one. For example, you could use two shades of the color assigned to different countries to distinguish gauges and radar. This way you can remove the repetitive labels that clutter the figure.

Response: OK, thanks for the suggestion.

Figure 8. Use the colors assigned to the countries to draw nice solid lines. You can add subtle light gray horizontal grid to the panels. Change the y-axis scale range to simplify the numbers, e.g. for the correlation you can use 0.5-1.0 range with horizontal lines only. The principle to follow here is to minimize the amount of ink for the same information content.

Response: OK, no problem

Figure 9. Very busy. You can de-clutter by simple removing the labels between the panels since both axes range is the same for all panels (good!)

Response: OK

Figure 10. Same as above. Did you explain the red bump for Denmark at 45 minutes scale?

Response: OK. Yes, the "bump" is already explained in the text.

Figure 11. I recommend remove the whole story of the X-band radar. Including it seems forced. That's not what the paper is all about. Write another study about the X-band radar performance.

Response: Yes, the story for the X-band radar is a bit different (shorter time period and different frequency). But we don't think it looks forced. The X-band data is not in the focus of the paper but provides additional interesting results at higher resolution.

Suggested references to add:
- Ciach, G.J. and W.F. Krajewski, Radar-rain gauge comparisons under observational uncertainties, Journal of Applied Meteorology, 38(10), 1519–1525, 1999.
- Ciach, G.J. and W.F. Krajewski, On the estimation of radar rainfall error variance, Advances in Water Resources, 22(6), 585–595, 1999.
- Ciach, G. J., Local random errors in tipping-bucket rain gauge measurements. Journal of Atmospheric and Oceanic Technology, 20, 752–759, 2003.

Response: Thanks for the suggestions! All three references will be added during revision.

---

## Author Comment (AC4) · 6 Nov 2019

Some very good points were raised during the discussion and many valuable suggestions for improving the quality of the paper were made. A lot of work still needs to be done but the overall impression of the three referees seems to be rather positive. Given the nature of the comments, we are very confident that we will be able to address all issues in a satisfactory manner during revision. Below, please find the response to each of the comments made by the reviewers during the discussion and how we plan to address them during revision. The most important modifications that will be made during revision are:

[Figure]

1. Introduction: - Following up on the comments made by referee 1, we will add more supporting material and more in-depth discussion about the link between radar rainfall measurements and their use in urban hydrology

2. Data section: - The revised paper will contain substantially more details about the different radar products and how they were generated. Same for the rain gauges.

3. Results section: - Additional analyses will be performed to study the spatial distribution of rain gauges, their distances to the radars and how these could explain the observed biases. - The current way of computing ARFs will be compared to an alternative method by Ciach and Krajewski (1999).

4. Results and Conclusion sections: - Parts of the results and conclusions will be rewritten to make a better and more clear distinction between representativeness errors (areal vs point) and overall accuracy of the radar products.

5. Figures - The figures will be updated according to the suggestions by referee 3.

---

## Author Response (AR1)

**List of Major Changes:**

Attached, please find the revised study about the accuracy of radar in time of heavy rain. As you can see, major changes were made to each of the sections in order to accommodate the referee's comments. The list of major changes include:

1. A new, longer introduction that emphasizes the importance of accurate rainfall estimates for hydrological applications.
2. More details about the individual gauge and radar products in Section 2.2
3. A new methodology for estimating the bias and quantifying the part of the bias due to differences in sampling volumes between radar and gauges (Section 2.3)
4. New results (Section 3) about the conditional bias with intensity and range.
5. Stronger, more precise conclusions with clear recommendations for future research (Section 4).
6. Two new figures  (Fig 7-8) for investigating the conditional bias with intensity and range.
7. One figure (previously Fig 8, RMSE vs time scale) was removed because it was redundant. Also, we feared that it could be confusing to interpret due to the fact that RMSE combines both information about scatter and bias and can therefore be hard to interpret.

**The authors would like to thank all the reviewers again for their constructive feedback and for their numerous suggestions that helped improve the paper. Below, please find a detailed answer to each referee's comments.**

**Referee 1:**

**Major comments:**

1. (Abstract) L15-19: Throughout the manuscript, supporting materials for urban hydrology and mitigations of attenuation are not presented. Revise this part and reflect what has been presented.

Response: The whole paper has been revised and several new references to urban hydrology have been included. The revised manuscript also contains a much more precise and in-depth discussion about how to mitigate peak intensity biases using rain gauges and polarimetry.

2a. The link with hydrology or urban flooding/forecast:
One of the objectives of this study is to better understand the link between rainfall and urban flooding (L7-9) or/and the use of radar in hydrology and flood forecasting (L84-85). However, very few discussions were presented in this aspect. Add either more supporting materials for flooding parts (link with the presented work) or clarify better the objective of the presented work.

Response: More details about the importance of accurate rainfall measurements for hydrology and flood forecasting were added in the Introduction.

2b. Hydrological model (L171, L205, L397, L472, L490) has been mentioned in several sections without reference cited and the statements are rather generally made, which requires improvement in either writing or strengthening the explanation with more supporting materials (particularly for the statement made in the conclusion).

Response: More references to hydrological modeling have been added and specific numbers are now given in the Introduction.

3. Better clarification and more supporting materials are required in results and conclusions (see the minor comments 16-37).

Response: The results and conclusion sections were completely rewritten during revision. Thanks to the new model in Section 2.3.1 and the new analyses of conditional bias and range-dependent bias, a much more precise quantification of the representativeness errors (areal vs point) and overall accuracy of the radar products is now possible.

**Minor Comments:**

1. L10-L11: Clarify better "the top 50 events", "overall agreement", "the peaks" of what.

Response: Done

2. L44: need clarification of "accuracy" (of what).

Response: Done

3. L46-47: This term "higher-level" composite is less objective and vague. Rephrase it.

Response: Done

4. L59-60: ", the longest...15-20 years at best." Is it the case for world-wide or those countries presented in the manuscript?

Response: To the best of our knowledge, we believe that this is the case worldwide.

5. L76-78: "Often...the results" This is not clearly written in the context. Specify better. Also, adding more backgrounds/references to support strong needs in multinational assessment and comparisons will be necessary. At least, in Europe, there has been an effort made with BALTRAD products (Michelson et al. 2018, referenced already in the manuscript but in later chapter) and with the OPERA products (e.g., Saltikoff et al. 2019, Park et al 2019), which can be referred in the introduction.

Response: We added the reference to the OPERA product and BALTRAD to the text and included some more details in the introduction to support the need for an international assessment and comparison.

6. Table 2: Clarify the data resolution original vs. used for the comparison, e.g., in the text Line 128, Danish data has been interpolated to 1 min. In Table 3, is the comparison done at 5 min and at 1 min?

Response: The comparisons for Denmark were done at the 5 min resolution to match the resolution of the radar and gauge data. Although 1-min gauge data could be used in theory (using advection interpolation), this is not recommended here as this would add additional uncertainty due to interpolation. Also, the gauges in Denmark are 0.1 mm RIMCO tipping buckets which means that the sampling uncertainty at 1 min would be very large.

7. L153-154: reference missing for the operational product.

Response: The reference to Koistinen et al. 2014 as been added to the text.

8. L164: "Polar radar measurements". Describe better, it seems a jargon, meaning radar measurement done at polar grid.

Response: Yes, the measurements are made over a polar grid and projected afterwards. The sentence has been reformulated to convex the right meaning.

9. L170: After applying HIPRAD, the temporal/spatial resolution of the data remains the same as shown in Table 2?

Response: Yes, the output has the same spatial and temporal resolution.

10. L178, "Aalborg" add country name and indicate the coverage of this radar in Fig1.

Response: Done

11. L188: what is "tas BALTRAD"?

Response: This product is now simply referred to as "BALTRAD".

12. L206-208: Add reference

Response: A reference to Rossa et al. (2011) has been added.

13. L290: "the HIPRAD" here, isn't it BALTRAD?

Response: No, these are two different products.

14. L249: "the highest available temporal" This term is used several times later, but isn't it the same as gauge sampling resolution (shown in table 1)? Is there any reason for such term? If so, explain better.

Response: The highest available temporal resolution refers to the highest common time resolution at which both radar and gauge data are available. This has been clarified during revision.

15. L 249: "Top event" → Event 1 (fig. 2), where are these gauges located in Fig 1?

Response: We are not allowed to disclose the exact location of the gauges but this is not really important here anyway. Indeed, as shown by Figure 8, there is no clear trend/bias with respect to the distance to the radar.

16. L253-254: Some results presented were already gauge adjusted and one (Finland) not. It is not clear to compare these numbers from literature examples (which is not clearly mentioned either if they were also derived before the adjustment or after?). Is it necessary?

Response: Yes, we believe that comparing them is useful. At the same time, we agree with the referee that this can be rather tricky and misleading if done improperly. We tried our best to clarify this during revision. Most literature values that we could find were for gauge-adjusted products. But a few studies have also looked at biases in non-adjusted products.

17. L258: "The third rainfall peak" indicate here figure 4 (perhaps better with 4a indicating Denmark).

Response: Done

18. L264-265: "the relatively large peak intensity biases of 2.17, 2.09, 1.98 and 1.73 for Denmark, Finland, the Netherlands and Sweden...confirms this hypothesis" if the hypothesis refers the previous sentence, the bias for Netherlands should be larger than that of Finland because the peak intensity is higher for NL than for Finland (L256), isn't it?

Response: This sentence does no longer exist in the revised version.

19. L272 "at these scales" and L275 "such small scales". What does it mean? Is it related to storm scale? Or do you mean that the comparison was done with the instantaneous and point estimates (that affects representativeness error)?

Response: It means that the measurements are compared at high temporal resolutions (i.e., 5, 10 or 15 minutes depending on the radar product). At these time scales, sampling effects can have a rather large impact on traditional error metrics such as bias and rmse. The sentence has been reformulated during revision and now reads as follows:

*"This is characteristic for sub-hourly aggregation time scales and can be explained by the large spatial and temporal variability of rainfall and the fact that radar and gauges do not measure precipitation at the same height and over the same volumes."*

20. L283: This is redundantly written (merge with L280-282)

Response: This sentence does not exists anymore in the revised version.

21. L300-301: Are these numbers MB after the ARFs reduction applied? is it also shown in Table 3?

Response: This sentence does not exist anymore in the revised version. A new Table (Table 4) now provides a better overview of ARFs and biases before/after correction for ARFs.

22. L302-302: Is the statement made before applying the ARFs? Clarify better. After ARFs, Swedish result shows the best, doesn't it?

Response: This part of the paper has been completely reformulated during revision and should now be easier to follow.

23. L306-307: This does not support any argument and redundantly written in L300. Rephrase or remove it.

Response: This part has been rewritten. Please see Section 3.4 (other sources of bias) for more details.

24. L324, L405: "deeper analysis" Avoid "deeper" (somewhat subjective word) and revise the sentence.

Response: Done

25. L325: "temporal aggregation time scale" -> aggregation time scale (isn't it the same as shown in Figures 8-10?)

Response:

26. L338-339: "Furthermore, the quality....an important role". Add supporting explanation.

Response: Done

27. L359: It is not clear in Table 3 that the Danish products are the best in terms of RRMSE and CC. Revise this part.

Response: Done

28. L363-364: "However, a closer analysis....only 0.2", what does it mean?

Response: The problematic sentence does not exist anymore.

29. L375-376: Clarify what is "viewpoints". Apart from the statement, how the attenuation and VPR correction applied to the group 2 data (Yes for Danish C band data, not explicitly indicated for the Swedish) were performed?

Response: More details about this have been added to the Data section.

30. L379: "a coarser scale" in time or/and space?

Response: In time (has been changed during revision).

31. L397-399: add reference. Is there any example run for the presented event?

Response: Some references have been added in the Introduction to explain this.

32. L418: "the same order...than for..." -> the same order...as for

Response: Done

33. L421-L422: This statement needs better supporting explanation, e.g., what dual-polarization capabilities was used in the processing of the data?

Response: This should now be clear thanks to the new information about the individual radar products. See response to major comment 3 of referee 2 for more details.

34. L469-470: "Bias correction...on peak intensity bias". Is this conclusion derived from all the presented cases for four countries? There are some explanations for the Dutch product (L348-349), but

not easy to find for the others. For Finland, the presented examples are not even bias corrected, so it is not clear what the authors mean.

Response: This sentence does not exist anymore in the revised version.

35. L471-472: Throughout the manuscript, "the importance of high-resolution radar observations in hydrological study" is hardly demonstrated/literature-reviewed with respect to the high-resolution radar products, which makes such conclusive statements weak. Add more solid outputs or references.

Response: More references have been added in the introduction, together with some explanations for why higher resolution is necessary and how it affects the timing and magnitude of predicted peak flows.

36. L488-489: Add references or strengthen supporting material for the referred rainfall uncertainties in hydrological models (e.g., some examples among any of the events 50 events*4 countries as a part of discussion or more explanation in L397-399).

Response: A reference to Bruni et al. (2015) has been added.

**Referee 2**

This paper compares the accuracy of weather radar rainfall using data from different countries (Denmark, Netherlands, Finland and Sweden). The study focuses on the top 50 heavy rainfall events which are more relevant for urban hydrology. The results showed that 1) radar underestimates rainfall rates; 2) radar products with higher spatial/temporal resolutions agree better with observations; 3) the combination of radar measurements from overlapping radars can improve rainfall rates. Although the results are interesting for the scientific community, there are a number of issues that the authors need to address before the paper is accepted for publication:

**Major comments:**

1) Rain gauge data quality. The rain gauge measurements used to validate the radar observations come from different operational agencies. It is obvious that the quality of the gauge measurements is not going to be the same among the different agencies and therefore this could impact your results. There is no discussion about this in the paper.

Response: Indeed, data quality plays a big role. More details about the type of rain gauges used in each country have been added to the data section. Systematic biases due to wind and calibration issues are important but unfortunately, we do not have enough information to reliably estimate them on an event-by-event basis. However, some typical values are now provided in the text based on literature. Also, it is important to remind the reviewer that basic visual quality control has been performed on the gauge and radar data for each of the 50 events. Suspicious or obviously wrong measurements were discarded during this step. Finally, note that gauges are not considered as ground truth in this study. Rather, the goal is to describe the overall discrepancies between radar and gauge measurements, combining all sources of errors (i.e., gauges, radars, algorithms, humans) as well as differences in measurement scales.

2) Rain gauge network density (Fig 1). It seems that the gauge network density is playing an important role in your results and there is little discussion on this. For Denmark the gauges are mainly clustered in a particular area (around 40-60km from radar site), for Finland the gauges are further away (beyond 50km) and cover different radars, for Sweden I can only see 4-5 gauges, whereas for the Netherlands all the gauges are more or less evenly distributed between 0-100km in range from the radar sites. This again will have important consequences in your results. For instance, VPR corrections will be important at far ranges. Attenuation due to heavy rain will also play a role. I will expect the radar rainfall error to increase with range and so the results will be better (or worse) depending on the location of the rain gauge network.

Response: Indeed, this was somewhat neglected during the analyses and discussion. Additional analyses were performed to study the range-dependent bias. Also, a new figure (Fig.8) has been added to show the distribution of rain gauges as a function of their distance to the radar(s). The analyses show that range-dependent biases are negligible compared to other factors (such as intensity-dependent bias).

3) Radar data quality. Every operational agency applies different corrections to the radar data. These corrections are extremely important and can help to explain some of the results. However, there is very little detail in the paper on the actual processing steps performed by each operational agency. Some corrections are discussed, but what about corrections for attenuation, VPR, partial beam blockage, etc for some of the countries. How do you ensure that the radar data have good data quality in both rain/no rain conditions? How does the operational agencies monitor the calibration of their radars (I do not mean comparisons with rain gauge observations)? Do the bias corrections include the same (or some) of the gauges that you used for your validation? If so, what are the implications? I think this section deserves a more detailed summary.

Response: We agree this is a very important issue. There are many factors at play here and unfortunately, it is impossible to address all of them in this paper. During revision, we added as much information as possible about each radar product, including how it was derived and what type of post-processing (e.g., bias adjustments) were applied. See Section 2.2 in the revised manuscript for more details. This makes it easier to interpret the different performances between countries.

4) The radars have different spatial/temporal resolutions. This is obviously a challenge when comparing the accuracy across different operational agencies. Would not be better to accumulate to the same spatial/temporal resolution (e.g. 2x2km, 15min) in order to have a fair assessment of the results? It seems to me that the different spatial resolutions have important implications in your comparisons.

Response: Actually, because the Swedish product is at 15 min resolution and the Dutch product is at 10 min resolution, the smallest common resolution (assuming we don't want to interpolate) is 2x2 km and 30 minutes. This is rather coarse compared with the lifetime of convective cells and probably of lesser interest for most readers. Also, aggregation to coarser scales is not recommended as simple arithmetic averaging of processed radar fields does not mimic what a lower resolution radar would see (e.g., due to the non-linear relation between rain rate and reflectivity and the multiple post-processing steps applied to the rainfall estimates). This is now clearly mentioned in the text (see Section 3.4). For all these reasons, we think it is best to work at the highest possible resolution for the main parts of the analyses.

5) The use of ARF can help to explain the discrepancies, but I suggest to compare with the method proposed by Ciach and Krajewski (1999) which actually uses the spatial correlation of the rainfall field within the radar grid resolution to separate (or explain) the variance due to the fact that gauges represent a point whereas radar rainfall is an areal measurement from the total variance (see also Bringi et al, 2011).

Response: Thank you for the suggestion. We carefully looked into the method of Ciach and Krajewski (1999) during revision and included their reference in the text. However, their approach was developed for additive error models and therefore not directly applicable to multiplicative biases. Moreover, our rain gauge networks were not dense enough to properly estimate the extension variance and estimating the nugget from the radar would not have made a lot of sense either since the radar data contain errors and biases. So instead, we opted for a comparatively simpler approach and proposed our own model (see Section 2.3.1) in which the differences in sampling volumes are an integral part of the random error terms (together with all other measurement errors). The two methods give different results, showing how difficult it is to separate the actual bias from bias due to the differences in measurement volumes.

6) Although the focus of heavy rainfall is important, what about the accuracy of radar rainfall for more conventional events (implications of the different corrections for radar errors) or in no rain conditions (e.g. implications of using robust clutter schemes, etc)? Are the results still consistent with those observed during heavy rainfall?

Response: Unfortunately, a systematic assessment of this issue was not feasible as only a small subset of the radar data archives has been processed so far (i.e. the top 100 events for each country, of which the 50 most intense after quality control were kept). However, it is worth pointing out that the 50 top events already contain a lot of "regular" time periods with low to moderate rainfall intensities. Therefore, a lot can be said already about the differences between average performance and performance during the peaks. In the revised paper, this is done by comparing the G/R ratios to the peak intensity bias (PIB) and quantifying the conditional bias with intensity (as done in Figure 7).

**Other comments:**

Fig 1. x/y labels? is that lat/lon?

Response: This figure has been replaced during revision.

Line 80. There is a reference that it is worth to look at related to the impact of spatial/temporal resolution in hydrodynamic modeling (Ochoa-Rodriguez et al, 2015).

Response: Thanks! The reference has been added to the Introduction.

Line 155. A lot of statements not justified: "Erroneous echoes and non-meteorological targets are removed using four different techniques. The algorithm used for correcting the vertical profile of reflectivity (VPR) is the same as in the operational product."

Response: Additional information about the radar products has been provided.

Line 160. "BRDC"?

Response: That's the name of the product. BRDC stands for BALTEX Radar Data Center, and BALTEX stands for Baltic Sea Experiment. A little note has been added in the text to explain this.

Fig 5. did you accumulate to 1h? or it is 5min,15min ...and so on?

Response: Fig 5 shows the results at 5, 10 and 15 min (as indicated)

Table 2. can you include more radar specs? e.g. beamwidth, scanning rate, radome type, pulse width, etc that can affect the measurements.

Response: Some additional details were added. However, it should be pointed out that these characteristics were not always constant over time due to hardware and software changes. This is now clearly stated in the text.

Line 420. "The total accumulated rainfall amounts per event (i.e., 10-30 mm) were lower though, suggesting that the events sampled by the X-band system were rather short and localized." For x-band radars, sometimes the radar signal might be lost due to attenuation in heavy rain and without signal there is no way to apply any correction. Is this the reason for the lower rainfall amounts? I think signal lost due to rain attenuation at X-band has to be carefully taken in to account.

Response: No, the signal was never lost during these events. We can't guarantee that the attenuation correction worked well but the results seem to suggest that there were no major issues. The lower rainfall amounts are simply due to the fact that there are only 2 years of data and that the events for the X-band radar observations were relatively short and localized compared with the others.

**Referee 3**

It is a very nice simple-minded but important paper. We need results such as those reported in the paper to monitor our progress in variety of hydrologic problems. Radar-rainfall estimation is one of many of such problems in hydrology. I have very few comments to suggest to improve the paper:

1. The authors say little about the type of rain gauges used in the studies. "Automated" does not define the type and the type has implications for the expected errors (sampling). I suggest including the reference by Ciach (2003) if some of the gauges are tipping buckets.

Response: Thank you for the suggestion. The reference to Ciach (2013) was added during revision and the new paper now contains more details about the gauge type (see Section 2.1) and their distance to the radar(s). Also, a new figure (Fig 8) has been added to the manuscript to show how the bias depends on the distance to the radar.

2. In the Conclusions, the authors say: "On average, the radar products with higher spatial resolutions were in better agreement with the gauges, thereby confirming the importance of high-resolution radar observations in hydrological studies." There are problems with this statement. First, it has been shown by several studies in the past that rain gauges have representativeness errors. The larger the area, the larger the error. Ciach and Krajewski (1999a,b) have established a framework on this that was followed my many subsequent studies. Therefore, it is expected that radar products with coarser resolution will show poorer agreement with rain gauges data. This says nothing regarding importance of high

resolution radar observations in hydrologic studies. In fact, for many applications the resolution is not the most important aspect of the radar-rainfall product.

Response: Thanks for the comment. We completely revised the paper to better account for this issue. The most important change is the new statistical model in Section 2.3 for separating the measurement support bias from the actual bias. Also, we paid more attention to conditional bias with intensity and range. Using this new model, we rewrote the entire results sections and conclusions.

3. The quality of the figures should be improved.

Response: We followed the reviewer's suggestions and did our best to improve the quality of the figures (when judged necessary).

4. I recommend removing the whole story of the X-band radar. Including it seems forced. That's not what the paper is all about. Write another study about the X-band radar performance.

Response: Yes, the story for the X-band radar is a bit different (shorter time period and different frequency). But we don't think it looks forced. The X-band data is not in the focus of the paper but it provides additional interesting results at higher resolutions that strengthen the conclusions of the paper.

[revised manuscript text omitted]
 2011. Their analysis covered more than 15 years but the radar data used to derive the statistics were not continuous in time. intensity-dependent biases. Quantifying these residual errors and studying their propagation in hydrological models is crucial for improving the timing and accuracy of flood predictions (Cunha et al., 2012; Bruni et al., 2015; Courty et al., 2018; Niemi et al., 2017). For example, in their study, Stransky et al. (2007) estimated that the propagation of biased radar measurements in urban drainage models could result in up to 30-45% errors in terms of peak flow magnitude. To limit error propagation, Schilling (1991) recommended that the bias affecting areal-averaged rainfall intensities should not exceed 10%.

Because of the difficulty to get long homogeneous radar archives, the studies published so far mostly focused on regional or national performances. Often, the methodologies used to carry out the analyses were different, which makes it hard to compare the results. Consequently, there is a strong need for systematic, multinational assessments and comparisons Over the years, each country has developed its own strategy for mitigating errors and biases in operational radar rainfall estimates. However, since there is no common benchmark and few international studies are available, the merits and weaknesses of each approach remain difficult to quantify objectively. This study sheds new light on current performances by conducting a multinational assessment of radar's ability to capture heavy rain . This paper sheds new light on this issue by providing a detailed analysis of events at scales of 5 min up to 2 hours. In total, 6 different radar products across 4 European countries (i.e., Denmark, the Netherlands, Finland and Sweden) . Inspired by the approach of Thorndahl et al. (2014b), we selected are considered. Special emphasis is put on analyzing the performance during the 50 most intense events for each country over the last 10 yearsto study the average agreement between radar and gauges as well as the discrepancies in terms of peak rainfall intensities. The study is performed within the context of the Water JPI funded 
[revised manuscript text omitted]

$\epsilon_{\text{rel}}$ For readers not familiar with the interpretation of multiplicative biases, note that it is also possible to express the G/R ratio and model bias $\beta$ as an average relative error. In this case, we have:

$$\underbrace{\epsilon_{\text{rel}}}\,Err_{\text{avg}} = \underline{100\%}\cdot\mathbb{E}\left[\frac{Y_i - X_i}{Y_i}\,\frac{R_g(t) - R_r(t)}{R_g(t)}\right] = 1 - \frac{1}{\text{MB}}\frac{1}{\beta}\cdot\mathbb{E}\left[\frac{1}{\varepsilon_i}\,\frac{1}{\varepsilon(t)}\right] = 1 - \frac{1}{\text{MB}}1 - \frac{\exp(\sigma_\varepsilon^2)\cdot\left(\exp(\sigma_\varepsilon^2)-1\right)}{\beta} \tag{6}$$

[revised manuscript text omitted]

~~The large scatter and relatively large RRMSE values of 116.4% to 139.1% highlight the strong disagreements between radar and gauge estimates at these scales. This is normal and can be explained by the fact that radar and gauges do not measure at the same height and over the same volume. It is important to note also that the gauge integrates precipitation over time whereas radar takes snapshots. Wind effects, changing microphysics and sampling uncertainties therefore also play an important role at such small scales. Despite the large scatter, linear correlation coefficients are relatively high~~ 
[revised manuscript text omitted]

625 A deeper analysis of this issue confirms that on average, higher rainfall intensities appear to be linked with slightly larger multiplicative biases

**3.4 Other sources of bias**

The conditional bias with intensity explains a lot of the differences between the radar products. However, this is only one part of the story and other confounding factors such as the distance between the radar(s) and the gauges also need to be considered.

630 Figure 8 shows the log-ratio of gauge versus radar estimates $\ln(\frac{R_g(t)}{R_r(t)})$ as a function of the distance to the nearest radar. Compared with intensity, the trend with distance appears to be much weaker. Out of the 4 considered products, only the Danish C-band exhibits a trend that is significantly different from zero (at the 5% level). This makes sense given that the Danish product only considers data from a single radar and only applies a mean field bias correction, making it more likely to be affected by range effects such as overshooting, non-uniform beam filling and attenuation. Based on our analyses, the multiplicative bias $\beta$

635 increases by 0.73% per km. However, since the range of distances between radar and gauges in Denmark is relatively small (from 29.2 to 74.2 km), bias values only vary from 1.06 to 1.47 at minimum and maximum distances respectively. Distance therefore only plays a minor role in explaining the variations in bias compared with intensity. Interestingly, the composite products in the Netherlands and Finland do not seem to suffer from significant conditional biases with distance, highlighting the advantage of combining data from different radars and viewpoints to mitigate range effects. The Swedish product currently does

640 not combine measurements from multiple radars in an optimal way, only using the measurements from the best (i.e., nearest) radar. However, the link between the bias and the average intensity remains rather weak, with rank correlation values of 0.33 in the Netherlands, 0.30 in Denmark, 0.04 in Finland and 0.19 in Sweden. Still, there appears to be a strong contrast between the average discrepancies between radar and gauges at the event scale, as shown in Figure ??(a) , and the large mismatches in terms of peak rainfall intensities in Figure ??(b) . In most cases, the highest intensities measured by the radar over the top 50

645 events barely match the lowest peak intensities measured by the gauges. The bias therefore appears to be largely influenced by event duration and the presence of lower rainfall intensities for which radar and gauges tend to be in better agreement than during the peaks. Swedish BRDC also contains an additional range-dependent bias correction (see Section 2.2.4) that appears to be rather efficient at removing large-scale trends with distance. However, the strong conditional bias with intensity in the Swedish BRDC also makes it harder to see potential range-dependent biases in the first place.

[revised manuscript text omitted]

690 **3.5 Agreement during the peaks**

 In this section, we take a closer look at  how well the rainfall peaks are captured by the radar. Figure 9 shows the  10%, 25%, 50%, 75% and 90% quantiles of peak intensity bias between radar and gauges as a function of aggregation time scale . The dashed horizontal lines denote the average
695  apparent bias (i.e., the G/R ratio). We see that the Netherlands and Finland  have relatively low median peak intensity biases of 1.82 and 1.88 at 10 min
700 ~~overall bias previously calculated for all 50 events. The hourly mean field bias correction in the Dutch product does not appear to provide a big advantage in terms of peak intensities, which could be expected given that gauge adjustments are applied at a lower resolution and do not specifically target peak intensities. Also, note how in the Finnish product, rainfall peaks tend to be underestimated only slightly more (i.e., +11.2%). They also appear to converge faster to the average MB value than in the Dutch product. This is interesting and could point to the benefits~~
705
 bias). Denmark and Sweden  on the other hand have substantially higher median PIB values of 2.96
710 and 2.24, (1.86 respectively 1.35 times higher than the average). Moreover, the rate at which the PIB decreases with aggregation time scale is different in each country. In Denmark and Sweden, the PIB remains well above the average  bias for all aggregation time scales
715  up to 2 , ~~shows a better agreement during the peaks. A possible explanation for this surprising result could be that the rain events in the Danish database are more intense and shorter than in the other countries. However, a closer analysis reveals a rank correlation coefficient between the PIB and peak intensity of only 0.20. Therefore, intensity is likely not the dominant factor at play here. Another explanation could be that bias-adjustment in the Danish radar product is performed on the basis of daily rainfall~~

[revised manuscript text omitted]
. ~~Looking at the RRMSE, we see that the Finnish and Swedish products agree slightly better with the gauges than BALTRAD (-4.12% and -4.52% respectively) while the Danish agrees slightly worse (+2.47%). There are many possible explanations for these differences and each case needs to be analyzed separately. For Sweden, the interpretation is rather easy: the only major difference between the Swedish BRDC product and the BALTRAD lies in the additional bias-correction scheme implemented in HIPRAD. Otherwise, everything is identical. Thus we can say with high confidence that the reduction in RRMSE between BALTRAD and BRDC is likely due to the use of the bias-adjustment scheme. This , however, does not appear to improve significantly the bias affecting the peak rainfall intensities, as shown by the boxplots in the lower panel of Figure 13. The Finnish product shows similar improvements in RRMSE compared with the BALTRAD as well as a slightly lower spread~~ Overall, the BALTRAD seems to perform rather similarly to the national products. It has slightly lower rank correlation coefficients and higher root mean square differences. The bias (as measured by the G/R ratio) is also very similar, except in Sweden where the BALTRAD appears to underestimate more with respect to the gauges (1.77 versus 1.66). This makes sense given that the BALTRAD does not include the HIPRAD adjustments which results in higher overall bias and conditional bias with intensity. Interestingly, the BALTRAD performs worse than the Danish C-band product in terms of overall bias but better in terms of median peak intensity bias.  There are many possible explanations for these differences. One reason could be the

[revised manuscript text omitted]

Overall, the X-band data for Denmark showed  promising results, outperforming all other C-band products in terms of accuracy and correlation, thereby demonstrating the value of high-resolution rainfall observations for urban hydrology. However,  due to the shorter data record, only 10 events over 2 years ~~were consideredfor the X-band radar analysis. Polarimetry also seemed to provide a slight advantage in times of heavy rain. However, due to the many confounding factors, it is hard to precisely quantify its added-value within the framework of this study. What we can say with high confidence is that dual-polarization and higher resolution alone are not sufficient to get reliable estimates of peak rainfall intensities. Other factors such as the ability to combine data from multiple radars and viewpoints seem to play a much more important role, as demonstrated by the superiorDutch and Finnish C-band products (despite their slightly lower resolution). By contrast, the single-radarproduct in Denmark, which had the highest spatial resolution (i.e., 500 m) and lowest overall RRMSE, did not perform well on the peaks at all, exhibiting the highest peak intensity biases of all 6 products. Even the lower resolution BALTRAD composite (2 km, 15 min) over Denmark performed better~~
[revised manuscript text omitted]

---

## Referee Report (RR1)

**Ref**: HESS_2019_427_version2_review
**Title**: The accuracy of weather radar in heavy rain: a comparative study for Denmark, the Netherlands, Finland and Sweden.

**Authors**: Marc Schleiss*, Jonas Olsson, Peter Berg, Tero Niemi, Teemu Kokkonen, Søren Thorndahl, Rasmus Nielsen, Jesper Ellerbæk Nielsen, Denica Bozhinova, and Seppo Pulkkinen

**General comments**

The authors have tried to perform a new bias estimation (section 2.3.1) and to present new analyses (section 3), which are nice ideas. However, the revised manuscript still lacks clarity on writing and the delivery of messages, which needs to be improved. In the following, I tried to point out some, but the authors should go through their manuscript more carefully and not to jump to conclusions without providing clear evidences.

I would not recommend for the publication without a major revision.

1. L368-372: The modelled bias by Eq. [5] for the Netherlands should be about 0.8. Where does 1.23 come from? The explanation on this value (that "the radar values even seem to be overestimated") is neither clear (e.g., if the bias refers to rainfall intensity, mm/h as shown in Y-axis, or to total accumulation for this period shown in the upper left – "gauge 56.9 mm, radar 41.6 mm") nor agree with the statement in L356 (the systematic pattern of "underestimation"), though the authors stated that such values should be carefully interpreted.
2. L420-421: This was a major conclusion (also appears in abstract). Does "the actual bias" mean the modelled one? If so, it cannot be "actual". If not, it is not clear where and how such interpretation has been derived, given that i) the results based on eq.5 which require further analyses due to its assumption, and ii) the results (table1) were made only with highest aggregation time so what the authors mean by "after accounting for differences in scale"?
3. L429-455: i) Definition of the "conditional bias" with respect to the rainfall intensity is not clear. In L434, it seems that it refers to the "log ratio of ($R\_g/R\_r$)" with respect to the rainfall intensity, which is not the same as the modelled bias $\beta$ shown in Eq 1, yet the results were described in terms of "bias", ii) in Figure 7, was 0 mm (no rain) is also included and if so why? How would the error bar for the fit (or something that tells a significance level of the fit) look in this graph? iii) L438, this looks the slope of red line, does it make sense to use %? iv) It is not easy to follow L442-L444, can the authors explain better where we see such evidence; e.g., the differences in apparent biases shown in Figure 6 look small among Denmark (1.59), Sweden (1.66), Finland (1.56) and the NL (1.4).
4. Section 3.6: Although the authors introduced the comparison with X-band radar, in L248-250, "the X-band data can be used to provide valuable insight into the advantages and challenges associated with using high-resolution X-band radar measurement in times of heavy rain", its analyses (L582-587) are poorly written and hardly provide insight as they aimed. If this radar is dual-pol, the results could have calculated with Kdp to prove their speculations (L583-585). It is also questionable in conclusion on the X-band data for Denmark (L640).
5. L547-550: where do we see that the "*multiplicative biases* between radar and gauges can *amplify* when data are aggregated to coarser time scales?" If the authors meant the "multiplicative bias" as the "peak intensity bias", we still see that the bias values get smaller with coarser aggregation time (Figure 10). Similarly, it is not clear what

this means in L550 that "single-radar products with daily (coarser than sub-hourly) rain gauge adjustments are *more vulnerable to error amplification*". Can the authors provide a clearer example of error amplification and when this gets more vulnerable?

6. L551-559: Despite the explanation of Fig.11, it is not clearly written how significantly this can affect to the choice of hydrological models for flood prediction (L549). Also, in L558-559, how this should be interpreted for both bias correction and reduction to be applied for hydrological models?

7. L582-587: Improve writing with clearer message.

**Minor comments**

1. L112: Time period for KNMI gauges (2003-2017) does not match with those indicated in Table 1.

2. L278-279: does this "[-]" mean unitless? Is it necessary?

3. L421-422: "Moreover,.." What does this sentence mean?

4. L535-536 (Figure 10). Because the presented case is the same as shown in Figure 4, the peak intensity bias values and unit at the first interval (at the highest temporal resolution) are expected to be identical as those mentioned in L373-374 for each event. However, as seen right Y-axis in Fig. 10, the "peak intensity bias" is expressed in terms of in %, i.e. 100*(1- 1/(factor))". Explain this value (technically it is not the "peak intensity bias") better in the text, and revise the sentence.

5. Figure 11, revise the title of the X-axis.

---

## Author Response (AR2)

**To the editor and reviewers:**

The authors would like to thank both referees and the editor for their time. Referee 1 was particularly meticulous, pointing out several small remaining issues and unclarities in the text. Thank you for this! Attached, please find the revised text and a point-by-point response to the reviewers' comments. Since there were no major criticisms left regarding the methods, results and conclusions, we think that the paper is now ready to be published.

**Reviewer 1**

**Major comments:**

1. L368-372: The modelled bias by Eq. [5] for the Netherlands should be about 0.8. Where does 1.23 come from? The explanation on this value (that "the radar values even seem to be overestimated") is neither clear (e.g., if the bias refers to rainfall intensity, mm/h as shown in Y-axis, or to total accumulation for this period shown in the upper left – "gauge 56.9 mm, radar 41.6 mm") nor agree with the statement in L356 (the systematic pattern of "underestimation"), though the authors stated that such values should be carefully interpreted.

> The radar underestimates by a factor 0.81. Therefore, it overestimates by a factor 1/0.81 = 1.23. This may not have been formulated very clearly. The corresponding text has been revised and some additional information has been added to help the reader make sense of the numbers.
>
> *"The true underlying model bias beta for the 4 depicted events is therefore estimated to be 1.04, 0.81, 1.02 and 1.18. In other words, once the differences in scale between radar and gauge data have been accounted for, radar only appears to underestimate rainfall rates by a factor 1.04 (3.8%) in Denmark, 1.02 (2.0%) in Finland and 1.18 (15.3%) in Sweden. In the Netherlands, the radar values even seem to be overestimated by a factor 1/0.81 = 1.23 (18.7%). The fact that radar might overestimate rainfall rates compared with gauges may seem contradictory at first (given that actual values are lower) but can be explained by the fact that beta also accounts for the relative variability of the radar and gauge observations. Nevertheless, beta values should be interpreted very carefully as they rely on the assumption that the errors between radar and gauges are independent and log-normally distributed with median 1. Figure 4 suggests that this might not always be the case. In particular, the bias between radar and gauges appears to increase during the peaks (see Section 3.3 for more details). In this case, the peak intensity biases for the top events in each country were 2.17 (Denmark), 2.09 (Finland), 1.98 (Netherlands) and 1.73 (Sweden), which is consistently larger than the average bias (as measured by the G/R ratio)."*

2. L420-421: This was a major conclusion (also appears in abstract). Does "the actual bias" mean the modelled one? If so, it cannot be "actual". If not, it is not clear where and how such interpretation has been derived, given that i) the results based on eq.5 which require further analyses due to its assumption

> Indeed, the terminology is important here. To make this clearer, we added a few sentences in the Methods section to explain to the reader what we mean by "apparent" and "actual" bias.
>
> *"To avoid any confusion, the following terminology is adopted:*
> * *The apparent bias (i.e., seemingly real or true, but not necessarily so) is the one that we see in the data. It is measured using the G/R ratio.*
> * *The actual bias (i.e., existing in fact; real) is the unknown underlying bias, i.e., the bias that we would measure if radar and gauges would have the same sampling volumes. The actual bias is always unknown. The best we can do is approximate it with the help of a statistical model."*
>
> We carefully went over the entire paper again to make sure that these expressions were used in a consistent way (including in the abstract, figures and tables).

2. ii) the results (table1) were made only with highest aggregation time so what do the authors mean by "after accounting for differences in scale"?

The expression "after accounting for differences in scale" has been replaced by *"after accounting for the differences in mean and variance"* and *"after accounting for their relative variability"*, which is more precise and easier to understand.

3. i) L429-455: Definition of the "conditional bias" with respect to the rainfall intensity is not clear. In L434, it seems that it refers to the "log ratio of (R_g/R_r)" with respect to the rainfall intensity, which is not the same as the modelled bias b shown in Eq 1, yet the results were described in terms of "bias"

Indeed, some additional details were needed here to help the reader understand the approach. The new explanations should be clearer:

*"Conditional biases are detected and quantified on the basis of the multiplicative bias model in Equations (1) and (2). If our assumptions are correct and there is no conditional bias, Equation (2) tells us that the average log-ratio between rain gauge and radar estimates should be a Gaussian random variable with constant mean and variance. Moreover, this result must hold independently of the rainfall intensity R_g(t). To detect the presence of a conditional bias in the G/R ratio, we therefore plot the values of ln(R_g(t)/R_r(t)) vs R_g(t) (at the highest available temporal resolution) and calculate the slope of the corresponding regression line, as shown in Figure 7. If the slope is positive, the bias increases with intensity. The relative rate of increase (in percentage) of the G/R ratio per mm/h is then given by 100(exp(m)-1), where m is the slope of ln(R_g(t)/R_r(t)) vs R_g(t)."*

3. ii) in Figure 7, was 0 mm (no rain) is also included and if so why? How would the error bar for the fit (or something that tells a significance level of the fit) look in this graph?

Zero values were not included as it is impossible to compute the log-ratios for them. Confidence intervals were not calculated. The low goodness of fit is not really an issue here as we are not interested in predicting the actual log-ratio but only in estimating the general tendency (i.e., the slope of the regression line). Obviously, the variance is larger for low rainfall rates. However, this is mostly indicative of measurement noise/uncertainty rather than bias. What really matters is that the slope for larger rainfall rates seems to be relatively well captured. Note that the statistical significance of the slope (at the 5% level) was assessed by applying a standard t-test on the fitted slope parameter obtained by least squares. Fitting and testing was done using the function lm() of the statistical software R.

3. iii) L438, this looks the slope of red line, does it make sense to use %?

This appears to be a misunderstanding. The values given in percentages are not the slopes but the relative rates of increase of the G/R ratio (in linear space, per mm/h). An additional sentence has been added to the text to explain how these values were inferred.

*"The relative rate of increase (in percentage) of the G/R ratio per mm/h is then given by 100(exp(m)-1), where m is the slope of ln(R_g(t)/R_r(t)) vs R_g(t)."*

3. iv) It is not easy to follow L442-L444, can the authors explain better where we see such evidence; e.g., the differences in apparent biases shown in Figure 6 look small among Denmark (1.59), Sweden (1.66), Finland (1.56) and the NL (1.4).

Some clarifications were added to the text. The effect of the conditional bias on the overall G/R ratio is rather modest because high rainfall intensities are rare. This is why it is important to also look at peak rainfall intensity bias.

*"The fact that both the Danish and Swedish products have large conditional biases also explains why their overall bias (as measured by the G/R ratio without conditioning on intensity) is slightly larger than for the Netherlands*

*and Finland. However, since large rainfall intensities are rare, the net effect of the conditional bias on the overall G/R ratio remains rather small."*

4. Section 3.6: Although the authors introduced the comparison with X-band radar, in L248-250, "the X-band data can be used to provide valuable insight into the advantages and challenges associated with using high-resolution X-band radar measurement in times of heavy rain", its analyses (L582-587) are poorly written and hardly provide insight as they aimed. If this radar is dual-pol, the results could have calculated with Kdp to prove their speculations (L583-585). It is also questionable in conclusion on the X-band data for Denmark (L640).

> The reviewer has a good point. The dual-pol capabilities of the Danish X-band radar have not been fully exploited yet and the paper would undoubtedly benefit from such additional analyses. However, there was simply not enough time (and funding) to look into this issue within MUFFIN. The reality is that there is currently no easy way to estimate R from Kdp using the Danish X-band radar without investing a substantial amount of work and money into the radar software. We hope to be able to do this one day as part of a follow-up project. But since MUFFIN ended in 2019, unfortunately, there is not much that we can currently do about this issue. We have added a couple of sentences to the text to better explain this.

> *"Unfortunately, the current software of the Danish X-band radar does not offer the possibility to estimate R from Kdp yet. The improvements due to switching from Z to Kdp could therefore not be assessed within the context of this study. Similarly, KNMI and DMI are currently working on better exploiting the new polarimetric capabilities of their C-band radars to better account for natural variations in the raindrop size distributions. However, these upgrades still require more research and could not be assessed formally here."*

5 i) L547-550: where do we see that the "multiplicative biases between radar and gauges can amplify when data are aggregated to coarser time scales?" If the authors meant the "multiplicative bias" as the "peak intensity bias", we still see that the bias values get smaller with coarser aggregation time (Figure 10). ii) Similarly, it is not clear what this means in L550 that "single-radar products with daily (coarser than sub-hourly) rain gauge adjustments are more vulnerable to error amplification". Can the authors provide a clearer example of error amplification and when this gets more vulnerable?

> The amplification relates to the peak intensity biases (PIB) for individual events. The point here is that while the average PIB (over a large number of events) decreases with aggregation time scale (as shown in Figure 9), there can be substantial increases in PIB at sub-hourly scales aggregation time scales within individual events (as shown in Figure 10). Since this may not have been 100% clear, we have added some additional details to the text to explain this.

> *"These examples [in Figure 10] show that even though globally speaking, the average peak intensity bias between radar and gauges converges to the average G/R ratio when the data are aggregated to coarser time scales (as shown in Figure 9), this might not always be the case locally and does not necessarily apply to all events. The reason for this is that the PIB depends on a multitude of confounding factors (e.g., calibration errors, natural variations in drop size distributions, range effects, wind, vertical variability, attenuation, etc...). When individual sources of error depend on each other or exhibit significant auto-correlation, their combined effect might cause the PIB to (locally) increase with aggregation time scale. In particular, strongly auto-correlated sources of bias such as changing drop size distributions, signal attenuation or wind effects can cause the PIB to increase with aggregation time scale."*

> *"Another important finding of our study is that single-radar products with daily rain gauge adjustments are more likely to contain increasing PIBs with aggregation time scale than composite products with hourly bias corrections. This makes sense as mean field bias adjustments can (partly) compensate for the bias in rainfall rate due to deviations from the Marshall-Palmer drop size distribution in the Z-R relationship. Similarly, radar compositing can mitigate the bias due to environmental factors such as range effects, vertical variability and attenuation. To show this, we computed, for each event, the time scale at which peak intensity bias reaches its maximum value. Figure 11 shows that in Denmark, 21/50 events exhibited a maximum PIB at a scale larger than*

*that of the highest available temporal resolution. Similarly, for the Swedish radar product, 26/50 cases of locally increasing peak intensity biases with aggregation time scale could be identified. By contrast, the Finnish and Dutch radar products, which make use of compositing and more frequent bias adjustments, only contained 14 and 8 such events, respectively. Further analysis reveals that most of the events with locally amplifying PIBs consist of two or more rainfall peaks separated by 10-30min, with rapidly fluctuating rainfall intensities between them (i.e., high intermittency). Some events with single rainfall peaks during which radar strongly underestimated rainfall rates for two or more time steps in a row were also identified. However, due to the limited temporal autocorrelation in heavy rain, most peak intensity bias values reached their maximum at time scales of 30 minutes or less."*

6. L551-559: Despite the explanation of Fig.11, it is not clearly written how significantly this can affect to the choice of hydrological models for flood prediction (L549). Also, in L558-559, how this should be interpreted for both bias correction and reduction to be applied for hydrological models?

We can't really make any concrete recommendations here as each event/catchment will be different and decisions must be made on a case-by-case basis. Still, we think it's important to point out that peak intensity biases in radar products can increase with aggregation time scale (which is counterintuitive) and that this might negatively affect the performance of hydrological models (especially in terms of their ability to predict the timing and intensity of flood peaks).

7. L582-587: Improve writing with clearer message.

The corresponding paragraphs have been rewritten to improve the flow of information and clarity of explanations.

*"The conditional bias with intensity affects the accuracy of the X-band radar in times of heavy rain, leading to high peak intensity biases. Figure 12d) shows that the median peak intensity bias at 5 min is 1.64 (39%) with 10% of the PIBs exceeding 3.1 (67.7%). One reason for this could be attenuation, which is known to play a major role at X-band. However, all reflectivity measurements have been corrected for attenuation prior to rainfall estimation. Also, Figure 12c) shows that there is no obvious change in the G/R ratio with the distance to the radar, as would be expected for attenuated signals. This leads us to conclude that similarly to the Danish and Swedish C-band products, the conditional bias with intensity is likely caused by the use of a fixed Z-R relation (together with daily bias adjustments). It also means that higher resolution alone is probably not enough to avoid strong conditional biases with intensity. The latter must be mitigated by other means, for example by replacing the fixed Z-R relationship with a R(Kdp) estimate in times of heavy rain or by performing more frequent bias adjustments with the help of gauges. Unfortunately, the current software of the Danish X-band radar does not offer the possibility to estimate R from Kdp yet. The improvements due to switching from Z to Kdp could therefore not be assessed within the context of this study. Similarly, KNMI and DMI are currently working on better exploiting the new polarimetric capabilities of their C-band radars to better account for natural variations in the raindrop size distributions. However, these upgrades still require more research and could not be assessed formally here."*

**Minor comments**

1. L112: Time period for KNMI gauges (2003-2017) does not match with those indicated in Table 1.

Thanks! The time period has been changed to 2008-2018 to match the information in Table 1.

2. L278-279: does this "[-]" mean unitless? Is it necessary?

Yes, [-] indicates unitless. This is standard notation.

3. L421-422: "Moreover,.." What does this sentence mean?

This paragraph has been rewritten to improve clarity and syntax.

*"The bias adjustment factor combines all these different factors together, which leads to a fairer comparison of the different radar products. The fact that the theoretical bias after accounting for differences in mean and variance might be as low as 10% (despite what the G/R ratio suggests) and that products with higher spatial/temporal resolutions seem to be affected by lower biases (in absolute value) is quite encouraging. However, one has to keep in mind that the representativity of beta strongly depends on the adequacy of the model proposed in Equation (1). Further analyses presented in the next section show that some of these assumptions might not be very realistic."*

4. L535-536 (Figure 10). Because the presented case is the same as shown in Figure 4, the peak intensity bias values and unit at the first interval (at the highest temporal resolution) are expected to be identical as those mentioned in L373-374 for each event. However, as seen right Y-axis in Fig. 10, the "peak intensity bias" is expressed in %, i.e. 100*(1- 1/(factor))". Explain this value (technically it is not the "peak intensity bias") better in the text, and revise the sentence.

      Thanks for pointing this out. We have changed the y-axis of the graph on the right to show the actual PIB values. The corresponding relative errors (in percentages) are given in the text.

5. Figure 11, revise the title of the X-axis.

      Done.

**Reviewer 2:**

**Minor comments:**

- line 395. "... Denmark (38.8%)" this value should be "37.3%" according to Fig 5.

      Thanks for spotting this typo. We have updated the number to 37.3%

[revised manuscript text omitted]